# A Theoretical Analysis of Self-Supervised Learning for Vision Transformers

**Yu Huang**[†,*], **Zixin Wen**[‡,*], **Yuejie Chi**[‡], **Yingbin Liang**[§]
[†]University of Pennsylvania  [‡]Carnegie Mellon University  [§]The Ohio State University

## Abstract

Self-supervised learning has become a cornerstone in computer vision, primarily divided into reconstruction-based methods like masked autoencoders (MAE) and discriminative methods such as contrastive learning (CL). Recent empirical observations reveal that MAE and CL capture different types of representations: CL tends to focus on global patterns, while MAE adeptly captures **both global and subtle local** information simultaneously. Despite a flurry of recent empirical investigations to shed light on this difference, theoretical understanding remains limited, especially on the dominant architecture **vision transformers** (ViTs). In this paper, to provide rigorous insights, we model the visual data distribution by considering two types of spatial features: dominant global features and comparatively minuscule local features, and study the impact of imbalance among these features. We analyze the training dynamics of one-layer softmax-based ViTs on both MAE and CL objectives using gradient descent. Our analysis shows that as the degree of feature imbalance varies, ViTs trained with the MAE objective effectively learn both global and local features to achieve near-optimal reconstruction, while the CL-trained ViTs favor predominantly global features, even under mild imbalance. These results provide a theoretical explanation for distinct behaviors of MAE and CL observed in empirical studies.

## 1 Introduction

Self-supervised learning (SSL) has been a leading approach to pretrain neural networks for downstream applications since the introduction of BERT (Devlin et al., 2018) and GPT (Radford et al., 2018) in natural language processing (NLP). On the other hand, in vision, self-supervised learning focused more on *discriminative* methods, which include contrastive learning (CL) (He et al., 2020; Chen et al., 2020) and non-contrastive learning methods (Grill et al., 2020; Chen et al., 2020; Caron et al., 2021; Zbontar et al., 2021). Inspired by masked language models in NLP and the seminal work of vision transformers (ViTs) (Dosovitskiy et al., 2020), *generative* approaches, such as masked reconstruction-based methods, have gained prominence in self-supervised vision pretraining. The masked autoencoders (MAE) (He et al., 2022) and SimMIM (Xie et al., 2022) have demonstrated the effectiveness of visual representation learning via reconstruction-based objectives.

Contrastive learning-like objectives promote instance discrimination among samples in the same batch of training. With suitable data augmentation, CL returns well-trained vision encoders like CLIP (Radford et al., 2021) and DINO (Caron et al., 2021) that can serve as backbones for state-of-the-art multimodal large language models (MLLMs) (Tong et al., 2024). Masked reconstruction objectives (e.g., MAE), on the other hand, enforce neural networks to reconstruct some or all patches of an image given masked inputs. In practice, the MAE-like approach proves to have intriguing generalization properties that differ significantly from the behaviors in CL. The seminal work (He et al., 2022) showed that MAE can visibly conduct visual reasoning to fill missing patches even under very high masking rates. Some critical observations from recent research (Wei et al., 2022b; Park et al., 2023; Xie et al., 2023) provide comparative studies of these SSL approaches. They concluded that the ViTs trained via generative objectives display **diverse attention patterns**: different query patches pay attention to distinct local areas. This is in sharp contrast to the discriminative approaches,

---

*The first two authors contributed equally.

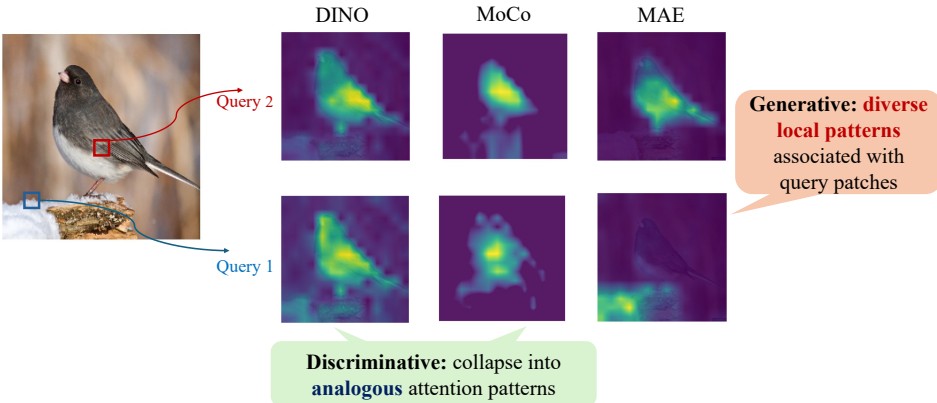

Figure 1: Visualization of attention maps in the last layer of ViT for query patches from two different spatial locations, similar to those presented in Park et al. (2023). The ViTs were trained by the generative self-supervised learning approach of masked reconstruction (MAE) and discriminative methods: DINO (Caron et al., 2021) and MoCo (Chen et al., 2021b).

whose attention heads focus primarily on the most significant global pattern regardless of where the query patches are, as shown in Figure 1. These empirical observations motivate the question: from a *theoretical* standpoint, how do ViTs pick up these observed attention patterns during the training process, respectively for different SSL methods?

Despite extensive empirical efforts of studying SSL in vision pretraining, its theoretical understanding is still nascent. Most existing theories of SSL focused on the discriminative approach (Arora et al., 2019; Chen et al., 2021a; Robinson et al., 2021; HaoChen et al., 2021; Tian et al., 2021; Wang et al., 2021; Wen & Li, 2021; 2022), especially (non-)contrastive learning. There are also a few attempts towards understanding methods using the generative approach like masked reconstructions (Cao et al., 2022; Zhang et al., 2022; HaoChen et al., 2021; Pan et al., 2022), which mainly adapt the theories developed for CL to their context. In fact, there are two major limitations of these prior works: *i)* **Transformers**, as the dominant architecture in practice, were not studied in the aforementioned works of self-supervised learning and *ii)* there still lacks a suitable theoretical framework that can provide convincing explanations for the empirical findings in Park et al. (2023); Xie et al. (2023), especially on the difference of the attention patterns learned by the different approaches of SSL. The above limitations highlight a significant gap in the literature on SSL for vision pretraining[1].

Motivated by the limited theoretical characterization of SSL for vision with transformers, especially in comparing CL and masked reconstruction objectives, we aim to address the following research questions:

---
**Our Research Questions**

Can we *theoretically* characterize the solutions that ViTs converge to in these two mainstream self-supervised learning approaches? How do differences in attention patterns emerge during their respective training processes?

---

**Contributions.** In this paper, we take a step toward answering the above questions. We study the gradient descent (GD) training process of one-layer softmax-based ViTs for both masked reconstruction and contrastive learning, focusing on **spatially structured data distributions** generalized from supervised learning settings (Jelassi et al., 2022). In our setting, each image is sampled from distinct clusters characterized by unique patch-wise feature associations. Each cluster contains two types of features: a large portion of patches reside in a global area and share global features, while the remaining local areas contain relatively few patches with their own local features. We measure the imbalance of feature distribution by a condition called the information gap $\Delta$, which is defined in eq. (4.1). Under such setting:

---
[1]More detailed discussions for related work can be found in Appendix A.

1. We provide global convergence guarantees for training ViTs on both the MAE and the CL loss fucntions. To the best of our knowledge, this is the first end-to-end guarantee for learning ViTs with self-supervised learning objectives;

2. We provide a comprehensive characterization of the training dynamics of *attention correlations* (see Definition 3.1) to illustrate the attention patterns to which ViTs converge: i). MAE provably learns **diverse** attention patterns, with each patch concentrating its attention on its designated area based on its position, even under a substantial information gap $\Delta$; ii). CL primarily learns a **global** attention pattern, causing all patches to focus on the global area regardless of their locations, even with a minor $\Delta$. These qualitative differences in the solutions learned by these two SSL methods provide strong theoretical support for the empirical behavior gaps observed in Park et al. (2023); Xie et al. (2023), and highlight the theoretical advantage of MAE in handling highly imbalanced data structures.

**Notation.** We introduce notations to be used throughout the paper. For any two functions $h(x)$ and $g(x)$, we use $h(x) = \Omega(g(x))$ (resp. $h(x) = O(g(x))$) to denote that there exist some universal constants $C_1 > 0$ and $a_1$, s.t. $|h(x)| \geq C_1|g(x)|$ (resp. $|h(x)| \leq C_1|g(x)|$) for all $x \geq a_1$; Furthermore, $h(x) = \Theta(g(x))$ indicates $h(x) = \Omega(g(x))$ and $h(x) = O(g(x))$ hold simultaneously. We use $\mathbb{1}\{\cdot\}$ to denote the indicator function, and let $[N] := \{1, 2, \ldots, N\}$. We use $\widetilde{O}$, $\widetilde{\Omega}$, and $\widetilde{\Theta}$ to further hide logarithmic factors in the respective notations. We use $\mathrm{poly}(P)$ and $\mathrm{polylog}(P)$ to represent large constant-degree polynomials of $P$ and $\log(P)$, respectively.

## 2 PROBLEM SETUP

In this section, we present our problem formulations for studying the training process of ViTs in self-supervised pretraining. We begin with some background information, followed by a description of our data distribution. We then detail the pretraining strategies using MAE and CL respectively with the specific transformer architecture considered in this paper.

### 2.1 BACKGROUND ON SELF-SUPERVISED LEARNING

**Masked reconstruction-based learning.** We follow the masked reconstruction frameworks in He et al. (2022); Xie et al. (2022). Each data sample $X \in \mathbb{R}^{d \times P}$ has the form $X = (X_{\mathbf{p}})_{\mathbf{p} \in \mathcal{P}}$, which has $|\mathcal{P}| = P$ patches, and each patch $X_{\mathbf{p}} \in \mathbb{R}^d$. Given a collection of images $\{X_i\}_{i \in [n]}$, we select a masking set $\mathcal{M}_i \subset \mathcal{P}$ for each image $X_i$, and mask these patches to a uniform value $\mathsf{M} \in \mathbb{R}^d$. The resulting masked images $\{\mathsf{M}(X_i)\}_{i \in [n]}$ are given by

$$\mathsf{M}(X_i)_{\mathbf{p}} = \left\{ \begin{array}{ll} [X_i]_{\mathbf{p}} & \mathbf{p} \in \mathcal{U}_i \\ \mathsf{M} & \mathbf{p} \in \mathcal{M}_i \end{array} \right. , \qquad i \in [n], \tag{2.1}$$

where $\mathcal{U}_i = \mathcal{P} \setminus \mathcal{M}_i$ is the index set of unmasked patches. Let $F : X \mapsto \widehat{X}$ be an architecture that outputs a reconstructed image $\widehat{X} \in \mathbb{R}^{d \times P}$ for any given input $X \in \mathbb{R}^{d \times P}$. The pretraining objective is then defined as the mean-squared reconstruction loss over a series of subsets $\mathcal{P}'_i \subset \mathcal{P}$ of the image as follows:

$$\mathcal{L}_{\mathtt{masked}}(F) = \frac{1}{n} \sum_{i=1}^{n} \sum_{\mathbf{p} \in \mathcal{P}'_i} \left\| [X_i]_{\mathbf{p}} - [F(\mathsf{M}(X_i))]_{\mathbf{p}} \right\|_2^2. \tag{2.2}$$

MAE (He et al., 2022) chose the subset $\mathcal{P}'_i$ as the set of masked patches $\mathcal{M}_i$, whereas SimMIM (Xie et al., 2022) aimed to reconstruct the full image $\mathcal{P}'_i = \mathcal{P}$. We do not explore the trade-offs between these two approaches in our study.

**Contrastive learning.** Contrastive learning (Chen et al., 2020) aims to learn meaningful representations $F$ by distinguishing between similar and dissimilar data points. For a given batch $\{X_i\}_{i \in [n]}$, we generate a positive pair $(X_i^{(1)}, X_i^{(2)})$ for each $i$ by applying random augmentations to $X_i$. Negative pairs $(X_i^{(1)}, X_j^{(2)})$ for $j \neq i$ are formed from different data points. The model $F$ is trained to minimize the following contrastive loss:

$$\mathcal{L}_{\mathtt{contrastive}}(F) = \frac{1}{n} \sum_{i=1}^{n} \left[ -\tau \log \left( \frac{e^{\mathsf{Sim}_F\left(X_i^{(1)}, X_i^{(2)}\right)/\tau}}{\sum_{j \in [n]} e^{\mathsf{Sim}_F\left(X_i^{(1)}, X_j^{(2)}\right)/\tau}} \right) \right], \tag{2.3}$$

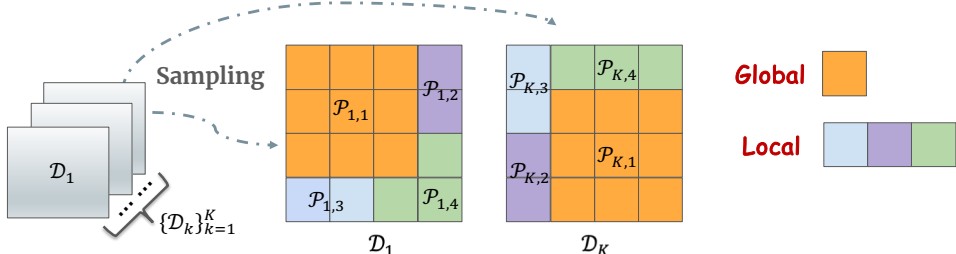

Figure 2: Illustration of our data distribution (see Definition 2.1). Each cluster $\mathcal{D}_k$ is segmented into distinct areas $\mathcal{P}_{k,j}$, with squares in the same color representing the same area $\mathcal{P}_{k,j}$. The global area $\mathcal{P}_{k,1}$ (depicted in orange) contains a larger count of patches compared to any other local areas. It is important to note that while we use spatially contiguous partitions for clarity in this illustration, our data model is also applicable to non-contiguous cases.

where $\mathsf{Sim}_F$ measures the similarity between two representations, and $\tau$ is a temperature parameter controlling the sharpness of the distribution.

## 2.2 DATA DISTRIBUTION

We assume the data samples $X \in \mathbb{R}^{d \times P}$ are drawn independently based on some data distribution $\mathcal{D}$. To capture the *feature-position (FP) correlation* in the learning problem, we consider the following setup for vision data. We assume that the data distribution consists of many different clusters, where each cluster captures a distinct spatial pattern, and hence is defined by a different partition of patches with a different set of visual features. We define the data distribution $\mathcal{D}$ formally as follows. An intuitive illustration of data generation is given in Figure 2.

**Definition 2.1** (Data distribution $\mathcal{D}$). The data distribution $\mathcal{D}$ has $K = O(\mathrm{polylog}(P))$ different clusters $\{\mathcal{D}_k\}_{k=1}^K$. For every cluster $\mathcal{D}_k, k \in [K]$, there is a corresponding partition of $\mathcal{P}$ into $N_k$ disjoint subsets $\mathcal{P} = \bigcup_{j=1}^{N_k} \mathcal{P}_{k,j}$ which we call **areas**. For each sample $X = (X_{\mathbf{p}})_{\mathbf{p} \in \mathcal{P}}$, its sampling process is as follows:

- We draw $\mathcal{D}_k$ uniformly at random from all clusters and draw a sample $X$ from $\mathcal{D}_k$.

- Given $k \in [K]$, for any $j \in [N_k]$, all patches $X_{\mathbf{p}}$ in the area $\mathcal{P}_{k,j}$ are given the same content $X_{\mathbf{p}} = v_{k,j} z_j(X)$, where $v_{k,j} \in \mathbb{R}^d$ is the *visual* feature and $z_j(X)$ is the latent variable. We assume $\bigcup_{k=1}^K \bigcup_{j=1}^{N_k} \{v_{k,j}\}$ are orthogonal to each other with unit norm.

- Given $k \in [K]$, for any $j \in [N_k]$, $z_j(X) \in [L, U]$, where $0 \leq L < U$ are on the order of $\Theta(1)$.[2]

**Global and local features, and empirical observations in prior works.** Image data naturally contains two types of features: the global features and the local features. For instance, in an image of an object, global features can capture the shape and texture of the object, such as the fur color of an animal, whereas local features describe specific details of local areas, such as the texture of leaves in the background. Recent empirical studies on self-supervised pretraining with ViTs (Park et al., 2023; Wei et al., 2022b) and observations in Figure 1 collectively show that masked pretraining exhibits the capacity to avoid attention collapse concentrating towards those global shapes by identifying diverse local attention patterns. Consequently, unraveling their mechanisms necessitates a thorough examination of data characteristics that embody both global and local features. In this paper, we characterize these two types of features by the following assumption on the data.

**Assumption 2.2** (Global feature vs local feature). Let $\mathcal{D}_k$ with $k \in [K]$ be a cluster from $\mathcal{D}$. We let $\mathcal{P}_{k,1}$ be the **global area** of cluster $\mathcal{D}_k$, and all the other areas $\mathcal{P}_{k,j}, j \in [N_k] \setminus \{1\}$ be the **local areas**. Since each area corresponds to an assigned feature, we also call them the *global* and *local* features, respectively. Moreover, we assume:

- Global area: given $k \in [K]$, we set $C_{k,1} = |\mathcal{P}_{k,1}| = \Theta(P^{\kappa_c})$ with $\kappa_c \in [0.5005, 1]$, where $C_{k,1}$ is the number of patches in the global area $\mathcal{P}_{k,1}$.

---

[2]The distribution of $z_j(X)$ can be arbitrary within the above support set.

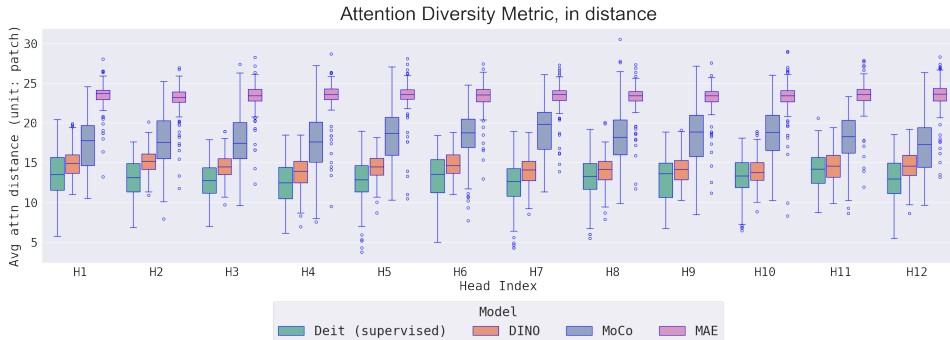

Figure 3: **Attention Diversity Metric:** We design a novel empirical metric, the **attention diversity metric**, to probe the last layer of ViTs trained by masked reconstructions (MAE), CL (MoCo), another discriminative SSL approach (DINO), and supervised learning (DeiT). Lower values of this metric signify focused attention on a similar area across different patches, reflecting a global pattern of focus. Conversely, higher values suggest that attention is dispersed, focusing on different, localized areas. The results show that the MAE model excels in capturing *diverse local patterns* compared to discriminative methods like CL. (see Appendix B for details).

- Local area: given $k \in [K]$, we choose $C_{k,j} = \Theta(P^{\kappa_s})$ with $\kappa_s \in [0.001, 0.5]$ for $j > 1$, where $C_{k,j}$ denotes the number of patches in the local area $\mathcal{P}_{k,j}$.

The rationale for defining the global feature in this manner stems from observing that patches representing global features ($C_{k,1}$) typically occur more frequently than those representing local features ($C_{k,j}$, for $j > 1$), since global features capture the primary visual information of an image, offering a dominant view, while local features focus on subtler details within the image. Our empirical observations (see Figure 3) further substantiate the significance of distinguishing between global and local patterns in data distributions, which is essential for elucidating the distinct behaviors exhibited by MAE and CL.

### 2.3 MASKED RECONSTRUCTION WITH TRANSFORMERS

**Transformer architecture.** A transformer block (Vaswani et al., 2017; Dosovitskiy et al., 2020) consists of a self-attention layer followed by an MLP layer. The self-attention layer has multiple heads, each of which consists of the following components: a query matrix $W^Q$, a key matrix $W^K$, and a value matrix $W^V$. Given an input $X$, the output of one head in the self-attention layer can be described by the following mapping:

$$G(X; W^Q, W^K, W^V) = \text{softmax}\left((W^Q X)^\top W^K X\right) \cdot (W^V X)^\top, \tag{2.4}$$

where the $\text{softmax}(\cdot)$ function is applied row-wisely and for a vector input $z \in \mathbb{R}^P$, the $i$-th entry of $\text{softmax}(z)$ is given by $\frac{\exp(z_i)}{\sum_{s=1}^{P} \exp(z_s)}$.

To simplify the theoretical analysis, we consolidate the product of query and key matrices $(W^Q)^\top W^K$ into one weight matrix denoted as $Q$. Furthermore, we set $W^V$ to be the identity matrix and fixed during the training. These simplifications are often taken in recent theoretical works (Jelassi et al., 2022; Huang et al., 2023; Zhang et al., 2023a) in order to allow tractable analysis. With these simplifications in place, eq. (2.4) can be rewritten as

$$G(X; Q) = \text{softmax}\left(X^\top Q X\right) \cdot X^\top. \tag{2.5}$$

Input tokens in transformers are indistinguishable without explicit spatial information. Therefore, positional encodings should be added to the input embeddings to retain this crucial positional context as in practices (Dosovitskiy et al., 2020; He et al., 2022). Our assumptions regarding the positional encodings are as follows:

**Assumption 2.3** (Positional encoding). We assume fixed positional encodings, which is consistent with the implementation in MAE (He et al., 2022): $E = (e_{\mathbf{p}})_{\mathbf{p} \in \mathcal{P}} \in \mathbb{R}^{d \times P}$ where positional embedding vectors $e_{\mathbf{p}}$ are orthogonal to each other and to all the features $v_{k,j}$, and are of unit-norm.

We now include positional embeddings in eq. (2.5) and introduce the network architecture for masked reconstruction used in this study.

**Definition 2.4** (ViTs for MAE). We assume that our vision transformer $F^{\mathtt{mae}}(X;Q)$ consists of a single-head self-attention layer with an attention weight matrix $Q \in \mathbb{R}^{d \times d}$. For an input image $X \sim \mathcal{D}$, we add positional encoding by letting $\widetilde{X} = X + E$. The attention score from patch $X_{\mathbf{p}}$ to patch $X_{\mathbf{q}}$ is denoted by

$$\mathbf{attn}_{\mathbf{p} \to \mathbf{q}}^{\mathtt{m}}(X;Q) := \frac{e^{\widetilde{X}_{\mathbf{p}}^{\top} Q \widetilde{X}_{\mathbf{q}}}}{\sum_{\mathbf{r} \in \mathcal{P}} e^{\widetilde{X}_{\mathbf{p}}^{\top} Q \widetilde{X}_{\mathbf{r}}}}, \quad \text{for } \mathbf{p}, \mathbf{q} \in \mathcal{P}. \tag{2.6}$$

Then the output of the transformer is given by

$$[F^{\mathtt{mae}}(X;Q)]_{\mathbf{p}} = \sum_{\mathbf{q} \in \mathcal{P}} \mathbf{attn}_{\mathbf{p} \to \mathbf{q}}^{\mathtt{m}}(X;Q) \cdot X_{\mathbf{q}}, \quad \text{for } \mathbf{p} \in \mathcal{P}. \tag{2.7}$$

Then we formally define the masking operation and the objective for our masked pretraining task.

**Definition 2.5** (Random masking). Let $\mathsf{M}(X) \to \mathbb{R}^{d \times P}$ denote the random masking operation, which randomly selects (without replacement) a subset of patches $\mathcal{M}$ in $X$ with a masking ratio $\gamma = \Theta(1) \in (0,1)$ and masks them to be $\mathsf{M} := \mathbf{0} \in \mathbb{R}^{d}$. The masked samples obey eq. (2.1).

**MAE objective.** To train the model $F^{\mathtt{mae}}(\mathsf{M}(X);Q)$, following the methodology described in MAE practice (He et al., 2022), we minimize the squared reconstruction error in eq. (2.2) only on masked patches, where $\mathsf{M}(X)$ follows Def. 2.5. The training objective thus can be written as

$$\mathcal{L}_{\mathtt{mae}}(Q) := \frac{1}{2}\mathbb{E}\left[\sum_{\mathbf{p} \in \mathcal{P}} \mathbb{1}\{\mathbf{p} \in \mathcal{M}\}\left\|[F^{\mathtt{mae}}(\mathsf{M}(X);Q)]_{\mathbf{p}} - X_{\mathbf{p}}\right\|^{2}\right], \tag{2.8}$$

where the expectation is with respect to both the data distribution and the masking. Note that our objective remains highly nonconvex with the model defined in Definition 2.4.

**Training algorithm.** The learning objective in eq. (2.8) is minimized via GD with learning rate $\eta > 0$. At $t = 0$, we initialize $Q^{(0)} := \mathbf{0}_{d \times d}$ as the zero matrix. The parameter is updated as follows:

$$Q^{(t+1)} = Q^{(t)} - \eta \nabla_Q \mathcal{L}_{\mathtt{mae}}(Q^{(t)}).$$

Note that the initialization of $Q^{(0)}$ results in any query patch uniformly attending to all patches.

## 2.4 Contrastive Learning with transformers

The transformer architecture used for CL is similar to that of MAE, but with a minor modification to accommodate contrastive loss, as outlined below.

**Definition 2.6** (ViTs for CL). We consider a vision transformer $F^{\mathtt{cl}}(X;Q)$ consisting of a single-head self-attention layer with an attention weight matrix $Q \in \mathbb{R}^{d \times d}$. For an input image $X$, the attention score from patch $X_{\mathbf{p}}$ to patch $X_{\mathbf{q}}$ is denoted by

$$\mathbf{attn}_{\mathbf{p} \to \mathbf{q}}^{\mathtt{c}}(X;Q) := \frac{e^{e_{\mathbf{p}}^{\top} Q X_{\mathbf{q}}}}{\sum_{\mathbf{r} \in \mathcal{P}} e^{e_{\mathbf{p}}^{\top} Q X_{\mathbf{r}}}}, \quad \text{for } \mathbf{p}, \mathbf{q} \in \mathcal{P}. \tag{2.9}$$

The output of the transformer is then computed as

$$F^{\mathtt{cl}}(X;Q) = \frac{1}{P} \sum_{\mathbf{p}, \mathbf{q} \in \mathcal{P}} \mathbf{attn}_{\mathbf{p} \to \mathbf{q}}^{\mathtt{c}}(X;Q) \cdot X_{\mathbf{q}} \quad \in \mathbb{R}^{d}. \tag{2.10}$$

which represents the average pooling of all the patches.

The key distinction is that we separate the positional and patch embeddings within the attention mechanism for technical simplicity. However, it is important to emphasize that these two types of embeddings remain coupled for attention calculations.

**Definition 2.7** (Data augmentation). For a sample $X \in \mathbb{R}^{d}$, we generate two new samples $X^{+}$ and $X^{++}$ by independently applying random masking as in Def. 2.5 with a ratio $\gamma_0 = \Theta(1)$, similar to the crop-resize operations used in practice. The unmasked sets for them are denoted as $\mathcal{U}^{+}$ and $\mathcal{U}^{++}$.

**CL objective.** Given a sample $X$, we first generate a pair of positive samples $\{X^+, X^{++}\}$ via Def. 2.7. Then we generate a batch of i.i.d. negative samples $\mathfrak{N} = \{X^{-,s}\}_{s \in [N_c]}$. Denoting $\mathfrak{B} = \mathfrak{N} \cup \{X^{++}\}$, we minimize the expected contrastive loss in eq. (2.3) with $\ell_2$-regularization:

$$\mathcal{L}_{\mathtt{cl}}(Q) := \mathbb{E}_{X^+, X^{++}, \mathfrak{N}} \left[ -\tau \log \left( \frac{e^{\mathsf{Sim}_{F^{\mathtt{cl}}}(X^+, X^{++})/\tau}}{\sum_{X' \in \mathfrak{B}} e^{\mathsf{Sim}_{F^{\mathtt{cl}}}(X^+, X')/\tau}} \right) \right] + \frac{\lambda}{2} \|Q\|_F^2, \qquad (2.11)$$

where $\| \cdot \|_F$ denotes the Frobenius norm, $\lambda > 0$ is the regularization parameter, and the similarity of the representations of $X$ and $X'$ obtained by $F^{\mathtt{cl}}(\cdot; Q)$ is defined as

$$\mathsf{Sim}_{F^{\mathtt{cl}}}(X, X') := \left\langle F^{\mathtt{cl}}(X; Q), \mathsf{StopGrad}\left(F^{\mathtt{cl}}(X'; Q)\right)\right\rangle.$$

The $\mathsf{StopGrad}(\cdot)$ operator ensures that no gradient is computed for this term. Additionally, no augmentation is applied to the negative samples. Both practices are standard in the literature on the theory of contrastive learning (Wen & Li, 2021; 2022). Similar to MAE, we update $Q$ by GD with zero-initialization:

$$Q^{(t+1)} = Q^{(t)} - \eta \nabla_Q \mathcal{L}_{\mathtt{cl}}(Q^{(t)}). \qquad (2.12)$$

In the following, any variable with a superscript $^{(t)}$ represents that variable at the $t$-th step of training.

## 3 ATTENTION PATTERNS AND FEATURE-POSITION CORRELATIONS

To show the significance of the data distribution design and understand the nature of our self-supervised learning tasks, in this section, we will provide some preliminary implications of the spatial structures in Def. 2.1. Intuitively, for MAE, for a given cluster $\mathcal{D}_k$, to reconstruct a missing patch $\mathbf{p} \in \mathcal{P}_{k,j} \cap \mathcal{M}$, the attention head should exploit all *unmasked* patches in the *target* area $\mathcal{P}_{k,j}$ to find the same visual feature $v_{k,j}$ to fill in the blank, which emphasizes the *locality* for $\mathbf{p}$ in different areas. However, CL focuses on any discriminative patterns regardless of the location of $\mathbf{p}$, which can align positive pairs but may lead to collapsed attention patterns. We will elaborate on these points by describing the *area attentions* and illustrating the intuition about how they can be learned via *attention correlations* (Def. 3.1).

**Area attention.** We first define a new notation for a cleaner presentation. For $X \sim \mathcal{D}$ and $\mathbf{p} \in \mathcal{P}$, we write the attention of patch $X_{\mathbf{p}}$ to a subset $\mathcal{A} \subset \mathcal{P}$ of patches by

$$\widetilde{\mathbf{Attn}}^\dagger_{\mathbf{p} \to \mathcal{A}}(X; Q) := \sum_{\mathbf{q} \in \mathcal{A}} \mathbf{attn}^\dagger_{\mathbf{p} \to \mathbf{q}}(X; Q), \quad \text{for } \dagger \in \{\mathtt{m}, \mathtt{c}\}.$$

**MAE's ability to learn locality with ViTs.** Let us first explain why the above notion of area attention matters in understanding how attention works in masked reconstruction. Suppose we have a sample $X$ picked from $\mathcal{D}_k$, and the patch $X_{\mathbf{p}}$ with $\mathbf{p} \in \mathcal{P}_{k,j}$ is masked. Then the prediction of $X_{\mathbf{p}}$ given masked input $\mathsf{M}(X)$ can be written as

$$[F^{\mathtt{mae}}(\mathsf{M}(X); Q)]_{\mathbf{p}} = \sum_{\mathbf{q} \in \mathcal{P}} \mathsf{M}(X)_{\mathbf{q}} \cdot \mathbf{attn}^{\mathtt{m}}_{\mathbf{p} \to \mathbf{q}}(\mathsf{M}(X); Q)$$

$$= \sum_{i \in [N_k]} z_i(X) v_{k,i} \cdot \widetilde{\mathbf{Attn}}^{\mathtt{m}}_{\mathbf{p} \to \mathcal{U} \cap \mathcal{P}_{k,i}}(\mathsf{M}(X); Q) \text{ (since } \mathsf{M}(X)_{\mathbf{q}} = \mathbf{0} \text{ if } \mathbf{q} \in \mathcal{M}).$$

To reconstruct the original patch $X_{\mathbf{p}}$, the transformer should not only focus on the correct area $\mathcal{P}_{k,j}$, but must also prioritize attention to the *unmasked* patches within this area. This specificity is denoted by the area attention $\widetilde{\mathbf{Attn}}^{\mathtt{m}}_{\mathbf{p} \to \mathcal{U} \cap \mathcal{P}_{k,j}}$ over $\mathcal{U} \cap \mathcal{P}_{k,j}$, a requirement imposed by masking operations. We refer to these location-dependent attention patterns as **locality**.

To further explain how ViTs perform such prioritization, we introduce the following quantities, which capture the major insights of our analysis to distinguish between MAE and contrastive learning.

**Definition 3.1.** (Attention correlations) Let $\mathbf{p} \in \mathcal{P}$, and we define attention correlations as:

1. Feature-Position (FP) Correlation: $\Phi_{\mathbf{p} \to v_{k,m}} := e_{\mathbf{p}}^\top Q v_{k,m}$, for $k \in [K]$ and $m \in [N_k]$;

2. Position-Position (PP) Correlation: $\Upsilon_{\mathbf{p} \to \mathbf{q}} := e_{\mathbf{p}}^\top Q e_{\mathbf{q}}, \ \forall \mathbf{q} \in \mathcal{P}$.

Due to our (zero) initialization of $Q^{(0)}$, we have $\Phi^{(0)}_{\mathbf{p} \to v_{k,m}} = \Upsilon^{(0)}_{\mathbf{p} \to \mathbf{q}} = 0$.

These two types of attention correlations, FP correlation $\Phi_{\mathbf{p} \to v_{k,m}}$ and PP correlation $\Phi_{\mathbf{p} \to \mathbf{q}}$, act as the exponent terms within the $\mathrm{softmax}$ calculations for attention scores. Given $\mathbf{p} \in \mathcal{P}_{k,j}$ is masked, the (unnormalized) attention $\mathbf{attn}^{\mathrm{m}}_{\mathbf{p} \to \mathbf{q}}$ directed towards an *unmasked* patch $\mathbf{q}$ is influenced jointly by these correlations. Hence, the described attention pattern for MAE can emerge from either a substantial FP correlation $\Phi_{\mathbf{p} \to v_{k,j}}$ or a significant PP correlation $\Phi_{\mathbf{p} \to \mathbf{q}}$ for $\mathbf{q}$ in the same area as $\mathbf{p}$. However, in our setting, the latter mechanism—learning via PP correlation—fails to produce desired attention patterns: *i).* such a mechanism inadvertently directs attention to the *masked* patches, which is not desirable; *ii).* such position association could be vulnerable to the variation across different clusters, i.e., $\mathbf{p}, \mathbf{q} \in \mathcal{P}_{k,j}$ does not necessarily hold for all $k \in [K]$. This also highlights that prior work (Jelassi et al., 2022) that relied solely on pure positional attention cannot fully explain the ViTs' ability to learn locality when the patch-wise associations are not fixed.

**Why CL may fail to explore the locality.** Now turning to CL, for $\mathcal{X} \in \mathcal{D}_k$, we have the following form of similarity between the positive pair:

$$\langle F^{\mathrm{cl}}(X^+; Q), F^{\mathrm{cl}}(X^{++}; Q) \rangle$$
$$= \tfrac{1}{P^2} \sum_{\mathbf{p}, \mathbf{p'} \in \mathcal{P}} \sum_{i=1}^{N_k} \widetilde{\mathbf{Attn}}^{\mathrm{c}}_{\mathbf{p} \to \mathcal{U}^+ \cap \mathcal{P}_{k,i}}(X^+; Q) \widetilde{\mathbf{Attn}}^{\mathrm{c}}_{\mathbf{p'} \to \mathcal{U}^{++} \cap \mathcal{P}_{k,i}}(X^{++}; Q).$$

Thus, to align the positive representations effectively, the optimal strategy is also to direct attention toward a specific area for each patch $\mathbf{p}$, i.e., greedily ensuring that only one area attention $\widetilde{\mathbf{Attn}}^{\mathrm{c}}_{\mathbf{p} \to \mathcal{U}^+ \cap \mathcal{P}_{k,i}}$ is activated for some $i \in [N_k]$. However, the above expression suggests that the selected area by the optimal strategy may not necessarily depend on the location $\mathbf{p}$, which could lead to a collapsed attention scenario where all patches focus on the same area. Regarding attention correlations, the attention mechanism defined in eq. (2.9) requires us to handle only the FP correlations among different features for CL. Theorem 4.4 in the next section confirms that a collapsed solution indeed occurs: ViTs trained with CL concentrate attention on the global area across all patches by exclusively capturing global FP correlations across all patches, i.e., $\Phi_{\mathbf{p} \to v_{k,1}}$ becomes large for $\forall \mathbf{p} \in \mathcal{P}$.

## 4 STATEMENTS OF MAIN RESULTS

In this section, we present our main theorems on the learning processes of ViTs in MAE and CL. We begin by introducing notations that will be used in theorem presentations.

**Information gap and a technical condition.** Based on our data model in Section 2.2, we introduce a notion of *information gap* to quantify the degree of imbalance between global and local areas (cf. Assumption 2.2). Denoted as $\Delta$, the information gap is defined as follows:

$$\Delta := (1 - \kappa_s) - 2(1 - \kappa_c). \tag{4.1}$$

Broadly speaking, a larger $\Delta$ means that the number of global features is much greater than local ones, indicating a significant imbalance. In contrast, a smaller value reflects only a slight imbalance.[3]

**Unmasked area attention.** Based on the crucial role of those unmasked patches for both reconstruction task and positive contrastive pairs, we further define the *unmasked area attention* as follows:

$$\mathbf{Attn}^{\dagger}_{\mathbf{p} \to \mathcal{P}_{k,m}}(X; Q) := \widetilde{\mathbf{Attn}}^{\dagger}_{\mathbf{p} \to \mathcal{U} \cap \mathcal{P}_{k,m}}(X; Q), \text{ for } \dagger \in \{\mathrm{m}, \mathrm{c}\}.$$

### 4.1 MAE LEARNS DIVERSE ATTENTION PATTERNS

Our results are structured into two parts: *i).* analysis of convergence (Theorem 4.1), which includes the global convergence guarantee of the masked reconstruction loss and characterization of the attention pattern at the end of training to demonstrate the diverse locality; *ii).* learning dynamics of attention correlations (Theorem 4.2), which shows how transformers capture target FP correlations while downplaying PP correlations as discussed in Section 3.

To properly evaluate the reconstruction performance, we further introduce the following notion of the reconstruction loss with respect to a specific patch $\mathbf{p} \in \mathcal{P}$:

$$\mathcal{L}_{\mathrm{mae}, \mathbf{p}}(Q) = \frac{1}{2} \mathbb{E} \left[ \mathbb{1}\{\mathbf{p} \in \mathcal{M}\} \left\| [F^{\mathrm{mae}}(\mathsf{M}(X); Q)]_{\mathbf{p}} - X_{\mathbf{p}} \right\|^2 \right]. \tag{4.2}$$

---

[3]Our study focuses on the regime where $\Delta$ is not too close to zero, i.e., $|\Delta| = \Omega(1)$, which allows for cleaner induction arguments. This condition could be potentially relaxed via more involved analysis.

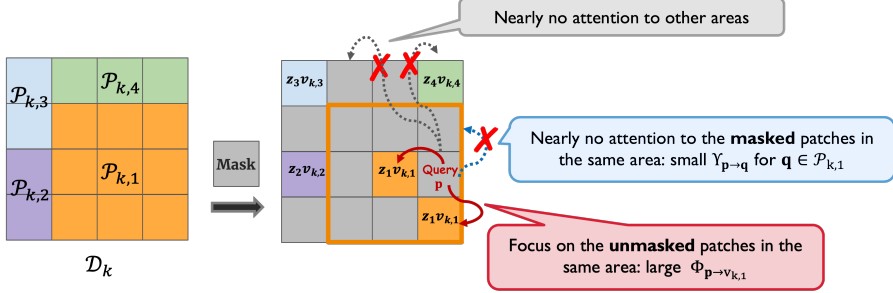

Figure 4: The mechanism of how the masked patch attends to other patches via attention correlations.

Now we present our first main result regarding the convergence of MAE.

**Theorem 4.1** (Training convergence). *Suppose the information gap* $\Delta \in [-0.5, -\Omega(1)] \cup [\Omega(1), 1]$. *For any* $0 < \epsilon < 1$, *suppose* $\mathrm{polylog}(P) \gg \log(\frac{1}{\epsilon})$. *We train the ViTs in Def. 2.4 by GD to minimize reconstruction loss in eq. (2.8) with* $\eta \ll \mathrm{poly}(P)$. *Then for each patch* $\mathbf{p} \in \mathcal{P}$, *we have*

1. *Loss converges:* $\mathcal{L}_{mae,\mathbf{p}}(Q^{(T^\star)}) - \mathcal{L}^\star_{mae,\mathbf{p}} \leq \epsilon$ *in* $T^\star = O\Big(\frac{1}{\eta} \log(P) P^{\max\{2(\frac{U}{L}-1),1\}(1-\kappa_s)} +$

   $\frac{1}{\eta\epsilon} \log\big(\frac{P}{\epsilon}\big)\Big)$ *iterations, where* $\mathcal{L}^\star_{mae,\mathbf{p}}$ *is the global minimum of the patch-level reconstruction loss in equation 4.2.*

2. **Area-wide** *pattern of attention: given cluster* $k \in [K]$, *and* $\mathbf{p} \in \mathcal{P}_{k,j}$ *for some* $j \in [N_k]$, *if* $X_\mathbf{p}$ *is masked, then the one-layer transformer nearly "pays all attention" to all unmasked patches in the same area* $\mathcal{P}_{k,j}$ *as* $\mathbf{p}$, *i.e.,*

$$\Big(1 - \mathbf{Attn}^m_{\mathbf{p} \to \mathcal{P}_{k,j}}\big(X; Q^{(T^\star)}\big)\Big)^2 \leq O(\epsilon).$$

Theorem 4.1 indicates that, at the time of convergence, for any masked query patch $X_\mathbf{p}$ in the $k$-th cluster, the transformer exhibits an *area-wide* pattern of attention, concentrating on those unmasked patches within the area that $\mathbf{p}$ lies in, as demonstrated in Section 3. The location of the patch determines such area-wide attention and can be achieved no matter if $\mathbf{p}$ belongs to the global or local areas, which jointly highlight the **diverse local patterns** for masked vision pretraining no matter degree of the imbalance.

Next, we detail the training phases of attention correlations in the following theorem, which explicitly confirms that the model **learns** target FP correlations while **ignoring** PP correlations to achieve the desirable area-wide attention patterns as suggested in Section 3 (illustrated in Figure 4).

**Theorem 4.2** (Learning Feature-Position correlations). *Following the same assumptions in Theorem 4.1, for* $\mathbf{p} \in \mathcal{P}$, *given* $k \in [K]$, *if* $\mathbf{p} \in \mathcal{P}_{k,j}$ *for some* $j \in [N_k]$, *we have*

*For positive information gap* $\Delta \in [\Omega(1), 1]$:

a. *Global areas* ($j = 1$) *learn FP correlation in* **one-phase**: $\Phi^{(t)}_{\mathbf{p} \to v_{k,1}}$ *monotonically increases to* $O(\log(P/\epsilon))$ *throughout the training, with all other attention correlations remain close to* 0.

b. *Local areas* ($j > 1$) *learn FP correlation in* **two-phase**: *In phase one, FP correlation* $\Phi^{(t)}_{\mathbf{p} \to v_{k,1}}$ *between local area and the global area feature quickly decreases to* $-\Theta(\log(P))$ *whereas all other attention correlation stay close to zero; In phase two, FP correlation* $\Phi^{(t)}_{\mathbf{p} \to v_{k,j}}$ *for the target local area starts to grow until convergence with all other attention correlations nearly unchanged.*

*For negative information gap* $\Delta \in [-0.5, -\Omega(1)]$:

c. *All areas learn FP correlation through* **one-phase**: $\Phi^{(t)}_{\mathbf{p} \to v_{k,j}}$ *monotonically increases to* $O(\log(P/\epsilon))$ *throughout the training, with all other attention correlations remain close to* 0.

The training dynamics are different depending on whether $\Delta$ is positive or negative, and further vary for positive $\Delta$ depending on whether $X_\mathbf{p}$ is situated in global or local areas. Typically, the target FP correlations are learned directly in a single phase. However, for a positive information gap $\Delta$, when patch $\mathbf{p}$ is located in a local area, the learning process contains an additional decoupling phase, to reduce the FP correlation with the non-target global features.

## 4.2 CONTRASTIVE LEARNING COLLAPSES TO GLOBAL ATTENTION PATTERNS

In contrast to MAE's ability to learn diverse local features regardless of the information gap, our results in this section demonstrate that CL inevitably collapses to global attention patterns by solely learning global FP correlations, even under a slight structural imbalance.

To prevent trivial solutions in CL, we adopt a noisy variant of the data distribution.

**Assumption 4.3** (Noisy data). We assume that the data used for contrastive learning is sampled from $\mathcal{D}^{\text{cl}}$. Specifically, to generate a sample $X \sim \mathcal{D}^{\text{cl}}$, we first draw $Z \sim \mathcal{D}$, then add independent and identically distributed (i.i.d.) noise $\zeta_{\mathbf{p}} \sim \mathcal{N}(0, \sigma_0^2 I_d)$ to each patch $Z_{\mathbf{p}}$. The resulting sample is defined as $X_{\mathbf{p}} = Z_{\mathbf{p}} + \zeta_{\mathbf{p}}$. We denote $X \in \mathcal{D}_k^{\text{cl}}$ if $Z \in \mathcal{D}_k$.

**Theorem 4.4** (Learning with contrastive objective). *Suppose the information gap* $\Delta \in [-0.5, -\Omega(1)] \cup [\Omega(1), 1]$. *We train the ViTs in Def. 2.6 by GD to minimize eq.* (2.11) *with* $\eta \ll \text{poly}(P)$, $\sigma_0^2 = \frac{1}{d}$, $\tau = O(\frac{1}{\log d})$. *Then after* $T^{\star} = O(\frac{\text{poly}(P) \log P}{\eta})$ *iterations, we have*

1. *Loss converges:* $\mathcal{L}_{cl}(Q^{(T^{\star})}) \leq \mathcal{L}_{cl}^{\star} + \frac{1}{\text{poly}(P)}$, *where* $\mathcal{L}_{cl}^{\star}$ *is the global minimum of the contrastive loss in eq.* (2.11).

2. *Attention concentration on* **global** *area : given* $X \in \mathcal{D}_k^{\text{cl}}$ *with* $k \in [K]$, *for any* $\mathbf{p} \in \mathcal{P}$, *with high probability, we have* $1 - \mathbf{Attn}_{\mathbf{p} \to \mathcal{P}_{k,1}}^{c}(X'; Q^{(T^{\star})}) = \frac{1}{\text{poly}(P)}$ *for* $X' \in \{X^+, X^{++}\}$.[4]

3. *All patches learn global FP correlation: given* $k \in [K]$, *for any* $\mathbf{p} \in \mathcal{P}$, $t \in [0, T^{\star}]$, $\Phi_{\mathbf{p} \to v_{k,1}}^{(t)} \gg \Phi_{\mathbf{p} \to v_{k,m}}^{(t)}$ *with* $m > 1$, *and at the convergence,* $\Phi_{\mathbf{p} \to v_{k,1}}^{(T^{\star})} = \Theta(\log P)$, $\Phi_{\mathbf{p} \to v_{k,m}}^{(T^{\star})} = o(1)$.

**Intuition behind learning global correlations.** As discussed in Section 3, the optimal alignment of two positive representations $F^{\text{cl}}(Q; X^+)$ and $F^{\text{cl}}(Q; X^{++})$ involves directing attention towards the same feature for each patch $\mathbf{p}$, possibly irrespective of its location. As long as the imbalanced structure, where global features dominate the data distribution, exists—even to a small degree—it leads to an order-wise stronger concentration of attention on global areas at initialization. Consequently, global FP correlations receive larger gradients compared to local ones. Therefore, global FP correlations are learned first, and focusing on these global correlations is sufficient for the CL objective to converge.

**Significance of the results.** Theorem 4.1 and Theorem 4.4 address a critical gap in understanding self-supervised pretraining by offering the first theoretical framework for learning with ViTs, one of the most advanced architectures in vision practice, whereas prior studies have primarily focused on linear models, CNNs, or MLPs (Wen & Li, 2021; Ji et al., 2023; Pan et al., 2022). Moreover, by identifying the collapsed solution in CL and emphasizing the effectiveness of MAE in capturing diverse attention patterns, we provide a qualitative comparison between MAE and contrastive learning, validating a non-trivial empirical observation (Park et al., 2023). This offers a comprehensive theoretical analysis of self-supervised learning with ViTs.

## 5 CONCLUSION

In this work, we study the training process of MAE and CL with one-layer softmax-based ViTs. Our key contribution is providing the first end-to-end convergence guarantees for these two prominent self-supervised approaches with transformer architectures. We characterize the attention patterns at convergence and show that MAE exhibits diverse attention patterns by learning feature-position correlations across all features, even with highly skewed feature distributions. In contrast, CL collapses to global attention patterns by focusing solely on global feature-position correlations, despite minimal distributional deviations between features. This provides theoretical justification for the behavior gap of MAE and CL observed in practice. Our proof techniques use phase decomposition based on the interplay between feature-position and position-wise correlations, avoiding the need to disentangle patches and positional encodings as in prior work. We anticipate that our theory will be valuable for future studies of spatial or temporal structures in state-of-the-art transformers and will advance theoretical research in deep learning.

---

[4]This also holds when no data augmentation is applied to $X$.

ACKNOWLEDGEMENTS

The work of Y. Huang is partially supported in part by the ONR grant N00014-22-1-2354, and the NSF grants CCF-2418156 and DMS-2143215. The work of Z. Wen and Y. Chi is supported in part by NSF under CCF-1901199, CCF-2007911, DMS-2134080, DMS-2134133, and by ONR under N00014-19-1-2404. The work of Y. Liang is supported in part by NSF under RINGS-2148253, CCF-1900145, and DMS-2134145.

REPRODUCIBILITY STATEMENT:

The main body of the paper presents only theoretical results, with all proofs provided in the appendices. Additionally, the appendices include proof sketches that offer intuitive explanations of the proof steps. The appendix also contains experimental results, with detailed descriptions of the experimental settings to facilitate result reproduction.

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

# APPENDIX

## A  RELATED WORK

**Empirical studies of transformers in vision.**    A number of works have aimed to understand the transformers in vision from different perspectives: comparison with CNNs (Raghu et al., 2021; Ghiasi et al., 2022; Park & Kim, 2022), robustness (Bhojanapalli et al., 2021; Paul & Chen, 2022), and role of positional embeddings (Melas-Kyriazi, 2021; Trockman & Kolter, 2022). Recent studies (Xie et al., 2023; Wei et al., 2022b; Park et al., 2023) have delved into ViTs with self-supervision to uncover the mechanisms at play, particularly through visualization and analysis of metrics related to self-attention. Xie et al. (2023) compared the masked image modeling (MIM) method with supervised models, revealing MIM's capacity to enhance diversity and locality across all ViT layers, w which significantly boosts performance on tasks with weak semantics following fine-tuning. Building on MIM's advantages, Wei et al. (2022b) further proposed a simple feature distillation method that incorporates locality into various self-supervised methods, leading to an overall improvement in the finetuning performance. Park et al. (2023) conducted a detailed comparison between masked image modeling (MIM) and contrastive learning. They demonstrated that contrastive learning will make the self-attentions collapse into homogeneity for all query patches due to the nature of discriminative learning, while MIM leads to a diverse self-attention map since it focuses on local patterns.

**Theory of self-supervised learning.**    A major line of theoretical studies falls into one of the most successful self-supervised learning approaches, contrastive learning (Wen & Li, 2021; Robinson et al., 2021; Chen et al., 2021a; Arora et al., 2019), and its variant non-contrastive self-supervised learning (Wen & Li, 2022; Pokle et al., 2022; Wang et al., 2021). Some other works study the mask prediction approach (Lee et al., 2021; Wei et al., 2021; Liu et al., 2022), which is the focus of this paper. Lee et al. (2021) provided statistical downstream guarantees for reconstructing missing patches. Wei et al. (2021) studied the benefits of head and prompt tuning with masked pretraining under a Hidden Markov Model framework. Liu et al. (2022) provided a parameter identifiability view to understand the benefit of masked prediction tasks, which linked the masked reconstruction tasks to the informativeness of the representation via identifiability techniques from tensor decomposition.

**Theory of transformers and attention models.**    Prior work has studied the theoretical properties of transformers from various aspects: representational power (Yun et al., 2019; Edelman et al., 2022; Vuckovic et al., 2020; Wei et al., 2022a; Sanford et al., 2024a), internal mechanism (Tarzanagh et al., 2023a; Weiss et al., 2021), limitations (Hahn, 2020; Sanford et al., 2024b), and PAC learning (Chen & Li, 2024). Recently, there has been a growing body of research studying in-context learning with transformers due to the remarkable emergent in-context ability of large language models (Zhang et al., 2023b; Von Oswald et al., 2023; Giannou et al., 2023; Ahn et al., 2023; Zhang et al., 2023a; Huang et al., 2023; Nichani et al., 2024; Li et al., 2024). Regarding the training dynamics of attention-based models, Li et al. (2023a) studied the training process of shallow ViTs in a classification task. Subsequent research expanded on this by exploring the graph transformer with positional encoding (Li et al., 2023b) and in-context learning performance of transformers with nonlinear self-attention and nonlinear MLP (Li et al., 2024). However, all of these analyses rely crucially on stringent assumptions on the initialization of transformers and hardly generalize to our setting. Tian et al. (2023) mathematically described how the attention map evolves trained by SGD for one-layer transformer but did not provide any convergence guarantee, and the follow-up work Tian et al. (2024) considered a generalized case with multiple layers. Tarzanagh et al. (2023b); Vasudeva et al. (2024) investigated the implicit bias for self-attention models trained with GD. Furthermore, Huang et al. (2023) proved the in-context convergence of a one-layer softmax transformer trained via GD and illustrated the attention dynamics throughout the training process. Yang et al. (2024) generalized such an in-context learning problem to a mult-head setting with non-linear task functions. Nichani et al. (2024) studied GD dynamics on a simplified two-layer attention-only transformer and proved that it can encode the causal structure in the first attention layer. However, none of the previous studies analyzed the training of transformers under self-supervised learning, which is the focus of this paper.

## B  EXPERIMENTS

Previous studies on the attention mechanisms of ViT-based pre-training approaches have mainly utilized a metric known as the attention distance (Dosovitskiy et al., 2020). Such a metric quantifies the average spatial distance between the query and key tokens, weighted by their self-attention coefficients.

The general interpretation is that larger attention distances indicate global understanding, and smaller values suggest a focus on local features. However, such a metric does not adequately determine if the self-attention mechanism is identifying a unique global pattern. A high attention distance could result from different patches focusing on varied distant areas, which does not necessarily imply that global information is being effectively synthesized. To address this limitation, we introduce a novel and revised version of average attention distance, called the attention diversity metric, which is designed to assess whether various patches are concentrating on a similar region, thereby directly capturing global information.

**Attention diversity metric, in distance.** This metric is computed for self-attention with a single head of the specific layer. For a given image divided into $N \times N$ patches, the process unfolds as follows: for each patch, it is employed as the query patch to calculate the attention weights towards all $N^2$ patches, and those with the top-$n$ attention weights are selected. Subsequently, the coordinates (e.g. $(i, j)$ with $i, j \in [N]$) of these top-$n$ patches are concatenated in sequence to form a $2 \times n$-dimensional vector. The final step computes the average distance between all these $2n$-dimensional vectors, i.e., $N^2 \times N^2$ vector pairs.

**Setup.** In this work, we compare the performance of ViT-B/16 encoder pre-trained on ImageNet-1K (Russakovsky et al., 2015) among the following four models: masked reconstruction model (MAE), contrastive learning model (MoCo v3 (Chen et al., 2021b)), other self-supervised model (DINO Caron et al. (2021)), and supervised model (DeiT Touvron et al. (2021)). We focus on 12 different attention heads in the last layer of ViT-B on different pre-trained models. The box plot visualizes the distribution of the top-10 averaged attention focus across 152 example images, as similarly done in Dosovitskiy et al. (2020).

**Implications.** The experiment results based on our new metric are provided in Figure 3. Lower values of the attention diversity metric signify a focused attention on a coherent area across different patches, reflecting a global pattern of focus. On the other hand, higher values suggest that attention is dispersed, focusing on different, localized areas. It can be seen that the masked pretraining model is particularly effective in learning more diverse attention patterns, setting it apart from other models that prioritize a uniform global information with less attention diversity. This aligns with and provides further evidence for the findings in Park et al. (2023).

## C  OVERVIEW OF THE PROOF TECHNIQUES

In this section, we explain our key proof techniques in analyzing the self-supervised pretraining of transformers, using MAE as an example. We focus on the reconstruction of a specific patch $X_{\mathbf{p}}$ for $\mathbf{p} \in \mathcal{P}$. We aim to elucidate the training phases through which the model learns FP correlations related to the area associated with $\mathbf{p}$ across different clusters $k \in [K]$.

Our characterization of training phases differentiates between whether $X_{\mathbf{p}}$ is located in the global or local areas and further varies based on whether $\Delta$ is positive or negative. Specifically, for $\Delta \in [\Omega(1), 1]$, we observe distinct learning dynamics for FP correlations between local and global areas:

- Local area attends to FP correlation in two-phase: given $k \in [K]$, if $a_{k,\mathbf{p}} \neq 1$, then
    1. $\Phi_{\mathbf{p} \to v_{k,1}}^{(t)}$ first quickly decreases whereas all other $\Phi_{\mathbf{p} \to v_{k,m}}^{(t)}$ with $m \neq 1$ and $\Upsilon_{\mathbf{p} \to \mathbf{q}}^{(t)}$ do not change much;
    2. after some point, the increase of $\Phi_{\mathbf{p} \to v_{k,a_{k,\mathbf{p}}}}^{(t)}$ takes dominance. Such $\Phi_{\mathbf{p} \to v_{k,a_{k,\mathbf{p}}}}^{(t)}$ will keep growing until convergence with all other FP and PP attention correlations nearly unchanged.
- Global areas learn FP correlation in one-phase: given $k \in [K]$, if $a_{k,\mathbf{p}} = 1$, the update of $\Phi_{\mathbf{p} \to v_{k,1}}^{(t)}$ will dominate throughout the training, whereas all other $\Phi_{\mathbf{p} \to v_{k,m}}^{(t)}$ with $m \neq 1$ and learned PP correlations remain close to 0.

For $\Delta \in [-0.5, -\Omega(1)]$, the behaviors of learning FP correlations are uniform for all areas. Namely, all areas learn FP correlation through one-phase: given $k \in [K]$, throughout the training, the increase

of $\Phi_{\mathbf{p}\to v_{k,a_{k,\mathbf{p}}}}^{(t)}$ dominates, whereas all other $\Phi_{\mathbf{p}\to v_{k,m}}^{(t)}$ with $m \neq a_{k,\mathbf{p}}$ and PP correlations $\Upsilon_{\mathbf{p}\to\mathbf{q}}^{(t)}$ remain close to 0.

For clarity, this section will mainly focus on the learning of *local* feature correlations with a positive information gap $\Delta \geq \Omega(1)$ in Appendices C.2 and C.3, which exhibits a two-phase process. The other scenarios will be discussed briefly in Appendix C.4.

## C.1 GD DYNAMICS OF ATTENTION CORRELATIONS

Based on the crucial roles that attention correlations play in determining the reconstruction loss, the main idea of our analysis is to track the dynamics of those attention correlations. We first provide the following GD updates of $\Phi_{\mathbf{p}\to v_{k,m}}^{(t)}$ and $\Upsilon_{\mathbf{p}\to\mathbf{q}}^{(t)}$ (see Appendix D.1.1 for formal statements).

**Lemma C.1** (FP correlations, informal). *Given $k \in [K]$, for $\mathbf{p} \in \mathcal{P}$, denote $n = a_{k,\mathbf{p}}$, let $\alpha_{\mathbf{p}\to v_{k,m}}^{(t)} = \frac{1}{\eta}\big(\Phi_{\mathbf{p}\to v_{k,m}}^{(t+1)} - \Phi_{\mathbf{p}\to v_{k,m}}^{(t)}\big)$ for $m \in [N_k]$, and suppose $X_{\mathbf{p}}$ is masked. Then*

*1. for the same area, $\alpha_{\mathbf{p}\to v_{k,n}}^{(t)} \approx \mathbf{Attn}_{\mathbf{p}\to\mathcal{P}_{k,n}}^{(t)}\left(1 - \mathbf{Attn}_{\mathbf{p}\to\mathcal{P}_{k,n}}^{(t)}\right)^2$;*

*2. if $k \in \mathcal{B}_{\mathbf{p}}$, for the global area,*

$$\alpha_{\mathbf{p}\to v_{k,1}}^{(t)} \approx -\mathbf{Attn}_{\mathbf{p}\to\mathcal{P}_{k,1}}^{(t)} \cdot \left(\mathbf{Attn}_{\mathbf{p}\to\mathcal{P}_{k,1}}^{(t)}\left(1 - \mathbf{Attn}_{\mathbf{p}\to\mathcal{P}_{k,1}}^{(t)}\right) + \mathbf{Attn}_{\mathbf{p}\to\mathcal{P}_{k,n}}^{(t)}\left(1 - \mathbf{Attn}_{\mathbf{p}\to\mathcal{P}_{k,n}}^{(t)}\right)\right);$$

*3. for other area $m \notin \{n\} \cup \{1\}$,*

$$\alpha_{\mathbf{p}\to v_{k,m}}^{(t)} \approx \mathbf{Attn}_{\mathbf{p}\to\mathcal{P}_{k,m}}^{(t)}\left(\mathbb{1}\{n \neq 1\}\left(\mathbf{Attn}_{\mathbf{p}\to\mathcal{P}_{k,1}}^{(t)}\right)^2 - \left(1 - \mathbf{Attn}_{\mathbf{p}\to\mathcal{P}_{k,n}}^{(t)}\right)\mathbf{Attn}_{\mathbf{p}\to\mathcal{P}_{k,n}}^{(t)}\right).$$

From Lemma C.1, it is observed that for $\mathbf{p} \in \mathcal{P}_{k,n}$, the feature correlation $\Phi_{\mathbf{p}\to v_{k,n}}^{(t)}$ exhibits a monotonically increasing trend over time because $\alpha_{\mathbf{p}\to v_{k,n}}^{(t)} \geq 0$. Furthermore, if $n > 1$, i.e., $\mathcal{P}_{k,n}$ is the local area, $\Phi_{\mathbf{p}\to v_{k,1}}^{(t)}$ will monotonically decrease.

**Lemma C.2** (PP attention correlations, informal). *Given $\mathbf{p}, \mathbf{q} \in \mathcal{P}$, let $\beta_{\mathbf{p}\to\mathbf{q}}^{(t)} = \frac{1}{\eta}\big(\Upsilon_{\mathbf{p}\to\mathbf{q}}^{(t+1)} - \Upsilon_{\mathbf{p}\to\mathbf{q}}^{(t)}\big)$, and suppose $X_{\mathbf{p}}$ is masked. Then $\beta_{\mathbf{p}\to\mathbf{q}}^{(t)} = \sum_{k\in[N]} \beta_{k,\mathbf{p}\to\mathbf{q}}^{(t)}$, where $\beta_{k,\mathbf{p}\to\mathbf{q}}^{(t)}$ satisfies*

*1. if $a_{k,\mathbf{p}} = a_{k,\mathbf{q}} = n$, $\beta_{k,\mathbf{p}\to\mathbf{q}}^{(t)} \approx \mathbf{attn}_{\mathbf{p}\to\mathbf{q}}^{(t)}\left(1 - \mathbf{Attn}_{\mathbf{p}\to\mathcal{P}_{k,n}}^{(t)}\right)^2$;*

*2. if $k \in \mathcal{B}_{\mathbf{p}} \cap \mathcal{C}_{\mathbf{q}}$, where $a_{k,\mathbf{p}} = n > 1$ and $a_{k,\mathbf{q}} = 1$:*

$$\beta_{k,\mathbf{p}\to\mathbf{q}}^{(t)} \approx -\mathbf{attn}_{\mathbf{p}\to\mathbf{q}}^{(t)} \cdot \left(\mathbf{Attn}_{\mathbf{p}\to\mathcal{P}_{k,1}}^{(t)}\left(1 - \mathbf{Attn}_{\mathbf{p}\to\mathcal{P}_{k,1}}^{(t)}\right) + \mathbf{Attn}_{\mathbf{p}\to\mathcal{P}_{k,n}}^{(t)}\left(1 - \mathbf{Attn}_{\mathbf{p}\to\mathcal{P}_{k,n}}^{(t)}\right)\right);$$

*3. if $a_{k,\mathbf{q}} = m \notin \{n\} \cup \{1\}$, where $a_{k,\mathbf{p}} = n$,*

$$\beta_{k,\mathbf{p}\to\mathbf{q}}^{(t)} \approx \mathbf{attn}_{\mathbf{p}\to\mathbf{q}}^{(t)} \cdot \left(\mathbb{1}\{n \neq 1\}\left(\mathbf{Attn}_{\mathbf{p}\to\mathcal{P}_{k,1}}^{(t)}\right)^2 - \left(1 - \mathbf{Attn}_{\mathbf{p}\to\mathcal{P}_{k,n}}^{(t)}\right)\mathbf{Attn}_{\mathbf{p}\to\mathcal{P}_{k,n}}^{(t)}\right).$$

Based on the above gradient update for $\Upsilon_{\mathbf{p}\to\mathbf{q}}^{(t)}$, we further introduce the following auxiliary quantity $\Upsilon_{k,\mathbf{p}\to\mathbf{q}}^{(t)}$, which can be interpreted as the PP attention correlation "projected" on the $k$-th cluster $\mathcal{D}_k$, and will be useful in the later proof.

$$\Upsilon_{k,\mathbf{p}\to\mathbf{q}}^{(t+1)} := \Upsilon_{k,\mathbf{p}\to\mathbf{q}}^{(t)} + \eta\beta_{k,\mathbf{p}\to\mathbf{q}}^{(t)}, \quad \text{with } \Upsilon_{k,\mathbf{p}\to\mathbf{q}}^{(0)} = 0. \tag{C.1}$$

We can directly verify that $\Upsilon_{\mathbf{p}\to\mathbf{q}}^{(t)} = \sum_{k\in[K]} \Upsilon_{k,\mathbf{p}\to\mathbf{q}}^{(t)}$.

The key observation by comparing Lemma C.1 and C.2 is that the gradient of projected PP attention $\beta_{k,\mathbf{p}\to\mathbf{q}}^{(t)}$ is smaller than the corresponding FP gradient $\alpha_{\mathbf{p}\to v_{k,a_{k,\mathbf{q}}}}^{(t)}$ in magnitude since $\mathbf{attn}_{\mathbf{p}\to\mathbf{q}}^{(t)} \approx \frac{\mathbf{Attn}_{\mathbf{p}\to\mathcal{P}_{k,a_{k,\mathbf{q}}}}^{(t)}}{(1-\gamma)C_{k,a_{k,\mathbf{q}}}}$. We will show that the interplay between the increase of $\Phi_{\mathbf{p}\to v_{k,n}}^{(t)}$ and the decrease of $\Phi_{\mathbf{p}\to v_{k,1}}^{(t)}$ determines the learning behaviors for the local patch $\mathbf{p} \in \mathcal{P}_{k,n}$ with $n > 1$, and which effect will happen first depends on the initial attention, which is also determined by the value of information gap $\Delta$.

## C.2 PHASE I: DECOUPLING THE GLOBAL FP CORRELATIONS

We now explain how the attention correlations evolve at the initial phase of the training to decouple the correlations of the non-target global features when $\mathbf{p}$ is located in the local area for the $k$-th cluster. This phase can be further divided into the following two stages.

**Stage 1.** At the beginning of training, $\Phi_{\mathbf{p}\to v_{k,m}}^{(0)} = \Upsilon_{k,\mathbf{p}\to\mathbf{q}}^{(0)} = 0$, and hence $\mathbf{attn}_{\mathbf{p}\to\mathbf{q}}^{(0)} = \frac{1}{P}$ for any $\mathbf{q} \in \mathcal{P}$, which implies that the transformer equally attends to each patch. However, with high probability, the number of unmasked global features in the global area $\mathcal{P}_{k,1}$ is much larger than others. Hence, $\mathbf{Attn}_{\mathbf{p}\to\mathcal{P}_{k,1}}^{(0)} = \frac{|\mathcal{U}\cap\mathcal{P}_{k,1}|}{P} \geq \Omega(\frac{1}{P^{1-\kappa_c}}) \gg \Theta(\frac{1}{P^{1-\kappa_s}}) = \mathbf{Attn}_{\mathbf{p}\to\mathcal{P}_{k,m}}^{(0)}$ for $m > 1$. Therefore, by Lemma C.1 and C.2, we immediately obtain

- $\alpha_{\mathbf{p}\to v_{k,1}}^{(0)} = -\Theta\left(\frac{1}{P^{2(1-\kappa_c)}}\right)$, whereas $\alpha_{\mathbf{p}\to v_{k,a_{k,\mathbf{p}}}}^{(0)} = \Theta\left(\frac{1}{P^{(1-\kappa_s)}}\right)$;
- all other FP correlation gradients $\alpha_{\mathbf{p}\to v_{k,m}}^{(0)}$ with $m \neq 1, a_{k,\mathbf{p}}$ are small;
- all projected PP correlation gradients $\beta_{k,\mathbf{p}\to\mathbf{q}}^{(0)}$ are small.

Since $\Delta = (1-\kappa_s) - 2(1-\kappa_c) \geq \Omega(1)$, it can be seen that $\Phi_{\mathbf{p}\to v_{k,1}}^{(t)}$ enjoys a much larger decreasing rate initially. This captures the decoupling process of the feature correlations with the global feature $v_{k,1}$ in the global area for $\mathbf{p}$. It can be shown that such an effect will dominate over a certain period that defines stage 1 of phase I. At the end of this stage, we will have $\Phi_{\mathbf{p}\to v_{k,1}}^{(t)} \leq -\Omega\left(\log(P)\right)$, whereas all FP attention correlation $\Phi_{\mathbf{p}\to v_{k,m}}^{(t)}$ with $m > 1$ and all projected PP correlations $\Upsilon_{k,\mathbf{p}\to\mathbf{q}}^{(t)}$ stay close to 0 (see Appendix F.1).

During stage 1, the significant decrease of the global FP correlation $\Phi_{\mathbf{p}\to v_{k,1}}^{(t)}$ leads to a reduction in the attention score $\mathbf{Attn}_{\mathbf{p}\to\mathcal{P}_{k,1}}^{(t)}$. Meanwhile, attention scores $\mathbf{Attn}_{\mathbf{p}\to\mathcal{P}_{k,m}}^{(t)}$ (where $m > 1$) for other patches remain consistent, reflecting a uniform distribution over unmasked patches within each area. By the end of stage 1, $\mathbf{Attn}_{\mathbf{p}\to\mathcal{P}_{k,1}}^{(t)}$ drops to a certain level, resulting in a decrease in $|\alpha_{\mathbf{p}\to v_{k,1}}^{(t)}|$ as it approaches $\alpha_{\mathbf{p}\to v_{k,n}}^{(t)}$, which indicates that stage 2 begins.

**Stage 2.** Soon as stage 2 begins, the dominant effect switches as $|\alpha_{\mathbf{p}\to v_{k,1}}^{(t)}|$ reaches the same order of magnitude as $\alpha_{\mathbf{p}\to v_{k,a_{k,\mathbf{p}}}}^{(t)}$. The following result shows that $\Phi_{\mathbf{p}\to v_{k,a_{k,\mathbf{p}}}}^{(t)}$ must update during stage 2.

**Lemma C.3** (Switching of dominant effects (See Appendix F.2)). *Under the same conditions as Theorem 4.1, for $\mathbf{p} \in \mathcal{P}$, there exists $\widetilde{T}_1$, such that at iteration $t = \widetilde{T}_1 + 1$, we have*

a. $\Phi_{\mathbf{p}\to v_{k,a_{k,\mathbf{p}}}}^{(\widetilde{T}_1+1)} \geq \Omega\left(\log(P)\right)$, and $\Phi_{\mathbf{p}\to v_{k,1}}^{(\widetilde{T}_1+1)} = -\Theta(\log(P))$;

b. *all other FP correlations $\Phi_{\mathbf{p}\to v_{k,m}}^{(t)}$ with $m \neq 1, a_{k,\mathbf{p}}$ are small;*

c. *all projected PP correlations $\Upsilon_{k,\mathbf{p}\to\mathbf{q}}^{(t)}$ are small.*

**Intuition of the transition.** Once $\Phi_{\mathbf{p}\to v_{k,1}}^{(t)}$ decreases to $-\frac{\Delta}{2L}\log(P)$, we observe that $|\alpha_{\mathbf{p}\to v_{k,1}}^{(t)}|$ is approximately equal to $\alpha_{\mathbf{p}\to v_{k,a_{k,\mathbf{p}}}}^{(t)}$. After this point, reducing $\Phi_{\mathbf{p}\to v_{k,1}}^{(t)}$ further is more challenging compared to the increase in $\Phi_{\mathbf{p}\to v_{k,a_{k,\mathbf{p}}}}^{(t)}$. To illustrate, a minimal decrease of $\Phi_{\mathbf{p}\to v_{k,1}}^{(t)}$ by an amount

of $\frac{0.001}{L} \log(P)$ will yield $|\alpha^{(t)}_{\mathbf{p} \to v_{k,1}}| \leq O(\frac{\alpha^{(t)}_{\mathbf{p} \to v_{k,n}}}{P^{0.002}})$. Such a discrepancy triggers the switch of the dominant effect.

## C.3 PHASE II: GROWTH OF TARGET LOCAL FP CORRELATION

Moving beyond phase I, FP correlation $\Phi^{(t)}_{\mathbf{p} \to v_{k,a_{k,\mathbf{p}}}}$ within the target local area $\mathbf{p}$ already enjoys a larger gradient $\alpha^{(t)}_{\mathbf{p} \to v_{k,a_{k,\mathbf{p}}}}$ than other $\Phi^{(t)}_{\mathbf{p} \to v_{k,m}}$ with $m \neq a_{k,\mathbf{p}}$ and all projected PP correlations $\Upsilon^{(t)}_{k,\mathbf{p} \to \mathbf{q}}$. We can show that the growth of $\Phi^{(t)}_{\mathbf{p} \to v_{k,a_{k,\mathbf{p}}}}$ will continue to dominate until the end of training by recognizing the following two stages.

**Rapid growth stage.** At the beginning of phase II, $\alpha^{(t)}_{\mathbf{p} \to v_{k,a_{k,\mathbf{p}}}}$ is mainly driven by $\mathbf{Attn}^{(t)}_{\mathbf{p} \to \mathcal{P}_{k,a_{k,\mathbf{p}}}}$ since $1 - \mathbf{Attn}^{(t)}_{\mathbf{p} \to \mathcal{P}_{k,a_{k,\mathbf{p}}}}$ remains at the constant order. Therefore, the growth of $\Phi^{(t)}_{\mathbf{p} \to v_{k,a_{k,\mathbf{p}}}}$ naturally results in a boost in $\mathbf{Attn}^{(t)}_{\mathbf{p} \to \mathcal{P}_{k,a_{k,\mathbf{p}}}}$, thereby promoting an increase in its own gradient $\alpha^{(t)}_{\mathbf{p} \to v_{k,a_{k,\mathbf{p}}}}$, which defines the rapid growth stage. On the other hand, we can prove that the following gap holds for FP and projected PP correlation gradients (see Appendix F.3):

- all other FP correlation gradients $\alpha^{(t)}_{\mathbf{p} \to v_{k,m}}$ with $m \neq a_{k,\mathbf{p}}$ are small;
- all projected PP correlation gradients $\beta^{(t)}_{k,\mathbf{p} \to \mathbf{q}}$ are small.

**Convergence stage.** After the rapid growth stage, the desired local pattern with a high target feature-position correlation $\Phi^{(t)}_{\mathbf{p} \to v_{k,a_{k,\mathbf{p}}}}$ is learned. In this last stage, it is demonstrated that the above conditions for non-target FP and projected PP correlations remain valid, while the growth of $\Phi^{(t)}_{\mathbf{p} \to v_{k,a_{k,\mathbf{p}}}}$ starts to decelerate as $\Phi^{(t)}_{\mathbf{p} \to v_{k,a_{k,\mathbf{p}}}}$ reaches $\Theta(\log(P))$, resulting in $\mathbf{Attn}^{(t)}_{\mathbf{p} \to \mathcal{P}_{k,n}} \approx \Omega(1)$, which leads to convergence (see Appendix F.4).

## C.4 LEARNING PROCESSES IN OTHER SCENARIOS

In this section, we talk about the learning process in other settings, including learning FP correlations for the local area when the information gap is negative, learning FP correlations for the global area, and failure to learn PP correlations.

**What is the role of positive information gap?** As described in stage 1 of phase 1 in Appendix C.2, the decoupling effect happens at the beginning of the training because $\alpha^{(0)}_{\mathbf{p} \to v_{k,1}} \gg \alpha^{(0)}_{\mathbf{p} \to v_{k,a_{k,\mathbf{p}}}}$ attributed to $\Delta \geq \Omega(1)$. However, in cases where $\Delta \leq -\Omega(1)$, this relationship reverses, with $\alpha^{(0)}_{\mathbf{p} \to v_{k,1}}$ becoming significantly smaller than $\alpha^{(0)}_{\mathbf{p} \to v_{k,a_{k,\mathbf{p}}}}$. Similarly, other FP gradients $\alpha^{(0)}_{\mathbf{p} \to v_{k,m}}$ with $m \neq 1, a_{k,\mathbf{p}}$ and all the projected gradients of PP correlation $\beta^{(0)}_{\mathbf{p} \to \mathbf{q}}$ are small in magnitude. Consequently, $\Phi^{(t)}_{\mathbf{p} \to v_{k,a_{k,\mathbf{p}}}}$ starts with a larger gradient, eliminating the need to decouple FP correlations for the global area. As a result, training skips the initial phase, and moves directly into Phase II, during which $\Phi^{(t)}_{\mathbf{p} \to v_{k,a_{k,\mathbf{p}}}}$ continues to increase until it converges (see Appendix G).

**Learning FP correlations for the global area.** When the patch $X_{\mathbf{p}}$ is located in the global area of cluster $k$, i.e., $a_{k,\mathbf{p}} = 1$, the attention score $\mathbf{Attn}^{(0)}_{\mathbf{p} \to \mathcal{P}_{k,1}}$ directed towards the target area $\mathcal{P}_{k,1}$ is initially higher compared to other attention scores due to the presence of a significant number of unmasked patches in the global area. This leads to an initially larger gradient $\alpha^{(0)}_{\mathbf{p} \to v_{k,a_{k,\mathbf{p}}}}$. Such an effect is independent of the value of $\Delta$. As a result, the training process skips the initial phase, which is typically necessary for the cases where $a_{k,\mathbf{p}} > 1$ with a positive information gap, and moves directly into Phase II (see Appendix H).

**All PP correlations are small.** Integrating the analysis from all previous discussions, we establish that for every cluster $k \in [K]$, regardless of its association with $\mathcal{C}_{\mathbf{p}}$ (global area) or $\mathcal{B}_{\mathbf{p}}$ (local area),

and for any patch $X_{\mathbf{q}}$ with $\mathbf{q} \in \mathcal{P}$, the projected PP correlation $\Upsilon^{(t)}_{k,\mathbf{p} \to \mathbf{q}}$ remains nearly zero in comparison to the significant changes observed in the FP correlation, because the gradient $\beta^{(t)}_{k,\mathbf{p} \to \mathbf{q}}$ is relatively negligible. Therefore, the overall PP correlation $\Upsilon^{(t)}_{\mathbf{p} \to \mathbf{q}} = \sum_{k=1}^{K} \Upsilon^{(t)}_{k,\mathbf{p} \to \mathbf{q}}$ also stays close to zero, given that the number of clusters $K = \Theta(1)$.

## D  PRELIMINARIES

In this section, we will introduce warm-up gradient computations and probabilistic lemmas that establish essential properties of the data and the loss function, which are pivotal for the technical proofs in the upcoming sections for masked pretraining. Throughout the appendix, we assume $N_k = N$ and $C_{k,n} = C_n$ for all $k \in [K]$ for simplicity. We will also omit the explicit dependence on $X$ for $z_n(X)$. We use $k_X \in [K]$ to denote the cluster index that a given image $X$ is drawn from. Furthermore, we will abbreviate $\mathcal{L}_{\texttt{mae}}(\mathcal{L}_{\texttt{mae},\mathbf{p}})$ as $\mathcal{L}(\mathcal{L}_{\mathbf{p}})$, and $F^{\texttt{mae}}$ as $F$ for simplicity, when the context makes it clear. We abbreviate $\mathbf{Attn}^{\texttt{m}}_{\mathbf{p} \to \mathcal{P}_{k,m}}(X; Q^{(t)})$ $(\mathbf{attn}^{\texttt{m}}_{\mathbf{p} \to \mathbf{q}}(X; Q^{(t)}))$ as $\mathbf{Attn}^{(t)}_{\mathbf{p} \to \mathcal{P}_{k,m}}($ $\mathbf{attn}^{(t)}_{\mathbf{p} \to \mathbf{q}})$, when the context makes it clear.

### D.1  GRADIENT COMPUTATIONS

We first calculate the gradient with respect to $Q$. We omit the superscript '$(t)$' and write $\mathcal{L}(Q)$ as $\mathcal{L}$ here for simplicity.

**Lemma D.1.** *The gradient of the loss function with respect to $Q$ is given by*

$$\frac{\partial \mathcal{L}}{\partial Q} = -\mathbb{E}\left[ \sum_{\mathbf{p} \in \mathcal{M}} \sum_{\mathbf{q}} \mathbf{attn}_{\mathbf{p} \to \mathbf{q}} \mathsf{M}(X)_{\mathbf{q}}^{\top} (X_{\mathbf{p}} - [F(\mathsf{M}(X); Q)]_{\mathbf{p}}) \cdot \right.$$
$$\left. \widetilde{\mathsf{M}}(X)_{\mathbf{p}} \left( \widetilde{\mathsf{M}}(X)_{\mathbf{q}} - \sum_{\mathbf{r}} \mathbf{attn}_{\mathbf{p} \to \mathbf{r}} \widetilde{\mathsf{M}}(X)_{\mathbf{r}} \right)^{\top} \right].$$

*Proof.* We begin with the chain rule and obtain

$$\frac{\partial \mathcal{L}}{\partial Q} = \mathbb{E}\left[ \sum_{\mathbf{p} \in \mathcal{M}} \frac{\partial [F(\mathsf{M}(X); Q)]_{\mathbf{p}}}{\partial Q} ([F(\mathsf{M}(X); Q)]_{\mathbf{p}} - X_{\mathbf{p}}) \right]$$
$$= \mathbb{E}\left[ \sum_{\mathbf{p} \in \mathcal{M}} \sum_{\mathbf{q}} \frac{\partial \mathbf{attn}_{\mathbf{p} \to \mathbf{q}}}{\partial Q} \mathsf{M}(X)_{\mathbf{q}}^{\top} ([F(\mathsf{M}(X); Q)]_{\mathbf{p}} - X_{\mathbf{p}}) \right]. \quad \text{(D.1)}$$

We focus on the gradient for each attention score:

$$\frac{\partial \mathbf{attn}_{\mathbf{p} \to \mathbf{q}}}{\partial Q} = \sum_{\mathbf{r}} \frac{\exp\left( \widetilde{\mathsf{M}}(X)_{\mathbf{p}}^{\top} Q (\widetilde{\mathsf{M}}(X)_{\mathbf{r}} + \widetilde{\mathsf{M}}(X)_{\mathbf{q}}) \right)}{\left( \sum_{\mathbf{r}} \exp(\widetilde{\mathsf{M}}(X)_{\mathbf{p}}^{\top} Q \widetilde{\mathsf{M}}(X)_{\mathbf{r}}) \right)^2} \widetilde{\mathsf{M}}(X)_{\mathbf{p}} (\widetilde{\mathsf{M}}(X)_{\mathbf{q}} - \widetilde{\mathsf{M}}(X)_{\mathbf{r}})^{\top}$$
$$= \mathbf{attn}_{\mathbf{p} \to \mathbf{q}} \sum_{\mathbf{r}} \mathbf{attn}_{\mathbf{p} \to \mathbf{r}} \widetilde{\mathsf{M}}(X)_{\mathbf{p}} (\widetilde{\mathsf{M}}(X)_{\mathbf{q}} - \widetilde{\mathsf{M}}(X)_{\mathbf{r}})^{\top}$$
$$= \mathbf{attn}_{\mathbf{p} \to \mathbf{q}} \widetilde{\mathsf{M}}(X)_{\mathbf{p}} \cdot \left[ \widetilde{\mathsf{M}}(X)_{\mathbf{q}} - \sum_{\mathbf{r}} \mathbf{attn}_{\mathbf{p} \to \mathbf{r}} \widetilde{\mathsf{M}}(X)_{\mathbf{r}} \right]^{\top}.$$

Substituting the above equation into equation D.1, we complete the proof. □

Recall that the quantities $\Phi^{(t)}_{\mathbf{p} \to v_{k,m}}$ and $\Upsilon^{(t)}_{\mathbf{p} \to \mathbf{q}}$ are defined in Definition 3.1. These quantities are associated with the attention weights for each token, and they play a crucial role in our analysis of learning dynamics. We will restate their definitions here for clarity.

**Definition D.2.** (Attention correlations) Given $\mathbf{p}, \mathbf{q} \in \mathcal{P}$, for $t \geq 0$, we define two types of attention correlations as follows:

1. Feature Attention Correlation: $\Phi^{(t)}_{\mathbf{p} \to v_{k,m}} := e_{\mathbf{p}}^{\top} Q^{(t)} v_{k,m}$ for $k \in [K]$ and $m \in [N]$;

2. Positional Attention Correlation: $\Upsilon^{(t)}_{\mathbf{p} \to \mathbf{q}} := e_{\mathbf{p}}^{\top} Q^{(t)} e_{\mathbf{q}}$.

By our initialization, we have $\Phi^{(0)}_{\mathbf{p} \to v_{k,m}} = \Upsilon^{(0)}_{\mathbf{p} \to \mathbf{q}} = 0$.

Next, we will apply the expression in Lemma D.1 to compute the gradient dynamics of these attention correlations.

### D.1.1 FORMAL STATEMENTS AND PROOF OF LEMMA C.1 AND C.2

We first introduce some notations. Given $\mathbf{r} \in \mathcal{U}$, for $\mathbf{p} \in \mathcal{P}$, $k \in [K]$ and $n \in [N]$ define the following quantities:

$$J_{\mathbf{r}}^{\mathbf{P}} := \mathsf{M}(X)_{\mathbf{r}}^{\top} \left( X_{\mathbf{p}} - [F(\mathsf{M}(X); Q)]_{\mathbf{p}} \right)$$

$$I_{\mathbf{r}}^{\mathbf{p}, k, n} := \left( \widetilde{\mathsf{M}}(X)_{\mathbf{r}} - \sum_{\mathbf{w} \in \mathcal{P}} \mathbf{attn}_{\mathbf{p} \to \mathbf{w}} \widetilde{\mathsf{M}}(X)_{\mathbf{w}} \right)^{\top} v_{k,n}$$

$$K_{\mathbf{r}}^{\mathbf{p}, \mathbf{q}} := \left( \widetilde{\mathsf{M}}(X)_{\mathbf{r}} - \sum_{\mathbf{w} \in \mathcal{P}} \mathbf{attn}_{\mathbf{p} \to \mathbf{w}} \widetilde{\mathsf{M}}(X)_{\mathbf{w}} \right)^{\top} e_{\mathbf{q}}$$

**Lemma D.3** (Formal statement of Lemma C.1). *Given $k \in [K]$, for $\mathbf{p} \in \mathcal{P}$, denote $n = a_{k,\mathbf{p}}$, let $\alpha^{(t)}_{\mathbf{p} \to v_{k,m}} = \frac{1}{\eta} \left( \Phi^{(t+1)}_{\mathbf{p} \to v_{k,m}} - \Phi^{(t)}_{\mathbf{p} \to v_{k,m}} \right)$ for $m \in [N_k]$, then*

*a. for $m = n$,*

$$\alpha^{(t)}_{\mathbf{p} \to v_{k,n}} = \mathbb{E}\left[ \mathbf{1}\{\mathbf{p} \in \mathcal{M}, k_X = k\} \mathbf{Attn}^{(t)}_{\mathbf{p} \to \mathcal{P}_{k,n}} \cdot \right.$$
$$\left. \left( z_n^3 \left( 1 - \mathbf{Attn}^{(t)}_{\mathbf{p} \to \mathcal{P}_{k,n}} \right)^2 + \sum_{a \neq n} z_a^2 z_n \left( \mathbf{Attn}^{(t)}_{\mathbf{p} \to \mathcal{P}_{k,a}} \right)^2 \right) \right];$$

*b. for $m \neq n$,*

$$\alpha^{(t)}_{\mathbf{p} \to v_{k,m}} = \mathbb{E}\left[ \mathbf{1}\{\mathbf{p} \in \mathcal{M}, k_X = k\} \mathbf{Attn}^{(t)}_{\mathbf{p} \to \mathcal{P}_{k,m}} \cdot \left( \sum_{a \neq m, n} z_a^2 z_m \left( \mathbf{Attn}^{(t)}_{\mathbf{p} \to \mathcal{P}_{k,a}} \right)^2 - \right. \right.$$
$$\left. \left. \left( z_m z_n^2 \left( 1 - \mathbf{Attn}^{(t)}_{\mathbf{p} \to \mathcal{P}_{k,n}} \right) \mathbf{Attn}^{(t)}_{\mathbf{p} \to \mathcal{P}_{k,n}} + z_m^3 \left( 1 - \mathbf{Attn}^{(t)}_{\mathbf{p} \to \mathcal{P}_{k,m}} \right) \mathbf{Attn}^{(t)}_{\mathbf{p} \to \mathcal{P}_{k,m}} \right) \right) \right].$$

*Proof.* From Lemma D.1, we have

$$\alpha^{(t)}_{\mathbf{p} \to v_{k,m}} = e_{\mathbf{p}}^{\top} \left( -\frac{\partial \mathcal{L}}{\partial Q} \right) v_{k,m}$$
$$= \mathbb{E}\left[ \mathbf{1}\{\mathbf{p} \in \mathcal{M}\} \sum_{\mathbf{r} \in \mathcal{U}} \mathbf{attn}_{\mathbf{p} \to \mathbf{r}} J_{\mathbf{r}}^{\mathbf{P}} \cdot I_{\mathbf{r}}^{\mathbf{p}, k, m} \right]$$
$$= \mathbb{E}\left[ \mathbf{1}\{\mathbf{p} \in \mathcal{M}, k_X = k\} \sum_{\mathbf{r} \in \mathcal{U}} \mathbf{attn}_{\mathbf{p} \to \mathbf{r}} J_{\mathbf{r}}^{\mathbf{P}} \cdot I_{\mathbf{r}}^{\mathbf{p}, k, m} \right]$$

where the last equality holds since when $k_X \neq k$, $I_{\mathbf{r}}^{\mathbf{p}, k, m} = 0$ due to orthogonality. Thus, in the following, we only need to consider the case $k_X = k$.

**Case 1:** $m = n$.

- For $\mathbf{r} \in \mathcal{U} \cap \mathcal{P}_{k,n}$, since $v_{k,n'} \perp v_{k,n}$ for $n' \neq n$, and $v_{k,n} \perp \{e_\mathbf{q}\}_{\mathbf{q} \in \mathcal{P}}$ we have

$$J_\mathbf{r}^\mathbf{P} = z_n v_{k,n}^\top \left( z_n v_{k,n} - \sum_{\mathbf{q} \in \mathcal{U} \cap \mathcal{P}_{k,n}} \mathbf{attn}_{\mathbf{p} \to \mathbf{q}} z_n v_{k,n} \right)$$

$$= z_n^2 \left( 1 - \mathbf{Attn}_{\mathbf{p} \to \mathcal{P}_{k,n}} \right)$$

$$I_\mathbf{r}^{\mathbf{P},k,n} = \left( z_n v_{k,n} - \sum_{\mathbf{q} \in \mathcal{U} \cap \mathcal{P}_{k,n}} \mathbf{attn}_{\mathbf{p} \to \mathbf{q}} z_n v_{k,n} \right)^\top v_{k,n} = J_\mathbf{r}^\mathbf{P}/z_n$$

- For $\mathbf{r} \in \mathcal{U} \cap \mathcal{P}_{k,n'}$ with $n' \neq n$

$$J_\mathbf{r}^\mathbf{P} = z_{n'} v_{k,n'}^\top \left( z_n v_{k,n} - \sum_{\mathbf{q} \in \mathcal{U} \cap \mathcal{P}_{k,n'}} \mathbf{attn}_{\mathbf{p} \to \mathbf{q}} z_{n'} v_{k,n'} \right)$$

$$= -z_{n'}^2 \mathbf{Attn}_{\mathbf{p} \to \mathcal{P}_{k,n'}}$$

$$I_\mathbf{r}^{\mathbf{P},k,n} = \left( z_{n'} v_{k,n'} - \sum_{\mathbf{q} \in \mathcal{U} \cap \mathcal{P}_{k,n}} \mathbf{attn}_{\mathbf{p} \to \mathbf{q}} z_n v_{k,n} \right)^\top v_{k,n}$$

$$= -z_n \mathbf{Attn}_{\mathbf{p} \to \mathcal{P}_{k,n}}$$

Putting it together, then we obtain:

$$e_\mathbf{p}^\top (-\frac{\partial L}{\partial Q}) v_{k,n} = \mathbb{E} \left[ \mathbf{1}\{\{\mathbf{p} \in \mathcal{M}, k_X = k\}\} \mathbf{Attn}_{\mathbf{p} \to \mathcal{P}_{k,n}}^{(t)} \cdot \right.$$

$$\left. \left( z_n^3 \left( 1 - \mathbf{Attn}_{\mathbf{p} \to \mathcal{P}_{k,n}}^{(t)} \right)^2 + \sum_{a \neq n} z_a^2 z_n \left( \mathbf{Attn}_{\mathbf{p} \to \mathcal{P}_{k,a}}^{(t)} \right)^2 \right) \right]$$

**Case** 2: $m \neq n$. Similarly

- For $\mathbf{r} \in \mathcal{U} \cap \mathcal{P}_{k,n}$

$$J_\mathbf{r}^\mathbf{P} = z_n v_{k,n}^\top \left( z_n v_{k,n} - \sum_{\mathbf{q} \in \mathcal{U} \cap \mathcal{P}_{k,n}} \mathbf{attn}_{\mathbf{p} \to \mathbf{q}} z_n v_{k,n} \right)$$

$$= z_n^2 (1 - \mathbf{Attn}_{\mathbf{p} \to \mathcal{P}_{k,n}})$$

$$I_\mathbf{r}^{\mathbf{P},k,m} = \left( z_n v_{k,n} - \sum_{\mathbf{q} \in \mathcal{U} \cap \mathcal{P}_{k,m}} \mathbf{attn}_{\mathbf{p} \to \mathbf{q}} z_m v_{k,m} \right)^\top v_{k,m}$$

$$= -z_m \mathbf{Attn}_{\mathbf{p} \to \mathcal{P}_{k,m}}$$

- For $\mathbf{r} \in \mathcal{U} \cap \mathcal{P}_{k,m}$

$$J_\mathbf{r}^\mathbf{P} = z_m v_{k,m}^\top \left( z_n v_{k,n} - \sum_{\mathbf{q} \in \mathcal{U} \cap \mathcal{P}_{k,m}} \mathbf{attn}_{\mathbf{p} \to \mathbf{q}}^{(t)} z_m v_{k,m} \right)$$

$$= -z_m^2 \mathbf{Attn}_{\mathbf{p} \to \mathcal{P}_{k,m}}$$

$$I_\mathbf{r}^{\mathbf{P},k,n} = \left( z_m v_{k,m} - \sum_{\mathbf{q} \in \mathcal{U} \cap \mathcal{P}_{k,m}} \mathbf{attn}_{\mathbf{p} \to \mathbf{q}}^{(t)} z_m v_{k,m} \right)^\top v_{k,m}$$

$$= z_n (1 - \mathbf{Attn}_{\mathbf{p} \to \mathcal{P}_{k,m}})$$

- For $\mathbf{r} \in \mathcal{U} \cap \mathcal{P}_{k,a}, a \neq n, m$

$$J_{\mathbf{r}}^{\mathbf{P}} = z_a v_{k,a}^\top \left( z_n v_{k,n} - \sum_{\mathbf{q} \in \mathcal{U} \cap \mathcal{P}_{k,a}} \mathbf{attn}_{\mathbf{p} \to \mathbf{q}}^{(t)} z_a v_{k,a} \right)$$

$$= -z_a^2 \mathbf{Attn}_{\mathbf{p} \to \mathcal{P}_{k,a}}$$

$$I_{\mathbf{r}}^{\mathbf{p},k,n} = \left( z_a v_{k,a} - \sum_{\mathbf{q} \in \mathcal{U} \cap \mathcal{P}_{k,m}} \mathbf{attn}_{\mathbf{p} \to \mathbf{q}}^{(t)} z_m v_{k,m} \right)^\top v_{k,m}$$

$$= -z_m \mathbf{Attn}_{\mathbf{p} \to \mathcal{P}_{k,m}}$$

Putting them together, then we complete the proof. $\qquad\square$

**Lemma D.4** (Formal statement of Lemma C.2). *Given $\mathbf{p}, \mathbf{q} \in \mathcal{P}$, let $\beta_{\mathbf{p} \to \mathbf{q}}^{(t)} = \frac{1}{\eta} \left( \Upsilon_{\mathbf{p} \to \mathbf{q}}^{(t+1)} - \Upsilon_{\mathbf{p} \to \mathbf{q}}^{(t)} \right)$, then*

$$\beta_{\mathbf{p} \to \mathbf{q}}^{(t)} = \sum_{k \in [K]} \beta_{k,\mathbf{p} \to \mathbf{q}}^{(t)}, \qquad \text{where } \beta_{k,\mathbf{p} \to \mathbf{q}}^{(t)} \text{ satisfies}$$

a. *if $a_{k,\mathbf{p}} = a_{k,\mathbf{q}} = n$,*

$$\beta_{k,\mathbf{p} \to \mathbf{q}}^{(t)} = \mathbb{E}\left[ \mathbf{1}\{\mathbf{p} \in \mathcal{M}, k_X = k\} \mathbf{attn}_{\mathbf{p} \to \mathbf{q}}^{(t)} \cdot \left( \sum_{a \neq n} z_a^2 \left( \mathbf{Attn}_{\mathbf{p} \to \mathcal{P}_{k,a}}^{(t)} \right)^2 + \right. \right.$$

$$\left. \left. z_n^2 \left( 1 - \mathbf{Attn}_{\mathbf{p} \to \mathcal{P}_{k,n}}^{(t)} \right) \left( \mathbf{1}\{\mathbf{q} \in \mathcal{U}\} - \mathbf{Attn}_{\mathbf{p} \to \mathcal{P}_{k,n}}^{(t)} \right) \right) \right];$$

b. *for $a_{k,\mathbf{p}} = n \neq m = a_{k,\mathbf{q}}$,*

$$\beta_{k,\mathbf{p} \to \mathbf{q}}^{(t)} = \mathbb{E}\left[ \mathbf{1}\{\mathbf{p} \in \mathcal{M},, k_X = k\} \mathbf{attn}_{\mathbf{p} \to \mathbf{q}}^{(t)} \cdot \left( \sum_{a \neq n} z_a^2 \left( \mathbf{Attn}_{\mathbf{p} \to \mathcal{P}_{k,a}}^{(t)} \right)^2 - \right. \right.$$

$$\left. \left. \left( z_n^2 \left( 1 - \mathbf{Attn}_{\mathbf{p} \to \mathcal{P}_{k,n}}^{(t)} \right) \mathbf{Attn}_{\mathbf{p} \to \mathcal{P}_{k,n}}^{(t)} + \mathbf{1}\{\mathbf{q} \in \mathcal{U}\} z_m^2 \mathbf{Attn}_{\mathbf{p} \to \mathcal{P}_{k,m}}^{(t)} \right) \right) \right].$$

*Proof.*

$$\beta_{\mathbf{p} \to \mathbf{q}}^{(t)} = e_{\mathbf{p}}^\top \left( -\frac{\partial \mathcal{L}}{\partial Q} \right) e_{\mathbf{q}} = \mathbb{E}[\mathbf{1}\{\mathbf{p} \in \mathcal{M}\} \sum_{\mathbf{r} \in \mathcal{U}} \mathbf{attn}_{\mathbf{p} \to \mathbf{r}}^{(t)} J_{\mathbf{r}}^{\mathbf{P}} K_{\mathbf{r}}^{\mathbf{p},\mathbf{q}}]$$

Then we let

$$\beta_{k,\mathbf{p} \to \mathbf{q}}^{(t)} := \mathbb{E}[\mathbf{1}\{\mathbf{p} \in \mathcal{M}, k_X = k\} \sum_{\mathbf{r} \in \mathcal{U}} \mathbf{attn}_{\mathbf{p} \to \mathbf{r}}^{(t)} J_{\mathbf{r}}^{\mathbf{P}} K_{\mathbf{r}}^{\mathbf{p},\mathbf{q}}].$$

In the following, we denote $a_{k,\mathbf{p}} = n$ and $a_{k,\mathbf{q}} = m$ for simplicity.

**Case 1:** $m = n$. If $\mathbf{q} \in \mathcal{U} \cap \mathcal{P}_{k,n}$:

- For $\mathbf{r} = \mathbf{q}$

$$J_{\mathbf{r}}^{\mathbf{P}} = z_n v_{k,n}^\top \left( z_n v_{k,n} - \sum_{\mathbf{w} \in \mathcal{U} \cap \mathcal{P}_{k,n}} \mathbf{attn}_{\mathbf{p} \to \mathbf{w}} z_n v_{k,n} \right)$$

$$= z_n^2 \left( 1 - \mathbf{Attn}_{\mathbf{p} \to \mathcal{P}_{k,n}} \right)$$

$$K_{\mathbf{r}}^{\mathbf{p},\mathbf{q}} = \left( e_{\mathbf{q}} - (\mathbf{attn}_{\mathbf{p} \to \mathbf{q}} e_{\mathbf{q}} + \sum_{\mathbf{w} \neq \mathbf{q}} \mathbf{attn}_{\mathbf{p} \to \mathbf{w}} e_{\mathbf{w}}) \right)^\top e_{\mathbf{q}}$$

$$= 1 - \mathbf{attn}_{\mathbf{p} \to \mathbf{q}}.$$

- For $\mathbf{r} \in \mathcal{U} \cap \mathcal{P}_{k,n}$, and $\mathbf{r} \neq \mathbf{q}$

$$J_{\mathbf{r}}^{\mathbf{P}} = z_n v_{k,n}^\top \left( z_n v_{k,n} - \sum_{\mathbf{w} \in \mathcal{U} \cap \mathcal{P}_{k,n}} \mathbf{attn_{p \to w}} z_n v_{k,n} \right)$$

$$= z_n^2 \left( 1 - \mathbf{Attn_{p \to \mathcal{P}_{k,n}}} \right)$$

$$K_{\mathbf{r}}^{\mathbf{P},\mathbf{q}} = (e_{\mathbf{r}} - (\mathbf{attn_{p \to q}} e_{\mathbf{q}} + \sum_{\mathbf{w} \neq \mathbf{q}} \mathbf{attn_{p \to w}} e_{\mathbf{w}}))^\top e_{\mathbf{q}}$$

$$= -\mathbf{attn_{p \to q}}$$

Thus

$$\sum_{\mathbf{r} \in \mathcal{U} \cap \mathcal{P}_{k,n}} \mathbf{attn_{p \to r}} J_{\mathbf{r}}^{\mathbf{P}} \cdot K_{\mathbf{r}}^{\mathbf{P},\mathbf{q}}$$

$$= z_n^2 \left( 1 - \sum_{\mathbf{w} \in \mathcal{U} \cap \mathcal{P}_{k,n}} \mathbf{attn_{p \to w}} \right)$$

$$\cdot \left( - \sum_{\mathbf{r} \in \mathcal{U} \cap \mathcal{P}_{k,n}} \mathbf{attn_{p \to r}} \mathbf{attn_{p \to q}} + \mathbf{attn_{p \to q}} \right)$$

$$= z_n^2 \left( 1 - \mathbf{Attn_{p \to \mathcal{P}_{k,n}}} \right)^2 \mathbf{attn_{p \to q}^{(t)}}$$

- For $\mathbf{r} \in \mathcal{U} \cap \mathcal{P}_{k,a}, a \neq n$

$$J_{\mathbf{r}}^{\mathbf{P}} = z_a v_{k,a}^\top \left( z_n v_{k,n} - \sum_{\mathbf{w} \in \mathcal{U} \cap \mathcal{P}_{k,a}} \mathbf{attn_{p \to w}} z_a v_{k,a} \right)$$

$$= -z_a^2 \sum_{\mathbf{w} \in \mathcal{U} \cap \mathcal{P}_{k,a}} \mathbf{attn_{p \to w}}$$

$$K_{\mathbf{r}}^{\mathbf{P},\mathbf{q}} = (e_{\mathbf{r}} - (\mathbf{attn_{p \to q}} e_{\mathbf{q}} + \sum_{\mathbf{w} \neq \mathbf{q}} \mathbf{attn_{p \to w}} e_{\mathbf{w}}))^\top e_{\mathbf{q}}$$

$$= -\mathbf{attn_{p \to q}}$$

Thus

$$\sum_{\mathbf{r} \in \mathcal{U}} \mathbf{attn_{p \to r}} J_{\mathbf{r}}^{\mathbf{P}} K_{\mathbf{r}}^{\mathbf{P},\mathbf{q}} = \mathbf{attn_{p \to q}} \cdot \left( z_n^2 \left( 1 - \mathbf{Attn_{p \to \mathcal{P}_{k,n}}} \right)^2 + \sum_{a \neq n} z_a^2 \left( \mathbf{Attn_{p \to \mathcal{P}_{k,a}}} \right)^2 \right)$$

If $\mathbf{q} \in \mathcal{M} \cap \mathcal{P}_{k,n}$:

- For $\mathbf{r} \in \mathcal{U} \cap \mathcal{P}_{k,n}$,

$$J_{\mathbf{r}}^{\mathbf{P}} = z_n v_{k,n}^\top \left( z_n v_{k,n} - \sum_{\mathbf{w} \in \mathcal{U} \cap \mathcal{P}_{k,n}} \mathbf{attn_{p \to w}} z_n v_{k,n} \right)$$

$$= z_n^2 \left( 1 - \mathbf{Attn_{p \to \mathcal{P}_{k,n}}} \right)$$

$$K_{\mathbf{r}}^{\mathbf{P},\mathbf{q}} = (e_{\mathbf{r}} - (\mathbf{attn_{p \to q}} e_{\mathbf{q}} + \sum_{\mathbf{w} \neq \mathbf{q}} \mathbf{attn_{p \to w}} e_{\mathbf{w}}))^\top e_{\mathbf{q}}$$

$$= -\mathbf{attn_{p \to q}}$$

- For $\mathbf{r} \in \mathcal{U} \cap \mathcal{P}_{k,a}, a \neq n$

$$J_{\mathbf{r}}^{\mathbf{P}} = z_a v_{k,a}^\top \left( z_n v_{k,n} - \sum_{\mathbf{w} \in \mathcal{U} \cap \mathcal{P}_{k,a}} \mathbf{attn_{p \to w}} z_a v_{k,a} \right)$$

$$= -z_a^2 \sum_{\mathbf{w} \in \mathcal{U} \cap \mathcal{P}_{k,a}} \mathbf{attn_{p \to w}}$$

$$K_{\mathbf{r}}^{\mathbf{p},\mathbf{q}} = (e_{\mathbf{r}} - (\mathbf{attn_{p \to q}} e_{\mathbf{q}} + \sum_{\mathbf{w} \neq \mathbf{q}} \mathbf{attn_{p \to w}} e_{\mathbf{w}}))^{\top} e_{\mathbf{q}}$$

$$= -\mathbf{attn_{p \to q}}$$

Thus

$$\sum_{\mathbf{r} \in \mathcal{U}} \mathbf{attn_{p \to r}} J_{\mathbf{r}}^{\mathbf{P}} K_{\mathbf{r}}^{\mathbf{p},\mathbf{q}}$$

$$= \mathbf{attn_{p \to q}} \cdot \left( z_n^2 \left(1 - \mathbf{Attn_{p \to \mathcal{P}_{k,n}}}\right)^2 - z_n^2 \left(1 - \mathbf{Attn_{p \to \mathcal{P}_{k,n}}}\right) + \sum_{a \neq n} z_a^2 \left(\mathbf{Attn_{p \to \mathcal{P}_{k,a}}}\right)^2 \right)$$

Putting it together,

$$\beta_{k,\mathbf{p} \to \mathbf{q}}^{(t)} = \mathbb{E} \left[ \mathbf{1}\{\mathbf{p} \in \mathcal{M}, k_X = k\} \mathbf{attn_{p \to q}} \cdot \right.$$

$$\left. \left( -z_n^2 \left(1 - \mathbf{Attn_{p \to \mathcal{P}_{k,n}}}\right) \mathbf{1}\{\mathbf{q} \in \mathcal{M}\} + z_n^2 \left(1 - \mathbf{Attn_{p \to \mathcal{P}_{k,n}}}\right)^2 + \sum_{m \neq n} z_m^2 \left(\mathbf{Attn_{p \to \mathcal{P}_{k,m}}}\right)^2 \right) \right]$$

**Case 2:** $m \neq n$. Similarly, if $\mathbf{q} \in \mathcal{U} \cap \mathcal{P}_{k,m}$:

- For $\mathbf{r} \in \mathcal{U} \cap \mathcal{P}_{k,n}$,

$$J_{\mathbf{r}}^{\mathbf{P}} = z_n v_{k,n}^{\top} \left( z_n v_{k,n} - \sum_{\mathbf{w} \in \mathcal{U} \cap \mathcal{P}_{k,n}} \mathbf{attn_{p \to w}} z_n v_{k,n} \right)$$

$$= z_n^2 (1 - \mathbf{Attn_{p \to \mathcal{P}_{k,n}}})$$

$$K_{\mathbf{r}}^{\mathbf{p},\mathbf{q}} = (e_{\mathbf{r}} - \mathbf{attn_{p \to q}} e_{\mathbf{q}} - \sum_{\mathbf{w} \neq \mathbf{q}} \mathbf{attn_{p \to w}} e_{\mathbf{w}})^{\top} e_{\mathbf{q}}$$

$$= -\mathbf{attn_{p \to q}}$$

- For $\mathbf{r} = \mathbf{q}$

$$J_{\mathbf{r}}^{\mathbf{P}} = z_m v_{k,m}^{\top} \left( z_n v_{k,n} - \sum_{\mathbf{w} \in \mathcal{U} \cap \mathcal{P}_{k,m}} \mathbf{attn_{p \to w}} z_m v_{k,m} \right)$$

$$= -z_m^2 \mathbf{Attn_{p \to \mathcal{P}_{k,m}}}$$

$$K_{\mathbf{r}}^{\mathbf{p},\mathbf{q}} = (e_{\mathbf{q}} - \mathbf{attn_{p \to q}} e_{\mathbf{q}} - \sum_{\mathbf{w} \neq \mathbf{w}} \mathbf{attn_{p \to w}} e_{\mathbf{w}})^{\top} e_{\mathbf{q}}$$

$$= 1 - \mathbf{attn_{p \to q}}$$

- For $\mathbf{r} \in \mathcal{U} \cap \mathcal{P}_{k,a}$, $a \neq n$, and $\mathbf{r} \neq \mathbf{q}$

$$J_{\mathbf{r}}^{\mathbf{P}} = z_a v_{k,a}^{\top} \left( z_n v_{k,n} - \sum_{\mathbf{w} \in \mathcal{U} \cap \mathcal{P}_{k,a}} \mathbf{attn_{p \to w}} z_a v_{k,a} \right)$$

$$= -z_a^2 \mathbf{Attn_{p \to \mathcal{P}_{k,a}}}$$

$$K_{\mathbf{r}}^{\mathbf{p},\mathbf{q}} = (e_{\mathbf{r}} - \mathbf{attn_{p \to q}} e_{\mathbf{q}} - \sum_{\mathbf{w} \neq \mathbf{q}} \mathbf{attn_{p \to w}} e_{\mathbf{w}})^{\top} e_{\mathbf{q}}$$

$$= -\mathbf{attn_{p \to q}}$$

Thus

$$\sum_{\mathbf{r}\in\mathcal{U}}\mathbf{attn_{p\to r}}J_{\mathbf{r}}^{\mathbf{P}}K_{\mathbf{r}}^{\mathbf{p,q}}$$

$$=\mathbf{attn_{p\to q}}\cdot\left(-z_n^2\left(1-\mathbf{Attn_{p\to\mathcal{P}_{k,n}}}\right)\mathbf{Attn_{p\to\mathcal{P}_{k,n}}}-z_m^2\mathbf{Attn_{p\to\mathcal{P}_{k,m}}}+\sum_{a\neq n}z_a^2\left(\mathbf{Attn_{p\to\mathcal{P}_{k,a}}}\right)^2\right)$$

If $\mathbf{q}\in\mathcal{M}\cap\mathcal{P}_{k,m}$:

- For $\mathbf{r}\in\mathcal{U}\cap\mathcal{P}_{k,n}$,

$$J_{\mathbf{r}}^{\mathbf{P}}=z_n v_{k,n}^{\top}\left(z_n v_{k,n}-\sum_{\mathbf{w}\in\mathcal{U}\cap\mathcal{P}_{k,n}}\mathbf{attn_{p\to w}}z_n v_{k,n}\right)$$
$$=z_n^2(1-\mathbf{Attn_{p\to\mathcal{P}_{k,n}}})$$
$$K_{\mathbf{r}}^{\mathbf{p,q}}=(e_{\mathbf{r}}-\mathbf{attn_{p\to q}}e_{\mathbf{q}}-\sum_{\mathbf{w}\neq\mathbf{q}}\mathbf{attn_{p\to w}}e_{\mathbf{w}})^{\top}e_{\mathbf{q}}$$
$$=-\mathbf{attn_{p\to q}}$$

- For $\mathbf{r}\in\mathcal{U}\cap\mathcal{P}_{k,a}$, $a\neq n$

$$J_{\mathbf{r}}^{\mathbf{P}}=z_a v_{k,a}^{\top}\left(z_n v_{k,n}-\sum_{\mathbf{w}\in\mathcal{U}\cap\mathcal{P}_{k,a}}\mathbf{attn_{p\to w}}z_a v_{k,a}\right)$$
$$=-z_a^2\mathbf{Attn_{p\to\mathcal{P}_{k,a}}}$$
$$K_{\mathbf{r}}^{\mathbf{p,q}}=(e_{\mathbf{r}}-\mathbf{attn_{p\to q}}e_{\mathbf{q}}-\sum_{\mathbf{w}\neq\mathbf{q}}\mathbf{attn_{p\to w}}e_{\mathbf{w}})^{\top}e_{\mathbf{q}}$$
$$=-\mathbf{attn_{p\to q}}$$

Thus

$$\sum_{\mathbf{r}\in\mathcal{U}}\mathbf{attn_{p\to r}}J_{\mathbf{r}}^{\mathbf{P}}K_{\mathbf{r}}^{\mathbf{p,q}}$$

$$=\mathbf{attn_{p\to q}}\cdot\left(-z_n^2\left(1-\mathbf{Attn_{p\to\mathcal{P}_{k,n}}}\right)\mathbf{Attn_{p\to\mathcal{P}_{k,n}}}+\sum_{a\neq n}z_a^2\left(\mathbf{Attn_{p\to\mathcal{P}_{k,a}}}\right)^2\right).$$

Therefore

$$\beta_{k,\mathbf{p\to q}}^{(t)}=\mathbb{E}\left[\mathbf{1}\{\mathbf{p}\in\mathcal{M},k_X=k\}\mathbf{attn_{p\to q}}\cdot\right.$$
$$\left(-z_n^2\left(1-\mathbf{Attn_{p\to\mathcal{P}_{k,n}}}\right)\mathbf{Attn_{p\to\mathcal{P}_{k,n}}}-\mathbf{1}\{\mathbf{q}\in\mathcal{U}\}z_m^2\mathbf{Attn_{p\to\mathcal{P}_{k,m}}}\right.$$
$$\left.\left.+\sum_{a\neq n}z_a^2\left(\mathbf{Attn_{p\to\mathcal{P}_{k,a}}}\right)^2\right)\right].$$

$\square$

Based on the above gradient update for $\Upsilon_{\mathbf{p\to q}}^{(t)}$, we further introduce the following auxiliary quantity, which will be useful in the later proof.

$$\Upsilon_{k,\mathbf{p\to q}}^{(t+1)}:=\Upsilon_{k,\mathbf{p\to q}}^{(t)}+\eta\beta_{k,\mathbf{p\to q}}^{(t)},\quad\text{with }\Upsilon_{k,\mathbf{p\to q}}^{(0)}=0\tag{D.2}$$

It is easy to verify that $\Upsilon_{\mathbf{p\to q}}^{(t)}=\sum_{k\in[K]}\Upsilon_{k,\mathbf{p\to q}}^{(t)}$.

## D.2 HIGH-PROBABILITY EVENT

We first introduce the following exponential bounds for the hypergeometric distribution Hyper $(m, D, M)$. Hyper $(m, D, M)$ describes the probability of certain successes (random draws for which the object drawn has a specified feature) in $m$ draws, without replacement, from a finite population of size $M$ that contains exactly $D$ objects with that feature, wherein each draw is either a success or a failure.

**Proposition D.5** (Greene & Wellner (2017)). *Suppose $S \sim$ Hyper $(m, D, M)$ with $1 \leq m, D \leq M$. Define $\mu_M := D/M$. Then for all $t > 0$*

$$P\left(|S - m\mu_M| > t\right) \leq 2\exp\left(-\frac{t^2}{4m\mu_M + 2t}\right).$$

We then utilize this property to prove the high-probability set introduced in Appendix C.1.

**Lemma D.6.** *For $k \in [K]$ $n \in [N]$, define*

$$\mathcal{E}_{k,n}(\gamma, P) := \{\mathsf{M} : |\mathcal{P}_{k,n} \cap \mathcal{U}| = \Theta(C_n)\}, \tag{D.3}$$

*we have*

$$\mathbb{P}(\mathsf{M} \in \mathcal{E}_{k,n}) \geq 1 - 2\exp(-c_{n,1}C_n) \tag{D.4}$$

*where $c_{n,0} > 0$ is some constant.*

*Proof.* Under the random masking strategy, given $k \in [K]$ and $n \in [N]$, $Y_{k,n} = |\mathcal{U} \cap \mathcal{P}_{k,n}|$ follows the hypergeometric distribution, i.e. $Y_{k,n} \sim$ Hyper $((1-\gamma)P, C_n, P)$. Then by tail bounds, for $t > 0$, we have:

$$\mathbb{P}[|Y_{k,n} - (1-\gamma)C_n| > t] \leq 2\exp(-\frac{t^2}{4(1-\gamma)C_n + 2t})$$

Letting $t = \Theta(C_n)$, we have

$$\mathbb{P}[Y_{k,n} = \Theta(C_n)] \geq 1 - 2e^{-c_{n,1}C_n}.$$

$\square$

We further have the following fact, which will be useful for proving the property of loss objective in the next subsection.

**Lemma D.7.** *For $k \in [K]$ and $n \in [N]$, we have*

$$\mathbb{P}(|\mathcal{U} \cap \mathcal{P}_{k,n}| = 0) \leq \exp(-c_{n,0}C_n). \tag{D.5}$$

*where $c_{n,0} > 0$ is some constant.*

*Proof.* By the form of probability density for Hyper $((1-\gamma)P, C_n, P)$, we have

$$\mathbb{P}(|\mathcal{U} \cap \mathcal{P}_{k,n}| = 0) = \frac{\binom{C_n}{0}\binom{P-C_n}{(1-\gamma)P}}{\binom{P}{(1-\gamma)P}}$$

$$\leq \gamma^{C_n} = \exp(-c_{n,0}C_n)).$$

$\square$

## D.3 PROPERTIES OF LOSS FUNCTION

Recall the training and regional reconstruction loss we consider are given by:

$$\mathcal{L}(Q) := \frac{1}{2}\mathbb{E}\left[\sum_{\mathbf{p}\in\mathcal{P}} \mathbb{1}\{\mathbf{p} \in \mathcal{M}\} \|[F(\mathsf{M}(X); Q, E)]_{\mathbf{p}} - X_{\mathbf{p}}\|^2\right] \tag{D.6}$$

$$\mathcal{L}_{\mathbf{p}}(Q) = \frac{1}{2}\mathbb{E}\left[\mathbb{1}\{\mathbf{p} \in \mathcal{M}\} \|[F(\mathsf{M}(X), E)]_{\mathbf{p}} - X_{\mathbf{p}}\|^2\right] \tag{D.7}$$

In this part, we will present several important lemmas for such a training objective. We first single out the following lemma, which connects the loss form with the attention score.

**Lemma D.8** (Loss Calculation). *The population loss $L(Q)$ can be decomposed into the following form:*

$$\mathcal{L}(Q) = \sum_{\mathbf{p} \in \mathcal{P}} \mathcal{L}_{\mathbf{p}}(Q), \text{ where}$$

$$\mathcal{L}_{\mathbf{p}}(Q) = \frac{1}{2} \sum_{k=1}^{K} \mathbb{E}\left[\mathbf{1}\{\mathbf{p} \in \mathcal{M}, k_X = k\} \cdot \right.$$
$$\left. \left( z_{a_{k,\mathbf{p}}}^2 \left(1 - \mathbf{Attn}_{\mathbf{p} \to \mathcal{P}_{k,a_{k,\mathbf{p}}}}^{(t)}\right)^2 + \sum_{a \neq a_{k,\mathbf{p}}} z_a^2 \left(\mathbf{Attn}_{\mathbf{p} \to \mathcal{P}_{k,a}}^{(t)}\right)^2 \right)\right]$$

*Proof.*

$$\mathcal{L}_{\mathbf{p}}(Q)$$
$$= \frac{1}{2} \sum_{k=1}^{K} \mathbb{E}\left[\mathbb{1}\{\mathbf{p} \in \mathcal{M}, k_X = k\} \|[F(\mathsf{M}(X), E)]_{\mathbf{p}} - X_{\mathbf{p}}\|^2\right]$$
$$= \frac{1}{2} \sum_{k=1}^{K} \mathbb{E}\left[\mathbb{1}\{\mathbf{p} \in \mathcal{M}, k_X = k\} \left\|\sum_{m \in [N]} \mathbf{Attn}_{\mathbf{p} \to \mathcal{P}_{k,m}} z_m v_{k,m} - z_{a_{k,\mathbf{p}}} v_{k,a_{k,\mathbf{p}}}\right\|^2\right]$$
$$\overset{(i)}{=} \frac{1}{2} \sum_{k=1}^{K} \mathbb{E}\left[\mathbb{1}\{\mathbf{p} \in \mathcal{M}, k_X = k\} \left( z_{a_{k,\mathbf{p}}}^2 \left(1 - \mathbf{Attn}_{\mathbf{p} \to \mathcal{P}_{k,a_{k,\mathbf{p}}}}\right)^2 + \sum_{m \neq a_{k,\mathbf{p}}} z_m^2 \left(\mathbf{Attn}_{\mathbf{p} \to \mathcal{P}_{k,m}}\right)^2 \right)\right]$$

where $(i)$ follows since the features are orthogonal. $\qquad\square$

We then introduce some additional crucial notations for the loss objectives.

$$\mathcal{L}_{\mathbf{p}}^* = \min_{Q \in \mathbb{R}^{d \times d}} \mathcal{L}_{\mathbf{p}}(Q), \tag{D.8a}$$

$$\mathcal{L}_{\mathbf{p}}^{low} = \frac{1}{2}(\sigma_z^2 + \frac{L^2}{N-1}) \sum_{k \in [K]} \mathbb{P}\left(|\mathcal{U} \cap \mathcal{P}_{k,z_{a_{k,\mathbf{p}}}}| = 0\right) \tag{D.8b}$$

$$\widetilde{\mathcal{L}}_{\mathbf{p}}(Q) = \sum_{k=1}^{K} \widetilde{\mathcal{L}}_{k,\mathbf{p}}(Q), \quad \text{where}$$

$$\widetilde{\mathcal{L}}_{k,\mathbf{p}}(Q) = \frac{1}{2}\mathbb{E}\left[\mathbf{1}\{\mathbf{p} \in \mathcal{M}, k_X = k, \mathsf{M} \in \mathcal{E}_{k,z_{a_{k,\mathbf{p}}}}\} \cdot \right.$$
$$\left. \left( z_{a_{k,\mathbf{p}}}^2 \left(1 - \mathbf{Attn}_{\mathbf{p} \to \mathcal{P}_{k,a_{k,\mathbf{p}}}}^{(t)}\right)^2 + \sum_{a \neq a_{k,\mathbf{p}}} z_a^2 \left(\mathbf{Attn}_{\mathbf{p} \to \mathcal{P}_{k,a}}^{(t)}\right)^2 \right)\right] \tag{D.8c}$$

Here $\sigma_z^2 = \mathbb{E}[Z_n(X)^2]$. $\mathcal{L}_{\mathbf{p}}^\star$ denotes the minimum value of the population loss in equation D.7, and $\mathcal{L}_{\mathbf{p}}^{low}$ represents the unavoidable errors for $\mathbf{p} \in \mathcal{P}$, given that all the patches in $\mathcal{P}_{k,a_{k,\mathbf{p}}}$ are masked. We will show that $\mathcal{L}_{\mathbf{p}}^{low}$ serves as a lower bound for $\mathcal{L}_{\mathbf{p}}^\star$, and demonstrate that the network trained with GD will attain nearly zero error compared to $\mathcal{L}_{\mathbf{p}}^{low}$. Our convergence will be established by the sub-optimality gap with respect to $\mathcal{L}_{\mathbf{p}}^{low}$, which necessarily implies the convergence to $\mathcal{L}_{\mathbf{p}}^\star$. (It also implies $\mathcal{L}_{\mathbf{p}}^\star - \mathcal{L}_{\mathbf{p}}^{low}$ is small.)

**Lemma D.9.** *For $\mathcal{L}_{\mathbf{p}}^\star$ and $\mathcal{L}_{\mathbf{p}}^{low}$ defined in equation D.8a and equation D.8b, respectively, we have $\mathcal{L}_{\mathbf{p}}^{low} \leq \mathcal{L}_{\mathbf{p}}^\star$ and they are both at the order of $\Theta\left(\exp\left(-\left(c_1 P^{\kappa_c} + \mathbb{1}\left\{1 \notin \cup_{k \in [K]}\{a_{k,\mathbf{p}}\}\right\}c_2 P^{\kappa_s}\right)\right)\right)$ where $c_1, c_2 > 0$ are some constants.*

*Proof.* We first prove $\mathcal{L}_{\mathbf{p}}^{\text{low}} \leq \mathcal{L}_{\mathbf{p}}^{\star}$:

$$
\mathcal{L}_{\mathbf{p}}^{\star} = \min_{Q \in \mathbb{R}^{d \times d}} \frac{1}{2} \sum_{k=1}^{K} \mathbb{E}\left[ \mathbf{1}\{\mathbf{p} \in \mathcal{M}, k_X = k\} \cdot \right.
$$
$$
\left. \left( z_{a_{k,\mathbf{p}}}^{3} \left( 1 - \mathbf{Attn}_{\mathbf{p} \to \mathcal{P}_{k,a_{k,\mathbf{p}}}}^{(t)} \right)^{2} + \sum_{a \neq a_{k,\mathbf{p}}} z_{a}^{2} z_{a_{k,\mathbf{p}}} \left( \mathbf{Attn}_{\mathbf{p} \to \mathcal{P}_{k,a}}^{(t)} \right)^{2} \right) \right]
$$
$$
\geq \min_{Q \in \mathbb{R}^{d \times d}} \frac{1}{2} \sum_{k=1}^{K} \mathbb{E}\left[ \mathbf{1}\{\mathbf{p} \in \mathcal{M}, k_X = k\} \mathbb{1}\{|\mathcal{U} \cap \mathcal{P}_{k,a_{\mathbf{p},k}}| = 0\} \cdot \right.
$$
$$
\left. \left( z_{a_{k,\mathbf{p}}}^{3} \left( 1 - \mathbf{Attn}_{\mathbf{p} \to \mathcal{P}_{k,a_{k,\mathbf{p}}}}^{(t)} \right)^{2} + \sum_{a \neq a_{k,\mathbf{p}}} z_{a}^{2} z_{a_{k,\mathbf{p}}} \left( \mathbf{Attn}_{\mathbf{p} \to \mathcal{P}_{k,a}}^{(t)} \right)^{2} \right) \right]
$$

Notice that when all patches in $\mathcal{P}_{k,a_{k,\mathbf{p}}}$ are masked, $\mathbf{Attn}_{\mathbf{p} \to \mathcal{P}_{k,a_{k,\mathbf{p}}}}^{(t)} = 0$. Moreover,

$$
\sum_{m \neq a_{k,\mathbf{p}}} z_m^2 \mathbf{Attn}_{\mathbf{p} \to \mathcal{P}_{k,m}}^{(t)} \geq \frac{L^2}{N-1}
$$

by Cauchy–Schwarz inequality. Thus

$$
\mathcal{L}_{\mathbf{p}}^{\star} \geq \frac{1}{2} \sum_{k=1}^{K} (\sigma_z^2 + \frac{L^2}{N-1}) \mathbb{P}\left( |\mathcal{U} \cap \mathcal{P}_{k,a_{k,\mathbf{p}}}| = 0 \right) = \mathcal{L}_{\mathbf{p}}^{\text{low}}.
$$

$\mathcal{L}_{\mathbf{p}}^{\text{low}} = \Theta\left( \exp\left( - \left( c_1 P^{\kappa_c} + \mathbb{1}\left\{ 1 \notin \cup_{k \in [K]}\{a_{k,\mathbf{p}}\} \right\} c_2 P^{\kappa_s} \right) \right) \right)$ immediately comes from Lemma D.7. Furthermore, we only need to show $\mathcal{L}_{\mathbf{p}}^{\star} = O\left( \exp\left( - \left( c_1 P^{\kappa_c} + \mathbb{1}\left\{ 1 \notin \right. \right. \right. \right.$ $\left. \left. \left. \cup_{k \in [K]}\{a_{k,\mathbf{p}}\} \right\} c_2 P^{\kappa_s} \right) \right) \right)$. This can be directly obtained by choosing $Q = \sigma I_d$ for some sufficiently large $\sigma$ and hence omitted here. $\qquad\square$

**Lemma D.10.** *Given* $\mathbf{p} \in \mathcal{P}$, *for any* $Q$, *we have*

$$
\widetilde{\mathcal{L}}_{\mathbf{p}}(Q) \leq L_{\mathbf{p}}(Q) - \mathcal{L}_{\mathbf{p}}^{\text{low}} \leq \widetilde{\mathcal{L}}_{\mathbf{p}}(Q) + O\left( \exp\left( - \left( c_3 P^{\kappa_c} + \mathbb{1}\left\{ 1 \notin \cup_{k \in [K]}\{a_{k,\mathbf{p}}\} \right\} c_4 P^{\kappa_s} \right) \right) \right).
$$

*where* $c_3, c_4 > 0$ *are some constants.*

*Proof.* The lower bound is directly obtained by the definition and thus we only prove the upper bound.

$L_{\mathbf{p}}(Q) - \widetilde{\mathcal{L}}_{\mathbf{p}}(Q)$

$$
= \frac{1}{2} \sum_{k=1}^{K} \mathbb{E}\left[ \mathbf{1}\{\mathbf{p} \in \mathcal{M}, k_X = k, \mathsf{M} \in \mathcal{E}_{k,z_{a_{k,\mathbf{p}}}}^{c}\} \cdot \left( z_{a_{k,\mathbf{p}}}^{2} \left( 1 - \mathbf{Attn}_{\mathbf{p} \to \mathcal{P}_{k,a_{k,\mathbf{p}}}}^{(t)} \right)^{2} + \sum_{a \neq a_{k,\mathbf{p}}} z_{a}^{2} \left( \mathbf{Attn}_{\mathbf{p} \to \mathcal{P}_{k,a}}^{(t)} \right)^{2} \right) \right]
$$
$$
\leq \sum_{k=1}^{K} U^2 \mathbb{P}(\mathsf{M} \in \mathcal{E}_{k,z_{a_{k,\mathbf{p}}}}^{c})
$$
$$
\leq O\left( \exp\left( - \left( c_3 P^{\kappa_c} + \mathbb{1}\left\{ 1 \notin \cup_{k \in [K]}\{a_{k,\mathbf{p}}\} \right\} c_4 P^{\kappa_s} \right) \right) \right).
$$

where the last inequality follows from Lemma D.6.

$\qquad\square$

# E    OVERALL INDUCTION HYPOTHESES AND PROOF PLAN FOR MAE

Our main proof utilizes the induction hypotheses. In this section, we introduce the main induction hypotheses for the positive and negative information gaps, which will later be proven to be valid throughout the entire learning process.

### E.1 POSITIVE INFORMATION GAP

We first state our induction hypothesis for the case that the information gap $\Delta$ is positive.

**Induction Hypothesis E.1.** For $t \le T$, given $\mathbf{p}, \mathbf{q} \in \mathcal{P}$, for $k \in [K]$, the following holds

    a. $\Phi^{(t)}_{\boldsymbol{p} \to v_{k,a_{k,\mathbf{p}}}}$ is monotonically increasing, and $\Phi^{(t)}_{\boldsymbol{p} \to v_{k,a_{k,\mathbf{p}}}} \in [0, \widetilde{O}(1)]$;

    b. if $a_{k,\mathbf{p}} \ne 1$, then $\Phi^{(t)}_{\boldsymbol{p} \to v_{k,1}}$ is monotonically decreasing and $\Phi^{(t)}_{\boldsymbol{p} \to v_{k,1}} \in [-\widetilde{O}(1), 0]$;

    c. $|\Phi^{(t)}_{\boldsymbol{p} \to v_{k,m}}| = \widetilde{O}(\frac{1}{P^{1-\kappa_s}})$ for $m \notin \{1\} \cup \{a_{k,\mathbf{p}}\}$;

    d. for $\mathbf{q} \ne \mathbf{p}$, $\Upsilon^{(t)}_{\mathbf{p} \to \mathbf{q}} = \widetilde{O}(\frac{1}{P^{\kappa_s}})$;

    e. $\Upsilon^{(t)}_{\mathbf{p} \to \mathbf{p}} = \widetilde{O}(\frac{1}{P})$.

### E.2 NEGATIVE INFORMATION GAP

Now we turn to the case that $\Delta \le -\Omega(1)$.

**Induction Hypothesis E.2.** For $t \le T$, given $\mathbf{p}, \mathbf{q} \in \mathcal{P}$, for $k \in [K]$, the following holds

    a. $\Phi^{(t)}_{\boldsymbol{p} \to v_{k,a_{k,\mathbf{p}}}}$ is monotonically increasing, and $\Phi^{(t)}_{\boldsymbol{p} \to v_{k,a_{k,\mathbf{p}}}} \in [0, \widetilde{O}(1)]$;

    b. if $a_{k,\mathbf{p}} \ne 1$, then $\Phi^{(t)}_{\boldsymbol{p} \to v_{k,1}}$ is monotonically decreasing and $\Phi^{(t)}_{\boldsymbol{p} \to v_{k,1}} \in [-\widetilde{O}(\frac{1}{P^{-\Delta}}), 0]$;

    c. $|\Phi^{(t)}_{\boldsymbol{p} \to v_{k,m}}| = \widetilde{O}(\frac{1}{P^{1-\kappa_s}})$ for $m \notin \{1\} \cup \{a_{k,\mathbf{p}}\}$;

    d. for $\mathbf{q} \ne \mathbf{p}$, $\Upsilon^{(t)}_{\mathbf{p} \to \mathbf{q}} = \widetilde{O}(\frac{1}{P^{\kappa_s}})$;

    e. $\Upsilon^{(t)}_{\mathbf{p} \to \mathbf{p}} = \widetilde{O}(\frac{1}{P})$.

### E.3 PROOF OUTLINE

In both settings, we can classify the process through which transformers learn the feature attention correlation $\Phi^{(t)}_{\boldsymbol{p} \to v_{k,a_{k,\mathbf{p}}}}$ into two distinct scenarios. These scenarios hinge on the spatial relation of the area $\boldsymbol{p}$ within the context of the $k$-th partition $\mathcal{D}_k$, specifically, whether $\boldsymbol{p}$ is located in the global area of the $k$-th cluster, i.e. whether $a_{k,\boldsymbol{p}} = 1$. The learning dynamics exhibit different behaviors of learning the local FP correlation in the local area with different $\Delta$, while the behaviors for features located in the global area are very similar, unaffected by the value of $\Delta$. Therefore, through Appendices F to H, we delve into the learning phases and provide technical proofs for the local area with $\Delta \ge \Omega(1)$, local area with $\Delta \le -\Omega(1)$ and the global area respectively. Finally, we will put this analysis together to prove that the Induction Hypothesis E.1 (resp. Induction Hypothesis E.2) holds during the entire training process, thereby validating the main theorems in Appendix I.

## F ANALYSIS FOR THE LOCAL AREA WITH POSITIVE INFORMATION GAP

In this section, we focus on a specific patch $\mathbf{p} \in \mathcal{P}$ with the $k$-th cluster for $k \in [K]$, and present the analysis for the case that $X_{\mathbf{p}}$ is located in the local area for the $k$-th cluster, i.e. $a_{k,\mathbf{p}} > 1$. We will analyze the case that $\Delta \ge \Omega(1)$. Throughout this section, we denote $a_{k,\boldsymbol{p}} = n$ for simplicity. We will analyze the convergence of the training process via two phases of dynamics. At the beginning of each phase, we will establish an induction hypothesis, which we expect to remain valid throughout that phase. Subsequently, we will analyze the dynamics under such a hypothesis within the phase, aiming to provide proof of the hypothesis by the end of the phase.

### F.1 PHASE I, STAGE 1

In this section, we shall discuss the initial stage of phase I. Firstly, we present the induction hypothesis in this stage.

We define the stage 1 of phase I as all iterations $t \leq T_1$, where

$$T_1 \triangleq \max \left\{ t : \Phi^{(t)}_{\boldsymbol{p} \to v_{k,n}} \geq -\frac{1}{U} \left( \frac{\Delta}{2} - 0.01 \right) \log(P) \right\}.$$

We state the following induction hypotheses, which will hold throughout this period:

**Induction Hypothesis F.1.** For each $0 \leq t \leq T_1$, $\mathbf{q} \in \mathcal{P} \setminus \{\mathbf{p}\}$, the following holds:

a. $\Phi^{(t)}_{\boldsymbol{p} \to v_{k,n}}$ is monotonically increasing, and $\Phi^{(t)}_{\boldsymbol{p} \to v_{k,n}} \in [0, O\left( \frac{\left( \frac{\Delta}{2} - 0.01 \right) \log(P)}{P^{0.02}} \right)]$;

b. $\Phi^{(t)}_{\boldsymbol{p} \to v_{k,1}}$ is monotonically decreasing and $\Phi^{(t)}_{\boldsymbol{p} \to v_{k,1}} \in [-\frac{1}{U} \left( \frac{\Delta}{2} - 0.01 \right) \log(P), 0]$;

c. $|\Phi^{(t)}_{\boldsymbol{p} \to v_{k,m}}| = O\left( \frac{\Phi^{(t)}_{\boldsymbol{p} \to v_{k,n}} - \Phi^{(t)}_{\boldsymbol{p} \to v_{k,1}}}{P^{1-\kappa_s}} \right)$ for $m \neq 1, n$;

d. $\Upsilon^{(t)}_{k, \boldsymbol{p} \to \mathbf{q}} = O\left( \frac{\Phi^{(t)}_{\boldsymbol{p} \to v_{k,n}}}{C_n} \right)$ for $a_{k,\mathbf{q}} = n$, $|\Upsilon^{(t)}_{k, \mathbf{p} \to \mathbf{p}}| = O\left( \frac{\Phi^{(t)}_{\boldsymbol{p} \to v_{k,n}} - \Phi^{(t)}_{\boldsymbol{p} \to v_{k,1}}}{P} \right)$;

e. $|\Upsilon^{(t)}_{k, \mathbf{p} \to \mathbf{q}}| = O\left( \frac{|\Phi^{(t)}_{\boldsymbol{p} \to v_{k,1}}|}{C_1} \right) + O\left( \frac{\Phi^{(t)}_{\boldsymbol{p} \to v_{k,n}} - \Phi^{(t)}_{\boldsymbol{p} \to v_{k,1}}}{P} \right)$ for $a_{k,\mathbf{q}} = 1$;

f. $|\Upsilon^{(t)}_{k, \mathbf{p} \to \mathbf{q}}| = O\left( \frac{\Phi^{(t)}_{\boldsymbol{p} \to v_{k,n}} - \Phi^{(t)}_{\boldsymbol{p} \to v_{k,1}}}{P} \right)$ for $a_{k,\mathbf{q}} \neq 1, n$.

#### F.1.1 PROPERTY OF ATTENTION SCORES

We first introduce several properties of the attention score if Induction Hypothesis E.1 and Induction Hypothesis F.1 hold.

**Lemma F.1.** *For $n > 1$, if Induction Hypothesis E.1 and Induction Hypothesis F.1 hold at iteration $t \leq T_1$, then the following holds*

1. $1 - \mathbf{Attn}^{(t)}_{\mathbf{p} \to \mathcal{P}_{k,n}} - \mathbf{Attn}^{(t)}_{\mathbf{p} \to \mathcal{P}_{k,1}} \geq \Omega(1)$;

2. *If* $\mathsf{M} \in \mathcal{E}_{k,n}$, $\mathbf{Attn}^{(t)}_{\mathbf{p} \to \mathcal{P}_{k,n}} = \Theta\left( \frac{1}{P^{1-\kappa_s}} \right)$;

3. *Moreover, if* $\mathsf{M} \in \mathcal{E}_{k,1}$, *we have* $\mathbf{Attn}^{(t)}_{\mathbf{p} \to \mathcal{P}_{k,1}} = \Omega\left( \frac{1}{P^{\frac{1-\kappa_s}{2} - 0.01}} \right)$;

4. *For* $\mathbf{q} \in \mathcal{M} \cap (\mathcal{P}_{k,n} \cup \mathcal{P}_{k,1})$, $\mathbf{attn}^{(t)}_{\mathbf{p} \to \mathbf{q}} = O\left( \frac{1 - \mathbf{Attn}^{(t)}_{\mathbf{p} \to \mathcal{P}_{k,1}} - \mathbf{Attn}^{(t)}_{\mathbf{p} \to \mathcal{P}_{k,n}}}{P} \right)$.

**Lemma F.2.** *For $n > 1$, if Induction Hypothesis E.1 and Induction Hypothesis F.1 hold at iteration $t \leq T_1$, then for $m \neq n, 1$, the following holds:*

1. *For any* $\mathbf{q} \in \mathcal{P}_{k,m}$, $\mathbf{attn}^{(t)}_{\mathbf{p} \to \mathbf{q}} \leq O\left( \frac{1 - \mathbf{Attn}^{(t)}_{\mathbf{p} \to \mathcal{P}_{k,1}} - \mathbf{Attn}^{(t)}_{\mathbf{p} \to \mathcal{P}_{k,n}}}{P} \right)$.

2. *Moreover,* $\mathbf{Attn}^{(t)}_{\mathbf{p} \to \mathcal{P}_{k,m}} \leq O\left( \frac{1 - \mathbf{Attn}^{(t)}_{\mathbf{p} \to \mathcal{P}_{k,1}} - \mathbf{Attn}^{(t)}_{\mathbf{p} \to \mathcal{P}_{k,n}}}{N} \right)$.

The above properties can be easily verified through direct calculations by using the definition in equation 2.6 and conditions in Induction Hypothesis F.1, which are omitted here for brevity.

### F.1.2 BOUNDING THE GRADIENT UPDATES FOR FP CORRELATIONS

**Lemma F.3.** *For $n > 1$, if Induction Hypothesis E.1 and Induction Hypothesis F.1 hold at iteration $0 \le t \le T_1$, then $\alpha^{(t)}_{\mathbf{p} \to v_{k,n}} \ge 0$ and satisfies:*

$$\alpha^{(t)}_{\mathbf{p} \to v_{k,n}} = \Theta\Big(\frac{C_n}{P}\Big) = \Theta\Big(\frac{1}{P^{1-\kappa_s}}\Big).$$

*Proof.* By Lemma C.2, we have

$$\alpha^{(t)}_{\mathbf{p} \to v_{k,n}}$$

$$= \mathbb{E}\left[ \mathbf{1}\{k_X = k, \mathbf{p} \in \mathcal{M}\} \mathbf{Attn}^{(t)}_{\mathbf{p} \to \mathcal{P}_{k,n}} \cdot \left( z_n^3 \left(1 - \mathbf{Attn}^{(t)}_{\mathbf{p} \to \mathcal{P}_{k,n}}\right)^2 + \sum_{m \neq n} z_m^2 z_n \left(\mathbf{Attn}^{(t)}_{\mathbf{p} \to \mathcal{P}_{k,m}}\right)^2 \right) \right]$$

$$= \mathbb{E}\left[ \mathbf{1}\{k_X = k, \mathcal{E}_{k,n} \cap \mathbf{p} \in \mathcal{M}\} \mathbf{Attn}^{(t)}_{\mathbf{p} \to \mathcal{P}_{k,n}} \cdot \left( z_n^3 \left(1 - \mathbf{Attn}^{(t)}_{\mathbf{p} \to \mathcal{P}_{k,n}}\right)^2 + \sum_{m \neq n} z_m^2 z_n \left(\mathbf{Attn}^{(t)}_{\mathbf{p} \to \mathcal{P}_{k,m}}\right)^2 \right) \right]$$

$$+ \mathbb{E}\left[ \mathbf{1}\{k_X = k, \mathcal{E}^c_{k,n} \cap \mathbf{p} \in \mathcal{M}\} \mathbf{Attn}^{(t)}_{\mathbf{p} \to \mathcal{P}_{k,n}} \cdot \left( z_n^3 \left(1 - \mathbf{Attn}^{(t)}_{\mathbf{p} \to \mathcal{P}_{k,n}}\right)^2 + \sum_{m \neq n} z_m^2 z_n \left(\mathbf{Attn}^{(t)}_{\mathbf{p} \to \mathcal{P}_{k,m}}\right)^2 \right) \right]$$

$$\le \mathbb{P}(\mathsf{M} \in \mathcal{E}_{k,n})$$

$$\cdot \mathbb{E}\left[ \mathbf{1}\{k_X = k, \mathbf{p} \in \mathcal{M}\} \mathbf{Attn}^{(t)}_{\mathbf{p} \to \mathcal{P}_{k,n}} \cdot \left( z_n^3 \left(1 - \mathbf{Attn}^{(t)}_{\mathbf{p} \to \mathcal{P}_{k,n}}\right)^2 + \sum_{m \neq n} z_m^2 z_n \left(\mathbf{Attn}^{(t)}_{\mathbf{p} \to \mathcal{P}_{k,m}}\right)^2 \right) \Big| \mathcal{E}_{k,n} \right]$$

$$+ O(1) \cdot \mathbb{P}(\mathsf{M} \in \mathcal{E}^c_{k,n})$$

$$\le O\Big(\frac{C_n}{P}\Big) + O(\exp(-c_{n,1} C_n))$$

$$\le O\Big(\frac{C_n}{P}\Big),$$

where the second inequality invokes Lemma F.1 and Lemma D.6, and the last inequality is due to $exp(-c_{n,1}C_n) \ll \frac{C_n}{P}$. Similarly, we can show that $\alpha^{(t)}_{\mathbf{p} \to v_{k,n}} \ge \Omega(\frac{C_n}{P})$.

$\square$

**Lemma F.4.** *For $n > 1$, if Induction Hypothesis E.1 and Induction Hypothesis F.1 hold at iteration $0 \le t \le T_1$, then $\alpha^{(t)}_{\mathbf{p} \to v_{k,1}} < 0$ and satisfies*

$$|\alpha^{(t)}_{\mathbf{p} \to v_{k,1}}| \ge \Omega\Big(\frac{1}{P^{2(\frac{1-\kappa_s}{2} - 0.01)}}\Big) = \Omega\Big(\frac{1}{P^{0.98 - \kappa_s}}\Big).$$

*Proof.* We first single out the following fact:

$$- z_1 z_n^2 \left(1 - \mathbf{Attn}^{(t)}_{\mathbf{p} \to \mathcal{P}_{k,n}}\right) \mathbf{Attn}^{(t)}_{\mathbf{p} \to \mathcal{P}_{k,n}} - z_1^3 \left(1 - \mathbf{Attn}^{(t)}_{\mathbf{p} \to \mathcal{P}_{k,1}}\right) \mathbf{Attn}^{(t)}_{\mathbf{p} \to \mathcal{P}_{k,1}} + \sum_{a \neq 1,n} z_a^2 z_1 \left(\mathbf{Attn}^{(t)}_{\mathbf{p} \to \mathcal{P}_{k,a}}\right)^2$$

$$\le z_1 \left( \max_{a \neq 1,n} z_a^2 \mathbf{Attn}^{(t)}_{\mathbf{p} \to \mathcal{P}_{k,a}} - z_n^2 \mathbf{Attn}^{(t)}_{\mathbf{p} \to \mathcal{P}_{k,n}} - z_1^2 \mathbf{Attn}^{(t)}_{\mathbf{p} \to \mathcal{P}_{k,1}} \right) (1 - \mathbf{Attn}^{(t)}_{\mathbf{p} \to \mathcal{P}_{k,n}} - \mathbf{Attn}^{(t)}_{\mathbf{p} \to \mathcal{P}_{k,1}})$$

$$= -z_1 (1 - \mathbf{Attn}^{(t)}_{\mathbf{p} \to \mathcal{P}_{k,n}} - \mathbf{Attn}^{(t)}_{\mathbf{p} \to \mathcal{P}_{k,1}}) \left( z_n^2 \mathbf{Attn}^{(t)}_{\mathbf{p} \to \mathcal{P}_{k,n}} + z_1^2 \mathbf{Attn}^{(t)}_{\mathbf{p} \to \mathcal{P}_{k,1}} - \max_{a \neq 1,n} z_a^2 \mathbf{Attn}^{(t)}_{\mathbf{p} \to \mathcal{P}_{k,a}} \right).$$

$$\text{(F.1)}$$

Therefore, by Lemma C.1, we have

$$\alpha^{(t)}_{\mathbf{p} \to v_{k,1}} \le \mathbb{E}\left[ \mathbf{1}\{k_X = k, \mathcal{E}_{k,1} \cap \mathbf{p} \in \mathcal{M}\} \mathbf{Attn}^{(t)}_{\mathbf{p} \to \mathcal{P}_{k,1}} \cdot \right.$$

$$\left(-z_1(1 - \mathbf{Attn}^{(t)}_{\mathbf{p}\to\mathcal{P}_{k,n}} - \mathbf{Attn}^{(t)}_{\mathbf{p}\to\mathcal{P}_{k,1}})\left(z_n^2\mathbf{Attn}^{(t)}_{\mathbf{p}\to\mathcal{P}_{k,n}} + z_1^2\mathbf{Attn}^{(t)}_{\mathbf{p}\to\mathcal{P}_{k,1}} - \max_{a\neq 1,n} z_a^2\mathbf{Attn}^{(t)}_{\mathbf{p}\to\mathcal{P}_{k,a}}\right)\right)\Bigg]$$

$$+ \mathbb{E}\left[\mathbf{1}\{k_X = k, \mathcal{E}^c_{k,1} \cap \mathbf{p} \in \mathcal{M}\}\mathbf{Attn}^{(t)}_{\mathbf{p}\to\mathcal{P}_{k,1}} \cdot \sum_{a\neq 1,n} z_1^2 z_a \left(\mathbf{Attn}^{(t)}_{\mathbf{p}\to\mathcal{P}_{k,a}}\right)^2\right]$$

$$\leq \mathbb{P}(\mathsf{M} \in \mathcal{E}_{k,1}) \cdot \left(-\left(\Omega(1) \cdot \Omega\left(\frac{1}{P^{2\times(\frac{1-\kappa_s}{2}-0.01)}}\right)\right)\right) + O(1) \cdot \mathbb{P}(\mathsf{M} \in \mathcal{E}^c_{k,1})$$

$$\leq -\Omega\left(\frac{1}{P^{2\times(\frac{1-\kappa_s}{2}-0.01)}}\right) = -\Omega\left(\frac{1}{P^{0.98-\kappa_s}}\right)$$

where the second inequality invokes Lemma F.1 and the last inequality comes from Lemma D.6. $\quad\square$

**Lemma F.5.** *At each iteration $t \leq T_1$, if Induction Hypothesis E.1 and Induction Hypothesis F.1 hold, then for any $m > 1$ with $m \neq n$, the following holds*

$$|\alpha^{(t)}_{\mathbf{p}\to v_{k,m}}| \leq O\left(\frac{\alpha^{(t)}_{\mathbf{p}\to v_{k,n}} - \alpha^{(t)}_{\mathbf{p}\to v_{k,1}}}{N}\right) = O\left(\frac{\alpha^{(t)}_{\mathbf{p}\to v_{k,n}} - \alpha^{(t)}_{\mathbf{p}\to v_{k,1}}}{P^{1-\kappa_s}}\right).$$

*Proof.* By Lemma C.1, for $m \neq n$, we have

$$\alpha^{(t)}_{\mathbf{p}\to v_{k,m}} \leq \mathbb{E}\left[\mathbf{1}\{k_X = k, \mathbf{p} \in \mathcal{M}\}\mathbf{Attn}^{(t)}_{\mathbf{p}\to\mathcal{P}_{k,m}} \cdot \left(\sum_{a\neq m,n} z_a^2 z_m \left(\mathbf{Attn}^{(t)}_{\mathbf{p}\to\mathcal{P}_{k,a}}\right)^2\right)\right] \quad \text{(F.2)}$$

$$-\alpha^{(t)}_{\mathbf{p}\to v_{k,m}} \leq \mathbb{E}\Bigg[\mathbf{1}\{k_X = k, \mathbf{p} \in \mathcal{M}\}\mathbf{Attn}^{(t)}_{\mathbf{p}\to\mathcal{P}_{k,m}} \cdot \left(z_m z_n^2 \left(1 - \mathbf{Attn}^{(t)}_{\mathbf{p}\to\mathcal{P}_{k,n}}\right)\mathbf{Attn}^{(t)}_{\mathbf{p}\to\mathcal{P}_{k,n}}\right.$$

$$\left.+ z_m^3 \left(1 - \mathbf{Attn}^{(t)}_{\mathbf{p}\to\mathcal{P}_{k,m}}\right)\mathbf{Attn}^{(t)}_{\mathbf{p}\to\mathcal{P}_{k,m}}\right)\Bigg] \quad \text{(F.3)}$$

For equation F.2, we have

$$\alpha^{(t)}_{\mathbf{p}\to v_{k,m}}$$

$$\leq \mathbb{E}\left[\mathbf{1}\{k_X = k, \mathcal{E}_{k,1} \cap \mathcal{E}_{k,n} \cap \mathbf{p} \in \mathcal{M}\}\mathbf{Attn}^{(t)}_{\mathbf{p}\to\mathcal{P}_{k,m}} \cdot \left(\sum_{a\neq m,n} z_a^2 z_m \left(\mathbf{Attn}^{(t)}_{\mathbf{p}\to\mathcal{P}_{k,a}}\right)^2\right)\right]$$

$$+ \mathbb{E}\left[\mathbf{1}\{k_X = k, (\mathcal{E}_{k,1} \cap \mathcal{E}_{k,n})^c \cap \mathbf{p} \in \mathcal{M}\}\mathbf{Attn}^{(t)}_{\mathbf{p}\to\mathcal{P}_{k,m}} \cdot \left(\sum_{a\neq m,n} z_a^2 z_m \left(\mathbf{Attn}^{(t)}_{\mathbf{p}\to\mathcal{P}_{k,a}}\right)^2\right)\right]$$

$$\leq \mathbb{E}\left[\mathbf{1}\{k_X = k, \mathcal{E}_{k,1} \cap \mathcal{E}_{k,n} \cap \mathbf{p} \in \mathcal{M}\}O\left(\frac{1 - \mathbf{Attn}^{(t)}_{\mathbf{p}\to\mathcal{P}_{k,1}} - \mathbf{Attn}^{(t)}_{\mathbf{p}\to\mathcal{P}_{k,n}}}{N}\right)\right.$$

$$\left. \cdot \left(z_1^2 z_m \left(\mathbf{Attn}^{(t)}_{\mathbf{p}\to\mathcal{P}_{k,1}}\right)^2 + O\left(\frac{1}{N}\right)\right)\right] + O(1) \cdot \mathbb{P}(\mathsf{M} \in (\mathcal{E}_{k,1} \cap \mathcal{E}_{k,n})^c)$$

$$\leq O\left(\frac{|\alpha^{(t)}_{\mathbf{p}\to v_{k,1}}|}{N}\right) + O(1) \cdot \mathbb{P}(\mathsf{M} \in (\mathcal{E}_{k,1} \cap \mathcal{E}_{k,n})^c)$$

$$\leq O\left(\frac{|\alpha^{(t)}_{\mathbf{p}\to v_{k,1}}|}{P^{1-\kappa_s}}\right)$$

where the second inequality is due to Lemma F.2, the last inequality follows from Lemma F.4 and Lemma D.6.

On the other hand, for equation F.3, we can use the similar argument by invoking Lemma F.2 and Lemma F.3, and thus obtain

$$-\alpha^{(t)}_{\mathbf{p}\to v_{k,m}} \leq O\left(\frac{\alpha^{(t)}_{\mathbf{p}\to v_{k,n}}}{P^{1-\kappa_s}}\right).$$

Putting them together, we have

$$|\alpha_{\mathbf{p} \to v_{k,m}}^{(t)}| \le O\Big( \frac{\alpha_{\mathbf{p} \to v_{k,n}}^{(t)} - \alpha_{\mathbf{p} \to v_{k,1}}^{(t)}}{P^{1-\kappa_s}} \Big).$$

$\square$

### F.1.3 Bounding the Gradient Updates for Positional Correlations

**Lemma F.6.** *For $n > 1$, if Induction Hypothesis E.1 and Induction Hypothesis F.1 hold at iteration $0 \le t \le T_1$, then for $\mathbf{q} \in \mathcal{P} \setminus \{\mathbf{p}\}$ and $a_{k,\mathbf{q}} = n$, we have $\beta_{k,\mathbf{p} \to \mathbf{q}}^{(t)} \ge 0$ and satisfies:*

$$\beta_{k,\mathbf{p} \to \mathbf{q}}^{(t)} = \Theta\Big( \frac{\alpha_{\mathbf{p} \to v_{k,n}}^{(t)}}{C_n} \Big).$$

*Furthermore, we have $|\beta_{k,\mathbf{p} \to \mathbf{p}}^{(t)}| = O\Big( \frac{\alpha_{\mathbf{p} \to v_{k,n}}^{(t)} - \alpha_{\mathbf{p} \to v_{k,1}}^{(t)}}{P} \Big)$.*

*Proof.* By Lemma C.2, for $\mathbf{q} \in \mathcal{P}_{k,n}$ with $\mathbf{q} \neq \mathbf{p}$, we have

$$\beta_{k,\mathbf{p} \to \mathbf{q}}^{(t)} =$$

$$\underbrace{\mathbb{E}\left[ \mathbf{1}\{k_X = k, \mathbf{p} \in \mathcal{M}, \mathbf{q} \in \mathcal{U}\}\mathbf{attn}_{\mathbf{p} \to \mathbf{q}}^{(t)} \cdot \Big( z_n^2 \Big( 1 - \mathbf{Attn}_{\mathbf{p} \to \mathcal{P}_{k,n}}^{(t)} \Big)^2 + \sum_{m \neq n} z_m^2 \Big( \mathbf{Attn}_{\mathbf{p} \to \mathcal{P}_{k,m}}^{(t)} \Big)^2 \Big) \right]}_{H_1}$$

$$+ \underbrace{\mathbb{E}\left[ \mathbf{1}\{k_X = k, \mathbf{p} \in \mathcal{M}, \mathbf{q} \in \mathcal{M}\}\mathbf{attn}_{\mathbf{p} \to \mathbf{q}}^{(t)} \cdot \Big( -z_n^2 \mathbf{Attn}_{\mathbf{p} \to \mathcal{P}_{k,n}}^{(t)} \Big( 1 - \mathbf{Attn}_{\mathbf{p} \to \mathcal{P}_{k,n}}^{(t)} \Big) \Big) \right]}_{H_2}$$

$$+ \underbrace{\mathbb{E}\left[ \mathbf{1}\{k_X = k, \mathbf{p} \in \mathcal{M}, \mathbf{q} \in \mathcal{M}\}\mathbf{attn}_{\mathbf{p} \to \mathbf{q}}^{(t)} \cdot \Big( \sum_{m \neq n} z_m^2 \Big( \mathbf{Attn}_{\mathbf{p} \to \mathcal{P}_{k,m}}^{(t)} \Big)^2 \Big) \right]}_{H_3}.$$

Firstly, for $H_1$, notice that

$$(C_n - 1)H_1 = \mathbb{E}\left[ \mathbf{1}\{k_X = k, \mathbf{p} \in \mathcal{M}\}\mathbf{Attn}_{\mathbf{p} \to \mathcal{P}_{k,n}}^{(t)} \cdot \Big( z_n^2 \Big( 1 - \mathbf{Attn}_{\mathbf{p} \to \mathcal{P}_{k,n}}^{(t)} \Big)^2 + \sum_{m \neq n} z_m^2 \Big( \mathbf{Attn}_{\mathbf{p} \to \mathcal{P}_{k,m}}^{(t)} \Big)^2 \Big) \right]$$

$$= \Theta(\alpha_{\mathbf{p} \to v_{k,n}}^{(t)}).$$

For $H_2$, since $\mathbf{p}, \mathbf{q} \in \mathcal{M}$, by Lemma F.1, we can upper bound $\mathbf{attn}_{\mathbf{p} \to \mathbf{q}}^{(t)}$ by $O\Big( \frac{1}{P} \Big)$, thus

$$-H_2 \le \mathbb{E}\left[ \mathbf{1}\{k_X = k, \mathbf{p} \in \mathcal{M}\}O\Big( \frac{1}{P} \Big) \cdot \Big( z_n^2 \mathbf{Attn}_{\mathbf{p} \to \mathcal{P}_{k,n}}^{(t)} \Big( 1 - \mathbf{Attn}_{\mathbf{p} \to \mathcal{P}_{k,n}}^{(t)} \Big) \Big) \right] \le O\Big( \frac{\alpha_{\mathbf{p} \to v_{k,n}}^{(t)}}{P} \Big).$$

Further notice that $H_3$ can be upper bounded by $O(H_1)$, putting it together, we have

$$\beta_{k,\mathbf{p} \to \mathbf{q}}^{(t)} = \Theta\Big( \frac{\alpha_{\mathbf{p} \to v_{k,n}}^{(t)}}{C_n} \Big).$$

Turn to $\beta_{k,\mathbf{p} \to \mathbf{p}}^{(t)}$, when $\mathbf{q} = \mathbf{p}$,

$$\beta_n^{(t)} = \underbrace{\mathbb{E}\left[ \mathbf{1}\{k_X = k, \mathbf{p} \in \mathcal{M}\}\mathbf{attn}_{\mathbf{p} \to \mathbf{p}}^{(t)} \cdot \Big( -z_n^2 \mathbf{Attn}_{\mathbf{p} \to \mathcal{P}_{k,n}}^{(t)} \Big( 1 - \mathbf{Attn}_{\mathbf{p} \to \mathcal{P}_{k,n}}^{(t)} \Big) \Big) \right]}_{J_2}$$

$$+ \mathbb{E}\left[\mathbf{1}\{k_X = k, \mathbf{p} \in \mathcal{M}\}\mathbf{attn}_{\mathbf{p} \to \mathbf{p}}^{(t)} \cdot \left(\underbrace{\sum_{m \neq n} z_m^2 \left(\mathbf{Attn}_{\mathbf{p} \to \mathcal{P}_{k,m}}^{(t)}\right)^2}_{}\right)\right].$$

$$\underbrace{\phantom{\mathbb{E}\left[\mathbf{1}\{k_X = k, \mathbf{p} \in \mathcal{M}\}\mathbf{attn}_{\mathbf{p} \to \mathbf{p}}^{(t)} \cdot \left(\sum_{m \neq n} z_m^2 \left(\mathbf{Attn}_{\mathbf{p} \to \mathcal{P}_{k,m}}^{(t)}\right)^2\right)\right]}}_{J_3}$$

We can bound $J_2$ in a similar way as $H_2$. Thus, we only focus on further bounding $J_3$:

$$J_3 \leq \mathbb{E}\left[\mathbf{1}\{k_X = k, \mathbf{p} \in \mathcal{M}\} O(\frac{1 - \mathbf{Attn}_{\mathbf{p} \to \mathcal{P}_{k,1}}^{(t)} - \mathbf{Attn}_{\mathbf{p} \to \mathcal{P}_{k,n}}^{(t)}}{P}) \cdot \left(\sum_{m \neq n} z_m^2 \left(\mathbf{Attn}_{\mathbf{p} \to \mathcal{P}_{k,m}}^{(t)}\right)^2\right)\right]$$

$$\leq O\left(\frac{|\alpha_{\mathbf{p} \to v_{k,1}}^{(t)}|}{P}\right).$$

where the first inequality holds by invoking Lemma F.1 and the last inequality follows similar arguments as analysis for equation F.2. $\square$

**Lemma F.7.** *For $n > 1$, if Induction Hypothesis E.1 and Induction Hypothesis F.1 hold at iteration $0 \leq t \leq T_1$, then for $\mathbf{q} \in \mathcal{P} \setminus \{\mathbf{p}\}$ and $a_{k,\mathbf{q}} = 1$, we have $\beta_{k,\mathbf{p} \to \mathbf{q}}^{(t)}$ satisfies:*

$$|\beta_{k,\mathbf{p} \to \mathbf{q}}^{(t)}| = O\left(\frac{|\alpha_{\mathbf{p} \to v_{k,n}}^{(t)} - \alpha_{\mathbf{p} \to v_{k,1}}^{(t)}|}{P}\right) + O\left(\frac{|\alpha_{\mathbf{p} \to v_{k,1}}^{(t)}|}{C_1}\right).$$

*Proof.* By Lemma C.2, for $\mathbf{q} \in \mathcal{P}_{k,1}$, we have

$$\beta_{k,\mathbf{p} \to \mathbf{q}}^{(t)} =$$

$$- \mathbb{E}\left[\mathbf{1}\{k_X = k, \mathbf{p} \in \mathcal{M}, \mathbf{q} \in \mathcal{U}\}\mathbf{attn}_{\mathbf{p} \to \mathbf{q}}^{(t)} \cdot \right.$$

$$\left.\left(z_1^2 \mathbf{Attn}_{\mathbf{p} \to \mathcal{P}_{k,1}}^{(t)}(1 - \mathbf{Attn}_{\mathbf{p} \to \mathcal{P}_{k,1}}^{(t)}) + z_n^2\left(1 - \mathbf{Attn}_{\mathbf{p} \to \mathcal{P}_{k,n}}^{(t)}\right)\mathbf{Attn}_{\mathbf{p} \to \mathcal{P}_{k,n}}^{(t)} - \sum_{a \neq 1, n} z_a^2 \left(\mathbf{Attn}_{\mathbf{p} \to \mathcal{P}_{k,a}}^{(t)}\right)^2\right)\right]$$

$$\text{(F.4)}$$

$$\underbrace{-\mathbb{E}\left[\mathbf{1}\{k_X = k, \mathbf{p} \in \mathcal{M}, \mathbf{q} \in \mathcal{M}\}\mathbf{attn}_{\mathbf{p} \to \mathbf{q}}^{(t)} \cdot \left(z_n^2\left(1 - \mathbf{Attn}_{\mathbf{p} \to \mathcal{P}_{k,n}}^{(t)}\right)\mathbf{Attn}_{\mathbf{p} \to \mathcal{P}_{k,n}}^{(t)}\right)\right]}_{G_2}$$

$$\underbrace{+\mathbb{E}\left[\mathbf{1}\{k_X = k, \mathbf{p} \in \mathcal{M}, \mathbf{q} \in \mathcal{M}\}\mathbf{attn}_{\mathbf{p} \to \mathbf{q}}^{(t)} \cdot \left(\sum_{a \neq n} z_a^2 \left(\mathbf{Attn}_{\mathbf{p} \to \mathcal{P}_{k,a}}^{(t)}\right)^2\right)\right]}_{G_3}.$$

For equation F.4 denoted as $G_1$, following the direct calculations, we have

$$-(C_1 - 1)G_1 = \Theta(\alpha_{\mathbf{p} \to v_{k,1}}^{(t)})$$

We can further bound $G_2$ and $G_3$ in a similar way as $H_2$ and $H_3$ in Lemma F.6 and thus obtain

$$-G_2 \leq O\left(\frac{\alpha_{\mathbf{p} \to v_{k,n}}^{(t)}}{P}\right),$$

$$G_3 \leq O\left(\frac{|\alpha_{\mathbf{p} \to v_{k,1}}^{(t)}|}{P}\right).$$

which completes the proof. $\square$

**Lemma F.8.** *For $n > 1$, if Induction Hypothesis E.1 and Induction Hypothesis F.1 hold at iteration $0 \leq t \leq T_1$, then for $\mathbf{q} \in \mathcal{P} \setminus \{\mathbf{p}\}$ and $n \neq a_{k,\mathbf{q}}$, $\beta_{k,\mathbf{p} \to \mathbf{q}}^{(t)}$ satisfies:*

$$|\beta_{k,\mathbf{p} \to \mathbf{q}}^{(t)}| = O\left(\frac{\alpha_{\mathbf{p} \to v_{k,n}}^{(t)} - \alpha_{\mathbf{p} \to v_{k,1}}^{(t)}}{P}\right).$$

*Proof.* By Lemma C.2, for $\mathbf{q} \in \mathcal{P}_{k,m}$, we have

$$\beta_{k,\mathbf{p}\to\mathbf{q}}^{(t)} =$$

$$-\mathbb{E}\left[\mathbf{1}\{k_X = k, \mathbf{p} \in \mathcal{M}, \mathbf{q} \in \mathcal{U}\}\mathbf{attn}_{\mathbf{p}\to\mathbf{q}}^{(t)}\cdot\right.$$

$$\left.\left(z_m^2\mathbf{Attn}_{\mathbf{p}\to\mathcal{P}_{k,m}}^{(t)}(1 - \mathbf{Attn}_{\mathbf{p}\to\mathcal{P}_{k,m}}^{(t)}) + z_n^2\left(1 - \mathbf{Attn}_{\mathbf{p}\to\mathcal{P}_{k,n}}^{(t)}\right)\mathbf{Attn}_{\mathbf{p}\to\mathcal{P}_{k,n}}^{(t)} - \sum_{a\neq n,m} z_a^2\left(\mathbf{Attn}_{\mathbf{p}\to\mathcal{P}_{k,a}}^{(t)}\right)^2\right)\right]$$

$$\text{(F.5)}$$

$$\underbrace{-\mathbb{E}\left[\mathbf{1}\{k_X = k, \mathbf{p} \in \mathcal{M}, \mathbf{q} \in \mathcal{M}\}\mathbf{attn}_{\mathbf{p}\to\mathbf{q}}^{(t)}\cdot\left(z_n^2\left(1 - \mathbf{Attn}_{\mathbf{p}\to\mathcal{P}_{k,n}}^{(t)}\right)\mathbf{Attn}_{\mathbf{p}\to\mathcal{P}_{k,n}}^{(t)}\right)\right]}_{I_2}$$

$$\underbrace{+\mathbb{E}\left[\mathbf{1}\{k_X = k, \mathbf{p} \in \mathcal{M}, \mathbf{q} \in \mathcal{M}\}\mathbf{attn}_{\mathbf{p}\to\mathbf{q}}^{(t)}\cdot\left(\sum_{a\neq n} z_a^2\left(\mathbf{Attn}_{\mathbf{p}\to\mathcal{P}_{k,a}}^{(t)}\right)^2\right)\right]}_{I_3}.$$

equation F.5 can be upper bounded by $O\left(\frac{|\alpha_{\mathbf{p}\to v_{k,m}}^{(t)}|}{C_m}\right) = O\left(\frac{|\alpha_{\mathbf{p}\to v_{k,1}}^{(t)} - \alpha_{\mathbf{p}\to v_{k,1}}^{(t)}|}{NC_m}\right) = O\left(\frac{|\alpha_{\mathbf{p}\to v_{k,1}}^{(t)} - \alpha_{\mathbf{p}\to v_{k,1}}^{(t)}|}{P}\right)$, where the first equality holds by invoking Lemma F.5. $I_2$ and $I_3$ can be bounded similarly as $G_2$ and $G_3$, which is omitted here. $\square$

### F.1.4 AT THE END OF PHASE I, STAGE 1

**Lemma F.9.** *For $n > 1$, if Induction Hypothesis E.1 and Induction Hypothesis F.1 hold for all $0 \leq t \leq T_1 = O\left(\frac{\log(P)P^{0.98-\kappa_s}}{\eta}\right)$, At iteration $t = T_1 + 1$, we have*

a. $\Phi_{\mathbf{p}\to v_{k,1}}^{(T_1+1)} \leq -\frac{1}{U}\left(\frac{\Delta}{2} - 0.01\right)\log(P)$;

b. $\mathbf{Attn}_{\mathbf{p}\to\mathcal{P}_{k,1}}^{(T_1+1)} = O\left(\frac{1}{P^{(1-\kappa_c)+\frac{L}{U}\left(\frac{\Delta}{2}-0.01\right)}}\right)$.

*Proof.* By comparing Lemma F.3 and Lemma F.4, we have $|\alpha_{\mathbf{p}\to v_{k,1}}^{(t)}| \gg \alpha_{\mathbf{p}\to v_{k,n}}^{(t)}$. Then the existence of $T_{1,k} = O\left(\frac{\log(P)P^{0.98-\kappa_s}}{\eta}\right)$ directly follows from Lemma F.4. $\square$

### F.2 PHASE I, STAGE 2

During stage 1, $\Phi_{\mathbf{p}\to v_{k,1}}^{(t)}$ significantly decreases to decouple the FP correlations with the global feature, resulting in a decrease in $\mathbf{Attn}_{\mathbf{p}\to\mathcal{P}_{k,1}}^{(t)}$, while other $\mathbf{Attn}_{\mathbf{p}\to\mathcal{P}_{k,n}}^{(t)}$ with $m > 1$ remain approximately at the order of $O\left(\frac{1}{P^{1-\kappa_s}}\right)$ ($\Theta\left(\frac{1}{P^{1-\kappa_s}}\right)$). By the end of phase I, $(\mathbf{Attn}_{\mathbf{p}\to\mathcal{P}_{k,1}}^{(t)})^2$ decreases to $O\left(\frac{1}{P^{1.96-2\kappa_s}}\right)$, leading to a decrease in $|\alpha_{\mathbf{p}\to v_{k,1}}^{(t)}|$ as it approaches towards $\alpha_{\mathbf{p}\to v_{k,n}}^{(t)}$. At this point, stage 2 begins. Shortly after entering this phase, the prior dominant role of the decrease of $\Phi_{\mathbf{p}\to v_{k,1}}^{(t)}$ in learning dynamics diminishes as $|\alpha_{\mathbf{p}\to v_{k,1}}^{(t)}|$ reaches the same order of magnitude as $\alpha_{\mathbf{p}\to v_{k,n}}^{(t)}$.

We define stage 2 of phase I as all iterations $T_1 < t \leq \widetilde{T}_1$, where

$$\widetilde{T}_1 \triangleq \max\left\{t > T_1 : \Phi_{\mathbf{p}\to v_{k,n}}^{(t)} - \Phi_{\mathbf{p}\to v_{k,1}}^{(t)} \leq \left(\frac{\Delta}{2L} + \frac{0.01}{L} + \frac{c_1^*(1-\kappa_s)}{U}\right)\log(P)\right\}.$$

for some small constant $c_1^* > 0$.

For computational convenience, we make the following assumptions for $\kappa_c$ and $\kappa_s$, which can be easily relaxed with the cost of additional calculations.

$$\frac{\Delta}{2}\left(\frac{1}{L} - \frac{1}{U}\right) + \frac{0.01}{L} + \frac{0.01}{U} \leq \frac{c_0^*(1-\kappa_s)}{U} \tag{F.6a}$$

$$(1 - \frac{c_1^* L}{U})(1 - \kappa_s) \le (1 - \kappa_c) + \frac{U}{L}(\frac{\Delta}{2} + 0.01) \tag{F.6b}$$

Here $c_0^*$ is some small. We state the following induction hypotheses, which will hold throughout this period:

**Induction Hypothesis F.2.** For each $T_1 < t \le \widetilde{T}_1$, $\mathbf{q} \in \mathcal{P} \setminus \{\mathbf{p}\}$, the following holds:

a. $\Phi_{\mathbf{p} \to v_{k,n}}^{(t)}$ is monotonically increasing, and $\Phi_{\mathbf{p} \to v_{k,n}}^{(t)} \in [0, \frac{c_0^* + c_1^*}{U} \log(P)]$;

b. $\Phi_{\mathbf{p} \to v_{k,1}}^{(t)}$ is monotonically decreasing and $\Phi_{\mathbf{p} \to v_{k,1}}^{(t)} \in [-\frac{1}{L}(\frac{\Delta}{2} + 0.01) \log(P), -\frac{1}{U}(\frac{\Delta}{2} - 0.01) \log(P)]$;

c. $|\Phi_{\mathbf{p} \to v_{k,m}}^{(t)}| = O\left(\frac{\Phi_{\mathbf{p} \to v_{k,n}}^{(t)} - \Phi_{\mathbf{p} \to v_{k,1}}^{(t)}}{P^{1-\kappa_s}}\right)$ for $m \ne 1, n$;

d. $\Upsilon_{k, \mathbf{p} \to \mathbf{q}}^{(t)} = O\left(\frac{\Phi_{\mathbf{p} \to v_{k,n}}^{(t)}}{C_n}\right)$ for $a_{k,\mathbf{q}} = n$, $|\Upsilon_{k, \mathbf{p} \to \mathbf{p}}^{(t)}| = O\left(\frac{\Phi_{\mathbf{p} \to v_{k,n}}^{(t)} - \Phi_{\mathbf{p} \to v_{k,1}}^{(t)}}{P}\right)$;

e. $|\Upsilon_{k, \mathbf{p} \to \mathbf{q}}^{(t)}| = O\left(\frac{|\Phi_{\mathbf{p} \to v_{k,1}}^{(t)}|}{C_1}\right) + O\left(\frac{\Phi_{\mathbf{p} \to v_{k,n}}^{(t)} - \Phi_{\mathbf{p} \to v_{k,1}}^{(t)}}{P}\right)$ for $a_{k,\mathbf{q}} = 1.$;

f. $|\Upsilon_{k, \mathbf{p} \to \mathbf{q}}^{(t)}| = O\left(\frac{\Phi_{\mathbf{p} \to v_{k,n}}^{(t)} - \Phi_{\mathbf{p} \to v_{k,1}}^{(t)}}{P}\right)$ for $a_{k,\mathbf{q}} \ne 1, n$.

### F.2.1 PROPERTY OF ATTENTION SCORES

We first single out several properties of attention scores that will be used for the proof of Induction Hypothesis F.2.

**Lemma F.10.** *if Induction Hypothesis E.1 and Induction Hypothesis F.2 hold at iteration* $T_1 + 1 \le t \le \widetilde{T}_1$, *then the following holds*

1. $1 - \mathbf{Attn}_{\mathbf{p} \to \mathcal{P}_{k,n}}^{(t)} - \mathbf{Attn}_{\mathbf{p} \to \mathcal{P}_{k,1}}^{(t)} \ge \Omega(1)$;

2. *if* $\mathsf{M} \in \mathcal{E}_{k,n}$, $\mathbf{Attn}_{\mathbf{p} \to \mathcal{P}_{k,n}}^{(t)} \in \left[\Omega\left(\frac{1}{P^{1-\kappa_s}}\right), O\left(\frac{1}{P^{(1-c_1^* - c_0^*)(1-\kappa_s)}}\right)\right]$ ;

3. *Moreover,* $\mathbf{Attn}_{\mathbf{p} \to \mathcal{P}_{k,1}}^{(t)} = O\left(\frac{1}{P^{(1-\kappa_c) + \frac{L}{U}(\frac{\Delta}{2} - 0.01)}}\right)$; *if* $\mathsf{M} \in \mathcal{E}_{k,1}$, *we have* $\mathbf{Attn}_{\mathbf{p} \to \mathcal{P}_{k,1}}^{(t)} = \Omega\left(\frac{1}{P^{(1-\kappa_c) + \frac{U}{L}(\frac{\Delta}{2} + 0.01)}}\right)$;

4. *for* $\mathbf{q} \in \mathcal{M} \cap (\mathcal{P}_{k,n} \cup \mathcal{P}_{k,1})$, $\mathbf{attn}_{\mathbf{p} \to \mathbf{q}}^{(t)} = O\left(\frac{1 - \mathbf{Attn}_{\mathbf{p} \to \mathcal{P}_{k,n}}^{(t)} - \mathbf{Attn}_{\mathbf{p} \to \mathcal{P}_{k,1}}^{(t)}}{P}\right)$.

**Lemma F.11.** *if Induction Hypothesis E.1 and Induction Hypothesis F.2 hold at iteration* $T_1 + 1 \le t \le \widetilde{T}_1$, *then for* $m \ne n$, *the following holds:*

1. *for any* $\mathbf{q} \in \mathcal{P}_{k,m}$, $\mathbf{attn}_{\mathbf{p} \to \mathbf{q}}^{(t)} \le O\left(\frac{1 - \mathbf{Attn}_{\mathbf{p} \to \mathcal{P}_{k,n}}^{(t)} - \mathbf{Attn}_{\mathbf{p} \to \mathcal{P}_{k,1}}^{(t)}}{P}\right)$;

2. *Moreover,* $\mathbf{Attn}_{\mathbf{p} \to \mathcal{P}_{k,m}}^{(t)} \le O\left(\frac{1 - \mathbf{Attn}_{\mathbf{p} \to \mathcal{P}_{k,1}}^{(t)} - \mathbf{Attn}_{\mathbf{p} \to \mathcal{P}_{k,n}}^{(t)}}{N}\right)$.

### F.2.2 BOUNDING THE GRADIENT UPDATES OF FP CORRELATIONS

**Lemma F.12.** *For* $n > 1$, *if Induction Hypothesis E.1 and Induction Hypothesis F.2 hold at iteration* $T_1 + 1 \le t \le \widetilde{T}_1$, *then* $\alpha_{\mathbf{p} \to v_{k,n}}^{(t)} \ge 0$ *and satisfies:*

$$\alpha_{\mathbf{p} \to v_{k,n}}^{(t)} = \Omega\left(\frac{1}{P^{1-\kappa_s}}\right).$$

*Proof.* By Lemma C.2, we have

$$\alpha_{\mathbf{p}\to v_{k,n}}^{(t)}$$

$$= \mathbb{E}\left[\mathbf{1}\{k_X = k, \mathbf{p} \in \mathcal{M}\}\mathbf{Attn}_{\mathbf{p}\to\mathcal{P}_{k,n}}^{(t)} \cdot \left(z_n^3\left(1 - \mathbf{Attn}_{\mathbf{p}\to\mathcal{P}_{k,n}}^{(t)}\right)^2 + \sum_{m\neq n} z_m^2 z_n \left(\mathbf{Attn}_{\mathbf{p}\to\mathcal{P}_{k,m}}^{(t)}\right)^2\right)\right]$$

$$= \mathbb{E}\left[\mathbf{1}\{k_X = k, \mathcal{E}_{k,n} \cap \mathbf{p} \in \mathcal{M}\}\mathbf{Attn}_{\mathbf{p}\to\mathcal{P}_{k,n}}^{(t)} \cdot \left(z_n^3\left(1 - \mathbf{Attn}_{\mathbf{p}\to\mathcal{P}_{k,n}}^{(t)}\right)^2 + \sum_{m\neq n} z_m^2 z_n \left(\mathbf{Attn}_{\mathbf{p}\to\mathcal{P}_{k,m}}^{(t)}\right)^2\right)\right]$$

$$+ \mathbb{E}\left[\mathbf{1}\{k_X = k, \mathcal{E}_{k,n}^c \cap \mathbf{p} \in \mathcal{M}\}\mathbf{Attn}_{\mathbf{p}\to\mathcal{P}_{k,n}}^{(t)} \cdot \left(z_n^3\left(1 - \mathbf{Attn}_{\mathbf{p}\to\mathcal{P}_{k,n}}^{(t)}\right)^2 + \sum_{m\neq n} z_m^2 z_n \left(\mathbf{Attn}_{\mathbf{p}\to\mathcal{P}_{k,m}}^{(t)}\right)^2\right)\right]$$

$$\gtrsim \mathbb{P}(\mathsf{M} \in \mathcal{E}_{k,n})$$

$$\cdot \mathbb{E}\left[\mathbf{1}\{k_X = k, \mathbf{p} \in \mathcal{M}\}\mathbf{Attn}_{\mathbf{p}\to\mathcal{P}_{k,n}}^{(t)} \cdot \left(z_n^3\left(1 - \mathbf{Attn}_{\mathbf{p}\to\mathcal{P}_{k,n}}^{(t)}\right)^2 + \sum_{m\neq n} z_m^2 z_n \left(\mathbf{Attn}_{\mathbf{p}\to\mathcal{P}_{k,m}}^{(t)}\right)^2\right)\Big|\mathcal{E}_{k,n}\right]$$

$$\geq \Omega\left(\frac{C_n}{P}\right)$$

where the last inequality invokes Lemma F.10.

$\square$

**Lemma F.13.** *For $n > 1$, if Induction Hypothesis E.1 and Induction Hypothesis F.2 hold at iteration $T_1 + 1 \leq t \leq \widetilde{T}_1$, then $\alpha_{\mathbf{p}\to v_{k,1}}^{(t)} < 0$ and satisfies*

$$|\alpha_{\mathbf{p}\to v_{k,1}}^{(t)}| \geq \Omega\left(\frac{1}{P^{2(1-\kappa_c)+\frac{U}{L}(\Delta+0.02)}}\right).$$

*Proof.* Following equation F.1, we have

$$- z_1 z_n^2\left(1 - \mathbf{Attn}_{\mathbf{p}\to\mathcal{P}_{k,n}}^{(t)}\right)\mathbf{Attn}_{\mathbf{p}\to\mathcal{P}_{k,n}}^{(t)} - z_1^3\left(1 - \mathbf{Attn}_{\mathbf{p}\to\mathcal{P}_{k,1}}^{(t)}\right)\mathbf{Attn}_{\mathbf{p}\to\mathcal{P}_{k,1}}^{(t)} + \sum_{a\neq 1,n} z_a^2 z_1\left(\mathbf{Attn}_{\mathbf{p}\to\mathcal{P}_{k,a}}^{(t)}\right)^2$$

$$\leq -z_1(1 - \mathbf{Attn}_{\mathbf{p}\to\mathcal{P}_{k,n}}^{(t)} - \mathbf{Attn}_{\mathbf{p}\to\mathcal{P}_{k,1}}^{(t)})\left(z_n^2\mathbf{Attn}_{\mathbf{p}\to\mathcal{P}_{k,n}}^{(t)} + z_1^2\mathbf{Attn}_{\mathbf{p}\to\mathcal{P}_{k,1}}^{(t)} - \max_{a\neq 1,n} z_a^2\mathbf{Attn}_{\mathbf{p}\to\mathcal{P}_{k,a}}^{(t)}\right)$$

Therefore, by Lemma C.1, we obtain

$$\alpha_{\mathbf{p}\to v_{k,1}}^{(t)} \leq \mathbb{E}\left[\mathbf{1}\{k_X = k, \mathcal{E}_{k,1} \cap \mathbf{p} \in \mathcal{M}\}\mathbf{Attn}_{\mathbf{p}\to\mathcal{P}_{k,1}}^{(t)} \cdot\right.$$

$$\left(-z_1(1 - \mathbf{Attn}_{\mathbf{p}\to\mathcal{P}_{k,n}}^{(t)} - \mathbf{Attn}_{\mathbf{p}\to\mathcal{P}_{k,1}}^{(t)})\right.$$

$$\left.\left.\cdot\left(z_n^2\mathbf{Attn}_{\mathbf{p}\to\mathcal{P}_{k,n}}^{(t)} + z_1^2\mathbf{Attn}_{\mathbf{p}\to\mathcal{P}_{k,1}}^{(t)} - \max_{a\neq 1,n} z_a^2\mathbf{Attn}_{\mathbf{p}\to\mathcal{P}_{k,a}}^{(t)}\right)\right)\right]$$

$$+ \mathbb{E}\left[\mathbf{1}\{k_X = k, \mathcal{E}_{k,1}^c \cap \mathbf{p} \in \mathcal{M}\}\mathbf{Attn}_{\mathbf{p}\to\mathcal{P}_{k,1}}^{(t)} \cdot \sum_{a\neq 1,n} z_1^2 z_a\left(\mathbf{Attn}_{\mathbf{p}\to\mathcal{P}_{k,a}}^{(t)}\right)^2\right]$$

$$\leq \mathbb{P}(\mathsf{M} \in \mathcal{E}_{k,1}) \cdot \left(-\Omega(1) \cdot \Omega\left(\frac{1}{P^{2(1-\kappa_c)+\frac{2U}{L}(\frac{\Delta}{2}+0.01)}}\right)\right) + O(1) \cdot \mathbb{P}(\mathsf{M} \in \mathcal{E}_{k,1}^c)$$

$$\leq -\Omega\left(\frac{1}{P^{2(1-\kappa_c)+\frac{U}{L}(\Delta+0.02)}}\right)$$

where the second inequality invokes Lemma F.10 and the last inequality comes from Lemma D.6. The upper bound can be obtained by using similar arguments and invoking the upper bound for $\mathbf{Attn}_{\mathbf{p}\to\mathcal{P}_{k,1}}^{(t)}$ in Lemma F.10.

$\square$

**Lemma F.14.** *For $n > 1$, if Induction Hypothesis E.1 and Induction Hypothesis F.2 hold at iteration $T_1 + 1 \le t \le \widetilde{T}_1$, then for any $m > 1$ with $m \ne n$, the following holds*

$$|\alpha^{(t)}_{\mathbf{p} \to v_{k,m}}| \le O\Big(\frac{\alpha^{(t)}_{\mathbf{p} \to v_{k,n}} - \alpha^{(t)}_{\mathbf{p} \to v_{k,1}}}{P^{1-\kappa_s}}\Big).$$

The proof is similar to Lemma F.5, and thus omitted here.

### F.2.3 BOUNDING THE GRADIENT UPDATES OF POSITIONAL CORRELATIONS

We then summarize the properties for gradient updates of positional correlations, which utilize the identical calculations as in Section F.1.3.

**Lemma F.15.** *For $n > 1$, if Induction Hypothesis E.1 and Induction Hypothesis F.2 hold at iteration $T_1 + 1 \le t \le \widetilde{T}_1$, then*

    *a. if $a_{k,\mathbf{q}} = n$ and $\mathbf{q} \ne \mathbf{p}$, $\beta^{(t)}_{k,\mathbf{p} \to \mathbf{q}} \ge 0$; $\beta^{(t)}_{k,\mathbf{p} \to \mathbf{q}} = \Theta\Big(\frac{\alpha^{(t)}_{\mathbf{p} \to v_{k,n}}}{C_n}\Big)$ and $|\beta^{(t)}_n| = O\big(\frac{\alpha^{(t)}_{\mathbf{p} \to v_{k,n}} - \alpha^{(t)}_{\mathbf{p} \to v_{k,1}}}{P}\big)$.*

    *b. if $a_{k,\mathbf{q}} = 1$, $|\beta^{(t)}_{k,\mathbf{p} \to \mathbf{q}}| = O\Big(\frac{\alpha^{(t)}_{\mathbf{p} \to v_{k,n}} - \alpha^{(t)}_{\mathbf{p} \to v_{k,1}}}{P}\Big) + O\Big(\frac{|\alpha^{(t)}_{\mathbf{p} \to v_{k,1}}|}{C_1}\Big)$.*

    *c. if $a_{k,\mathbf{q}} = m$ and $m \ne 1, n$, $|\beta^{(t)}_{k,\mathbf{p} \to \mathbf{q}}| = O\Big(\frac{\alpha^{(t)}_{\mathbf{p} \to v_{k,n}} - \alpha^{(t)}_{\mathbf{p} \to v_{k,1}}}{P}\Big)$.*

### F.2.4 END OF PHASE I, STAGE 2

**Lemma F.16.** *Induction Hypothesis F.2 holds for all iteration $T_1 + 1 \le t \le \widetilde{T}_1 = T_1 + O\Big(\frac{\log(P)P^{1-\kappa_s}}{\eta}\Big)$, and at iteration $t = \widetilde{T}_1 + 1$, we have*

    *a. $\Phi^{(\widetilde{T}_1+1)}_{\mathbf{p} \to v_{k,n}} \ge \frac{c_1^*(1-\kappa_s)\log(P)}{U}$;*

    *b. $\Phi^{(\widetilde{T}_1+1)}_{\mathbf{p} \to v_{k,1}} \ge -(\frac{\Delta}{2L} + \frac{0.01}{L})\log(P)$.*

*Proof.* The existence of $\widetilde{T}_1 = T_1 + O\Big(\frac{\log(P)P^{1-\kappa_s}}{\eta}\Big)$ directly follows from Lemma F.12 and Lemma F.13. Moreover, since $\alpha^{(t)}_{\mathbf{p} \to v_{k,1}} < 0$, then

$$\Phi^{(\widetilde{T}_1+1)}_{\mathbf{p} \to v_{k,n}} \le \Big(\frac{\Delta}{2L} + \frac{0.01}{L} + \frac{c_1^*(1-\kappa_s)}{U}\Big)\log(P) - \frac{1}{U}\Big(\frac{\Delta}{2} - 0.01\Big) \le \frac{(c_0^* + c_1^*)(1-\kappa_s)}{U}\log(P)$$

where the last inequality invokes equation F.6a. Now suppose $\Phi^{(\widetilde{T}_1+1)}_{\mathbf{p} \to v_{k,n}} < \frac{c_1^*(1-\kappa_s)\log(P)}{U}$, then $\Phi^{(\widetilde{T}_1+1)}_{\mathbf{p} \to v_{k,1}} < -(\frac{\Delta}{2L} + \frac{0.01}{L})\log(P)$. Denote the first time that $\Phi^{(t)}_{\mathbf{p} \to v_{k,1}}$ reaches $-(\frac{\Delta}{2L} + \frac{0.001}{L})\log(P)$ as $\widetilde{T}$. Note that $\widetilde{T} < \widetilde{T}_1$ since $\alpha^{(t)}_{\mathbf{p} \to v_{k,1}}$, the change of $\Phi^{(t)}_{\mathbf{p} \to v_{k,1}}$, satisfies $|\alpha^{(t)}_{\mathbf{p} \to v_{k,1}}| \ll \log(P)$. Then for $t \ge \widetilde{T}$, the following holds:

    1. $\mathbf{Attn}^{(t)}_{\mathbf{p} \to \mathcal{P}_{k,n}} \ge \Omega\Big(\frac{1}{P^{1-\kappa_s}}\Big)$;

    2. $\mathbf{Attn}^{(t)}_{\mathbf{p} \to \mathcal{P}_{k,1}} \le O\Big(\frac{1}{P^{\frac{1-\kappa_s}{2}+0.001}}\Big)$.

Therefore,

$$|\alpha^{(t)}_{\mathbf{p} \to v_{k,1}}| \le \mathbb{E}\Big[\mathbf{1}\{k_X = k, \mathcal{E}_{k,1} \cap \mathbf{p} \in \mathcal{M}\}\mathbf{Attn}^{(t)}_{\mathbf{p} \to \mathcal{P}_{k,1}} \cdot$$
$$z_1\Big(z_n^2\mathbf{Attn}^{(t)}_{\mathbf{p} \to \mathcal{P}_{k,n}}(1 - \mathbf{Attn}^{(t)}_{\mathbf{p} \to \mathcal{P}_{k,n}}) + z_1^2\mathbf{Attn}^{(t)}_{\mathbf{p} \to \mathcal{P}_{k,1}}(1 - \mathbf{Attn}^{(t)}_{\mathbf{p} \to \mathcal{P}_{k,1}})\Big)\Big]$$

$$+ \mathbb{E}\left[\mathbf{1}\{k_X = k, \mathcal{E}_{k,1}^c \cap \mathbf{p} \in \mathcal{M}\}\mathbf{Attn}_{\mathbf{p}\to\mathcal{P}_{k,1}}^{(t)} \cdot \sum_{a\neq 1,n} z_1^2 z_a \left(\mathbf{Attn}_{\mathbf{p}\to\mathcal{P}_{k,a}}^{(t)}\right)^2\right]$$

$$\leq O(\frac{\alpha_{\mathbf{p}\to v_{k,1}}^{(t)}}{P^{\frac{1-\kappa_s}{2}+0.001}}) + \mathbb{P}(\mathsf{M}\in\mathcal{E}_{k,1}) \cdot \left(O(1)\cdot O\left(\frac{1}{P^{\frac{1-\kappa_s}{2}+0.001}}\right)\right) + O(1)\cdot\mathbb{P}(\mathsf{M}\in\mathcal{E}_{k,1}^c)$$

$$\leq O\left(\frac{\alpha_{\mathbf{p}\to v_{k,1}}^{(t)}}{P^{\frac{1-\kappa_s}{2}+0.001}}\right) + O\left(\frac{1}{P^{(1-\kappa_s)+0.002}}\right).$$

Lemma F.12 still holds, and thus

$$|\alpha_{\mathbf{p}\to v_{k,1}}^{(t)}| \leq O\left(\frac{\alpha_{\mathbf{p}\to v_{k,n}}^{(t)}}{P^{0.002}}\right).$$

Since $|\Phi_{\mathbf{p}\to v_{k,1}}^{(\widetilde{T}_1+1)} - \Phi_{\mathbf{p}\to v_{k,1}}^{(\widetilde{T})}| \geq \Omega\left(\log(P)\right)$, we have

$$\Phi_{\mathbf{p}\to v_{k,n}}^{(\widetilde{T}_1+1)} \geq |\Phi_{\mathbf{p}\to v_{k,1}}^{(\widetilde{T}_1+1)} - \Phi_{\mathbf{p}\to v_{k,1}}^{(\widetilde{T})}| \cdot \Omega(P^{0.002}) + \Phi_{\mathbf{p}\to v_{k,n}}^{(\widetilde{T})} \gg \Omega(P^{0.002}\log(P)),$$

which contradicts the assumption that $\Phi_{\mathbf{p}\to v_{k,n}}^{(\widetilde{T}_1+1)} < \frac{c_1^*(1-\kappa_s)\log(P)}{U}$. □

## F.3 PHASE II, STAGE 1

For $n > 1$, we define stage 1 of phase II as all iterations $\widetilde{T}_1 + 1 \leq t \leq T_2$, where

$$T_2 \triangleq \max\left\{t : \Phi_{\mathbf{p}\to v_{k,n}}^{(t)} \leq \frac{(1-\kappa_s)}{L}\log(P)\right\}.$$

We state the following induction hypotheses, which will hold throughout this stage:

**Induction Hypothesis F.3.** For each $\widetilde{T}_1 + 1 \leq t \leq T_2$, $\mathbf{q} \in \mathcal{P} \setminus \{\mathbf{p}\}$, the following holds:

a. $\Phi_{\mathbf{p}\to v_{k,n}}^{(t)}$ is monotonically increasing, and $\Phi_{\mathbf{p}\to v_{k,n}}^{(t)} \in \left[\frac{c_1^*(1-\kappa_s)}{U}\log(P), \frac{(1-\kappa_s)}{L}\log(P)\right]$;

b. $\Phi_{\mathbf{p}\to v_{k,1}}^{(t)}$ is monotonically decreasing and

$$\Phi_{\mathbf{p}\to v_{k,1}}^{(t)} \in \left[-\frac{1}{L}\left(\frac{\Delta}{2} + 0.01\right)\log(P) - o(1), -\frac{1}{U}\left(\frac{\Delta}{2} - 0.01\right)\log(P)\right];$$

c. $|\Phi_{\mathbf{p}\to v_{k,m}}^{(t)}| = O\left(\frac{\Phi_{\mathbf{p}\to v_{k,n}}^{(t)} - \Phi_{\mathbf{p}\to v_{k,1}}^{(t)}}{P^{1-\kappa_s}}\right)$ for $m \neq 1, n$;

d. $\Upsilon_{k,\mathbf{p}\to\mathbf{q}}^{(t)} = O\left(\frac{\Phi_{\mathbf{p}\to v_{k,n}}^{(t)}}{C_n}\right)$ for $a_{k,\mathbf{q}} = n$, $|\Upsilon_{k,\mathbf{p}\to\mathbf{p}}^{(t)}| = O\left(\frac{\Phi_{\mathbf{p}\to v_{k,n}}^{(t)} - \Phi_{\mathbf{p}\to v_{k,1}}^{(t)}}{P}\right)$;

e. $|\Upsilon_{k,\mathbf{p}\to\mathbf{q}}^{(t)}| = O\left(\frac{|\Phi_{\mathbf{p}\to v_{k,1}}^{(t)}|}{C_1}\right) + O\left(\frac{\Phi_{\mathbf{p}\to v_{k,n}}^{(t)} - \Phi_{\mathbf{p}\to v_{k,1}}^{(t)}}{P}\right)$ for $a_{k,\mathbf{q}} = 1.$;

f. $|\Upsilon_{k,\mathbf{p}\to\mathbf{q}}^{(t)}| = O\left(\frac{\Phi_{\mathbf{p}\to v_{k,n}}^{(t)} - \Phi_{\mathbf{p}\to v_{k,1}}^{(t)}}{P}\right)$ for $a_{k,\mathbf{q}} \neq 1, n.$

### F.3.1 PROPERTY OF ATTENTION SCORES

We first single out several properties of attention scores that will be used for the proof of Induction Hypothesis F.3.

**Lemma F.17.** *if Induction Hypothesis E.1 and Induction Hypothesis F.3 hold at iteration $\widetilde{T}_1 + 1 \leq t \leq T_2$, then the following holds*

*1. if $\mathsf{M} \in \mathcal{E}_{k,n}$, $\mathbf{Attn}_{\mathbf{p}\to\mathcal{P}_{k,n}}^{(t)} \geq \Omega\left(\frac{1}{P^{(1-\frac{c_1^*L}{U})(1-\kappa_s)}}\right)$. Moreover, if $\mathbf{Attn}_{\mathbf{p}\to\mathcal{P}_{k,n}}^{(t)}$ does not reach the constant level, $1 - \mathbf{Attn}_{\mathbf{p}\to\mathcal{P}_{k,n}}^{(t)} = \Omega(1)$; otherwise, $1 - \mathbf{Attn}_{\mathbf{p}\to\mathcal{P}_{k,n}}^{(t)} = \Omega\left(\frac{1}{P^{(\frac{U}{L}-1)(1-\kappa_s)}}\right).$*

2. $\mathbf{Attn}^{(t)}_{\mathbf{p}\to\mathcal{P}_{k,1}} = O\left(\frac{1-\mathbf{Attn}^{(t)}_{\mathbf{p}\to\mathcal{P}_{k,n}}}{P^{(1-\kappa_c)+\frac{L}{U}(\frac{\Delta}{2}-0.01)}}\right);$ if $\mathsf{M} \in \mathcal{E}_{k,1}$, we have $\mathbf{Attn}^{(t)}_{\mathbf{p}\to\mathcal{P}_{k,1}} = \Omega\left(\frac{1}{P^{(1-\kappa_c)+\frac{U}{L}(\frac{\Delta}{2}+0.01)}}\right);$

3. for $\mathbf{q} \in \mathcal{M} \cap (\mathcal{P}_{k,n} \cup \mathcal{P}_{k,1})$, $\mathbf{attn}^{(t)}_{\mathbf{p}\to\mathbf{q}} = O\left(\frac{1-\mathbf{Attn}^{(t)}_{\mathbf{p}\to\mathcal{P}_{k,n}}}{P}\right)$

**Lemma F.18.** *if Induction Hypothesis E.1 and Induction Hypothesis F.3 hold at iteration $\widetilde{T}_1 + 1 \leq t \leq T_2$, then for $m \neq n$, the following holds:*

1. *for any $\mathbf{q} \in \mathcal{P}_{k,m}$, $\mathbf{attn}^{(t)}_{\mathbf{p}\to\mathbf{q}} \leq O\left(\frac{1-\mathbf{Attn}^{(t)}_{\mathbf{p}\to\mathcal{P}_{k,n}}}{P}\right).$*

2. *Moreover, $\mathbf{Attn}^{(t)}_{\mathbf{p}\to\mathcal{P}_{k,n}} \leq O\left(\frac{1-\mathbf{Attn}^{(t)}_{\mathbf{p}\to\mathcal{P}_{k,1}}-\mathbf{Attn}^{(t)}_{\mathbf{p}\to\mathcal{P}_{k,n}}}{N}\right).$*

### F.3.2 BOUNDING THE GRADIENT UPDATES OF FP CORRELATIONS

**Lemma F.19.** *if Induction Hypothesis E.1 and Induction Hypothesis F.3 hold at iteration $\widetilde{T}_1 + 1 \leq t \leq T_2$, then $\alpha^{(t)}_{\mathbf{p}\to v_{k,n}} \geq 0$ and satisfies:*

$$\alpha^{(t)}_{\mathbf{p}\to v_{k,n}} \geq \min\left\{\Omega\left(\frac{1}{P^{(1-\frac{c_1^* L}{U})(1-\kappa_s)}}\right), \Omega\left(\frac{1}{P^{2(\frac{U}{L}-1)(1-\kappa_s)}}\right)\right\}.$$

*Proof.* By Lemma C.2, we have

$\alpha^{(t)}_{\mathbf{p}\to v_{k,n}}$

$= \mathbb{E}\left[\mathbf{1}\{k_X = k, \mathbf{p} \in \mathcal{M}\}\mathbf{Attn}^{(t)}_{\mathbf{p}\to\mathcal{P}_{k,n}} \cdot \left(z_n^3\left(1 - \mathbf{Attn}^{(t)}_{\mathbf{p}\to\mathcal{P}_{k,n}}\right)^2 + \sum_{m\neq n} z_m^2 z_n \left(\mathbf{Attn}^{(t)}_{\mathbf{p}\to\mathcal{P}_{k,m}}\right)^2\right)\right]$

$= \mathbb{E}\left[\mathbf{1}\{k_X = k, \mathcal{E}_{k,n} \cap \mathbf{p} \in \mathcal{M}\}\mathbf{Attn}^{(t)}_{\mathbf{p}\to\mathcal{P}_{k,n}} \cdot \left(z_n^3\left(1 - \mathbf{Attn}^{(t)}_{\mathbf{p}\to\mathcal{P}_{k,n}}\right)^2 + \sum_{m\neq n} z_m^2 z_n \left(\mathbf{Attn}^{(t)}_{\mathbf{p}\to\mathcal{P}_{k,m}}\right)^2\right)\right]$

$+ \mathbb{E}\left[\mathbf{1}\{k_X = k, \mathcal{E}_{k,n}^c \cap \mathbf{p} \in \mathcal{M}\}\mathbf{Attn}^{(t)}_{\mathbf{p}\to\mathcal{P}_{k,n}} \cdot \left(z_n^3\left(1 - \mathbf{Attn}^{(t)}_{\mathbf{p}\to\mathcal{P}_{k,n}}\right)^2 + \sum_{m\neq n} z_m^2 z_n \left(\mathbf{Attn}^{(t)}_{\mathbf{p}\to\mathcal{P}_{k,m}}\right)^2\right)\right]$

$\gtrsim \mathbb{P}(\mathsf{M} \in \mathcal{E}_{k,n})\cdot$

$\mathbb{E}\left[\mathbf{1}\{k_X = k, \mathbf{p} \in \mathcal{M}\}\mathbf{Attn}^{(t)}_{\mathbf{p}\to\mathcal{P}_{k,n}} \cdot \left(z_n^3\left(1 - \mathbf{Attn}^{(t)}_{\mathbf{p}\to\mathcal{P}_{k,n}}\right)^2 + \sum_{m\neq n} z_m^2 z_n \left(\mathbf{Attn}^{(t)}_{\mathbf{p}\to\mathcal{P}_{k,m}}\right)^2\right)\Big|\mathcal{E}_{k,n}\right]$

$+ O(1) \cdot \mathbb{P}(\mathsf{M} \in \mathcal{E}_{k,n}^c)$

$\gtrsim \min\left\{\Omega\left(\frac{1}{P^{(1-\frac{c_1^* L}{U})(1-\kappa_s)}}\right), \Omega\left(\frac{1}{P^{2(\frac{U}{L}-1)(1-\kappa_s)}}\right)\right\}$

where the last inequality invokes Lemma F.17 by observing that for $\mathsf{M} \in \mathcal{E}_{k,n}$,

$$\mathbf{Attn}^{(t)}_{\mathbf{p}\to\mathcal{P}_{k,n}}(1 - \mathbf{Attn}^{(t)}_{\mathbf{p}\to\mathcal{P}_{k,n}})^2 \geq \min\left\{\Omega\left(\frac{1}{P^{(1-\frac{c_1^* L}{U})(1-\kappa_s)}}\right)\cdot\Omega(1), \Omega(1)\cdot\Omega\left(\frac{1}{P^{2\times(\frac{U}{L}-1)(1-\kappa_s)}}\right)\right\}.$$

$\square$

**Lemma F.20.** *For $n > 1$, if Induction Hypothesis E.1 and Induction Hypothesis F.3 hold at iteration $\widetilde{T}_1 + 1 \leq t \leq T_2$, then $\alpha^{(t)}_{\mathbf{p}\to v_{k,1}} < 0$ and satisfies*

$$|\alpha^{(t)}_{\mathbf{p}\to v_{k,m}}| \geq \min\left\{\Omega\left(\frac{1}{P^{(1-\frac{c_1^* L}{U})(1-\kappa_s)}}\right), \Omega\left(\frac{1}{P^{(\frac{U}{L}-1)(1-\kappa_s)}}\right)\right\}\cdot\Omega\left(\frac{1}{P^{(1-\kappa_c)+\frac{L}{U}(\frac{\Delta}{2}-0.01)}}\right),$$

$$|\alpha^{(t)}_{\mathbf{p} \to v_{k,m}}| \le \max\left\{ O\left(\frac{\alpha^{(t)}_{\mathbf{p} \to v_{k,n}}}{P^{(1-\kappa_c)+\frac{L}{U}(\Delta/2-0.01)}}\right), O\left(\frac{\alpha^{(t)}_{\mathbf{p} \to v_{k,n}}}{P^{2(1-\kappa_c)+\frac{L}{U}(\Delta-0.02)-(1-\frac{c_1^*L}{U})(1-\kappa_s)}}\right)\right\}.$$

*Proof.* Following equation F.1, we have

$$-z_1 z_n^2 \left(1 - \mathbf{Attn}^{(t)}_{\mathbf{p} \to \mathcal{P}_{k,n}}\right) \mathbf{Attn}^{(t)}_{\mathbf{p} \to \mathcal{P}_{k,n}} - z_1^3 \left(1 - \mathbf{Attn}^{(t)}_{\mathbf{p} \to \mathcal{P}_{k,1}}\right) \mathbf{Attn}^{(t)}_{\mathbf{p} \to \mathcal{P}_{k,1}} + \sum_{a \ne 1,n} z_a^2 z_1 \left(\mathbf{Attn}^{(t)}_{\mathbf{p} \to \mathcal{P}_{k,a}}\right)^2$$

$$\le -z_1 (1 - \mathbf{Attn}^{(t)}_{\mathbf{p} \to \mathcal{P}_{k,n}} - \mathbf{Attn}^{(t)}_{\mathbf{p} \to \mathcal{P}_{k,1}}) \left(z_n^2 \mathbf{Attn}^{(t)}_{\mathbf{p} \to \mathcal{P}_{k,n}} + z_1^2 \mathbf{Attn}^{(t)}_{\mathbf{p} \to \mathcal{P}_{k,1}} - \max_{a \ne 1,n} z_a^2 \mathbf{Attn}^{(t)}_{\mathbf{p} \to \mathcal{P}_{k,a}}\right).$$

Therefore, by Lemma C.1, we obtain

$$\alpha^{(t)}_{\mathbf{p} \to v_{k,1}} \le \mathbb{E}\left[\mathbf{1}\{k_X = k, \mathcal{E}_{k,1} \cap \mathbf{p} \in \mathcal{M}\} \mathbf{Attn}^{(t)}_{\mathbf{p} \to \mathcal{P}_{k,1}} \cdot \right.$$
$$\left. \left(-z_1(1 - \mathbf{Attn}^{(t)}_{\mathbf{p} \to \mathcal{P}_{k,n}} - \mathbf{Attn}^{(t)}_{\mathbf{p} \to \mathcal{P}_{k,1}}) \left(z_n^2 \mathbf{Attn}^{(t)}_{\mathbf{p} \to \mathcal{P}_{k,n}} + z_1^2 \mathbf{Attn}^{(t)}_{\mathbf{p} \to \mathcal{P}_{k,1}} - \max_{a \ne 1,n} z_a^2 \mathbf{Attn}^{(t)}_{\mathbf{p} \to \mathcal{P}_{k,a}}\right)\right)\right]$$

$$+ \mathbb{E}\left[\mathbf{1}\{k_X = k, \mathcal{E}^c_{k,1} \cap \mathbf{p} \in \mathcal{M}\} \mathbf{Attn}^{(t)}_{\mathbf{p} \to \mathcal{P}_{k,1}} \cdot \sum_{a \ne 1,n} z_1^2 z_a \left(\mathbf{Attn}^{(t)}_{\mathbf{p} \to \mathcal{P}_{k,a}}\right)^2\right]$$

$$\le -\min\left\{\Omega\left(\frac{1}{P^{(1-\frac{c_1^*L}{U})(1-\kappa_s)}}\right), \Omega\left(\frac{1}{P^{(\frac{U}{L}-1)(1-\kappa_s)}}\right)\right\} \cdot \Omega\left(\frac{1}{P^{(1-\kappa_c)+\frac{L}{U}(\frac{\Delta}{2}-0.01)}}\right)$$

where the second inequality invokes Lemma F.17 and equation F.6b. Moreover,

$$|\alpha^{(t)}_{\mathbf{p} \to v_{k,1}}| \lesssim \mathbb{E}\left[\mathbf{1}\{k_X = k, \mathcal{E}_{k,1} \cap \mathcal{E}_{k,n} \cap \mathbf{p} \in \mathcal{M}\} \mathbf{Attn}^{(t)}_{\mathbf{p} \to \mathcal{P}_{k,1}} \cdot \right.$$
$$\left. \left(z_1 z_n^2 \left(1 - \mathbf{Attn}^{(t)}_{\mathbf{p} \to \mathcal{P}_{k,n}}\right) \mathbf{Attn}^{(t)}_{\mathbf{p} \to \mathcal{P}_{k,n}} + z_1^3 \left(1 - \mathbf{Attn}^{(t)}_{\mathbf{p} \to \mathcal{P}_{k,1}}\right) \mathbf{Attn}^{(t)}_{\mathbf{p} \to \mathcal{P}_{k,1}}\right)\right]$$

$$= \mathbb{E}\left[\mathbf{1}\{k_X = k, \mathcal{E}_{k,1} \cap \mathcal{E}_{k,n} \cap \mathbf{p} \in \mathcal{M}\} z_1 z_n^2 \mathbf{Attn}^{(t)}_{\mathbf{p} \to \mathcal{P}_{k,1}} \cdot \left(1 - \mathbf{Attn}^{(t)}_{\mathbf{p} \to \mathcal{P}_{k,n}}\right) \mathbf{Attn}^{(t)}_{\mathbf{p} \to \mathcal{P}_{k,n}}\right]$$

$$+ \mathbb{E}\left[\mathbf{1}\{k_X = k, \mathcal{E}_{k,1} \cap \mathcal{E}_{k,n} \cap \mathbf{p} \in \mathcal{M}\} z_1^3 (\mathbf{Attn}^{(t)}_{\mathbf{p} \to \mathcal{P}_{k,1}})^2 \cdot \left(1 - \mathbf{Attn}^{(t)}_{\mathbf{p} \to \mathcal{P}_{k,1}}\right)\right]$$

$$\le \max\left\{O\left(\frac{\alpha^{(t)}_{\mathbf{p} \to v_{k,n}}}{P^{(1-\kappa_c)+\frac{L}{U}(\frac{\Delta}{2}-0.01)}}\right), O\left(\frac{\alpha^{(t)}_{\mathbf{p} \to v_{k,n}}}{P^{2(1-\kappa_c)+\frac{2L}{U}(\frac{\Delta}{2}-0.01)-(1-\frac{c_1^*L}{U})(1-\kappa_s)}}\right)\right\}$$

where the second inequality invokes Lemma F.17. $\square$

**Lemma F.21.** *For $n > 1$, if Induction Hypothesis E.1 and Induction Hypothesis F.3 hold at iteration $\widetilde{T}_1 + 1 \le t \le T_2$ for any $m > 1$ with $m \ne n$, the following holds*

$$|\alpha^{(t)}_{\mathbf{p} \to v_{k,m}}| \le O\left(\frac{\alpha^{(t)}_{\mathbf{p} \to v_{k,n}} - \alpha^{(t)}_{\mathbf{p} \to v_{k,1}}}{P^{1-\kappa_s}}\right).$$

The proof is similar to Lemma F.5, and thus omitted here.

### F.3.3 BOUNDING THE GRADIENT UPDATES OF POSITIONAL CORRELATIONS

We then summarize the properties for gradient updates of positional correlations, which utilizes the identical calculations as in Section F.1.3.

**Lemma F.22.** *For $n > 1$, if Induction Hypothesis E.1 and Induction Hypothesis F.3 hold at iteration $\widetilde{T}_1 + 1 \le t \le T_2$, then*

    *a. if $a_{k,\mathbf{q}} = n$ and $\mathbf{q} \ne \mathbf{p}$, $\beta^{(t)}_{k,\mathbf{p} \to \mathbf{q}} \ge 0$; $\beta^{(t)}_{k,\mathbf{p} \to \mathbf{q}} = \Theta(\frac{\alpha^{(t)}_{\mathbf{p} \to v_{k,n}}}{C_n})$ and $|\beta^{(t)}_n| = O\left(\frac{\alpha^{(t)}_{\mathbf{p} \to v_{k,n}} - \alpha^{(t)}_{\mathbf{p} \to v_{k,1}}}{P}\right).$*

    *b. if $a_{k,\mathbf{q}} = 1$, $|\beta^{(t)}_{k,\mathbf{p} \to \mathbf{q}}| = O\left(\frac{\alpha^{(t)}_{\mathbf{p} \to v_{k,n}} - \alpha^{(t)}_{\mathbf{p} \to v_{k,1}}}{P}\right) + O\left(\frac{|\alpha^{(t)}_{\mathbf{p} \to v_{k,1}}|}{C_1}\right).$*

    *c. if $a_{k,\mathbf{q}} = m$ and $m \ne 1, n$, $|\beta^{(t)}_{k,\mathbf{p} \to \mathbf{q}}| = O\left(\frac{\alpha^{(t)}_{\mathbf{p} \to v_{k,n}} - \alpha^{(t)}_{\mathbf{p} \to v_{k,1}}}{P}\right).$*

### F.3.4 END OF PHASE II, STAGE 1

**Lemma F.23.** *Induction Hypothesis F.3 holds for all $\widetilde{T}_1 + 1 \leq t \leq T_2$, and at iteration $t = T_2 + 1$, we have*

a. $\Phi_{\mathbf{p} \to v_{k,n}}^{(t)} > \frac{(1-\kappa_s)}{L} \log(P)$;

b. $\mathbf{Attn}_{\mathbf{p} \to \mathcal{P}_{k,n}}^{(t)} = \Omega(1)$ *if* $\mathsf{M} \in \mathcal{E}_{k,n}$.

*Proof.* By comparing Lemma F.19 and Lemma F.20-F.23, we have $\alpha_{\mathbf{p} \to v_{k,n}}^{(t)} \gg |\alpha_{\mathbf{p} \to v_{k,m}}^{(t)}|, |\beta_{k,\mathbf{p} \to \mathbf{q}}^{(t)}|$. Then the existence of $T_2 = \widetilde{T}_1 + O\left(\frac{\log(P)P^\Lambda}{\eta}\right)$ directly follows from Lemma F.19, where

$$\Lambda = \max\left\{(1 - \frac{c_1^* L}{U}), 2(\frac{U}{L} - 1)\right\} \cdot (1 - \kappa_s).$$

The second statement can be directly verified by noticing that $\Phi_{\mathbf{p} \to v_{k,n}}^{(t)} > \frac{(1-\kappa_s)}{L} \log(P)$ while all other attention correlations are sufficiently small. $\square$

### F.4 PHASE II, STAGE 2

In this final stage, we establish that these structures indeed represent the solutions toward which the algorithm converges.

Given any $0 < \epsilon < 1$, for $n > 1$, define

$$T_2^\epsilon \triangleq \max\left\{t > T_2 : \Phi_{\mathbf{p} \to v_{k,n}}^{(t)} \leq \log\left(c_5\left(\left(\frac{3}{\epsilon}\right)^{\frac{1}{2}} - 1\right)N\right)\right\}. \tag{F.7}$$

where $c_5$ is some largely enough constant.

We state the following induction hypotheses, which will hold throughout this stage:

**Induction Hypothesis F.4.** For $n > 1$, suppose $\text{polylog}(P) \gg \log(\frac{1}{\epsilon})$, for each $T_2 + 1 \leq t \leq T_2^\epsilon$, $\mathbf{q} \in \mathcal{P} \setminus \{\mathbf{p}\}$, the following holds:

a. $\Phi_{\mathbf{p} \to v_{k,n}}^{(t)}$ is monotonically increasing, and $\Phi_{\mathbf{p} \to v_{k,n}}^{(t)} \in [\frac{(1-\kappa_s)}{L} \log(P), O(\log(P/\epsilon))]$;

b. $\Phi_{\mathbf{p} \to v_{k,1}}^{(t)}$ is monotonically decreasing and $\Phi_{\mathbf{p} \to v_{k,1}}^{(t)} \in \left[-\frac{1}{L}\left(\frac{\Delta}{2} + 0.01\right)\log(P) - o(1), -\frac{1}{U}\left(\frac{\Delta}{2} - 0.01\right)\log(P)\right]$;

c. $|\Phi_{\mathbf{p} \to v_{k,m}}^{(t)}| = O\left(\frac{\Phi_{\mathbf{p} \to v_{k,n}}^{(t)} - \Phi_{\mathbf{p} \to v_{k,1}}^{(t)}}{P^{1-\kappa_s}}\right)$ for $m \neq 1, n$;

d. $\Upsilon_{k,\mathbf{p} \to \mathbf{q}}^{(t)} = O\left(\frac{\Phi_{\mathbf{p} \to v_{k,n}}^{(t)}}{C_n}\right)$ for $a_{k,\mathbf{q}} = n$, $|\Upsilon_{k,\mathbf{p} \to \mathbf{p}}^{(t)}| = O\left(\frac{\Phi_{\mathbf{p} \to v_{k,n}}^{(t)} - \Phi_{\mathbf{p} \to v_{k,1}}^{(t)}}{P}\right)$;

e. $|\Upsilon_{k,\mathbf{p} \to \mathbf{q}}^{(t)}| = O\left(\frac{|\Phi_{\mathbf{p} \to v_{k,1}}^{(t)}|}{C_1}\right) + O\left(\frac{\Phi_{\mathbf{p} \to v_{k,n}}^{(t)} - \Phi_{\mathbf{p} \to v_{k,1}}^{(t)}}{P}\right)$ for $a_{k,\mathbf{q}} = 1.$;

f. $|\Upsilon_{k,\mathbf{p} \to \mathbf{q}}^{(t)}| = O\left(\frac{\Phi_{\mathbf{p} \to v_{k,n}}^{(t)} - \Phi_{\mathbf{p} \to v_{k,1}}^{(t)}}{P}\right)$ for $a_{k,\mathbf{q}} \neq 1, n$.

### F.4.1 PROPERTY OF ATTENTION SCORES

We first single out several properties of attention scores that will be used for the proof of Induction Hypothesis F.4.

**Lemma F.24.** *if Induction Hypothesis E.1 and Induction Hypothesis F.4 hold at iteration $T_{n,2} < t \leq T_{n,2}^\epsilon$, then the following holds*

1. *if* $\mathsf{M} \in \mathcal{E}_{k,n}$, $\mathbf{Attn}^{(t)}_{\mathbf{p} \to \mathcal{P}_{k,n}} = \Omega(1)$ *and* $(1 - \mathbf{Attn}^{(t)}_{\mathbf{p} \to \mathcal{P}_{k,n}})^2 \geq O(\epsilon)$.

2. *Moreover,* $\mathbf{Attn}^{(t)}_{\mathbf{p} \to \mathcal{P}_{k,1}} = O\Big( \frac{1 - \mathbf{Attn}^{(t)}_{\mathbf{p} \to \mathcal{P}_{k,n}}}{P^{(1-\kappa_c) + \frac{L}{U}(\frac{\Delta}{2} - 0.01)}} \Big)$; *if* $\mathsf{M} \in \mathcal{E}_{k,1}$, *we have* $\mathbf{Attn}^{(t)}_{\mathbf{p} \to \mathcal{P}_{k,1}} = \Omega\Big( \frac{1 - \mathbf{Attn}^{(t)}_{\mathbf{p} \to \mathcal{P}_{k,n}}}{P^{(1-\kappa_c) + \frac{U}{L}(\frac{\Delta}{2} + 0.01)}} \Big)$;

3. *for* $\mathbf{q} \in \mathcal{M} \cap (\mathcal{P}_{k,n} \cup \mathcal{P}_{k,1})$, $\mathbf{attn}^{(t)}_{\mathbf{p} \to \mathbf{q}} = O\Big( \frac{1 - \mathbf{Attn}^{(t)}_{\mathbf{p} \to \mathcal{P}_{k,n}}}{P} \Big)$.

**Lemma F.25.** *if Induction Hypothesis E.1 and Induction Hypothesis F.4 hold at iteration* $T_{n,2} < t \leq T^{\epsilon}_{n,2}$*,then for* $m \neq n$*, the following holds:*

1. *for any* $\mathbf{q} \in \mathcal{P}_{k,m}$, $\mathbf{attn}^{(t)}_{\mathbf{p} \to \mathbf{q}} \leq O\Big( \frac{1 - \mathbf{Attn}^{(t)}_{\mathbf{p} \to \mathcal{P}_{k,n}}}{P} \Big)$.

2. $\mathbf{Attn}^{(t)}_{\mathbf{p} \to \mathcal{P}_{k,n}} \leq O\Big( \frac{1 - \mathbf{Attn}^{(t)}_{\mathbf{p} \to \mathcal{P}_{k,n}}}{N} \Big)$, *and if* $\mathsf{M} \in \mathcal{E}_{k,m}$, $\mathbf{Attn}^{(t)}_{\mathbf{p} \to \mathcal{P}_{k,n}} = \Theta\Big( \frac{1 - \mathbf{Attn}^{(t)}_{\mathbf{p} \to \mathcal{P}_{k,n}}}{N} \Big)$.

### F.4.2 BOUNDING THE GRADIENT UPDATES OF FP CORRELATIONS

**Lemma F.26.** *For* $n > 1$*, if Induction Hypothesis E.1 and Induction Hypothesis F.4 hold at iteration* $T_2 + 1 \leq t \leq T^{\epsilon}_2$*, then* $\alpha^{(t)}_{\mathbf{p} \to v_{k,n}} \geq 0$ *and satisfies:*

$$\alpha^{(t)}_{\mathbf{p} \to v_{k,n}} \geq \Omega(\epsilon).$$

*Proof.* By Lemma C.2, we have

$$\alpha^{(t)}_{\mathbf{p} \to v_{k,n}}$$

$$= \mathbb{E}\left[ \mathbf{1}\{k_X = k, \mathbf{p} \in \mathcal{M}\} \mathbf{Attn}^{(t)}_{\mathbf{p} \to \mathcal{P}_{k,n}} \cdot \left( z^3_n \left( 1 - \mathbf{Attn}^{(t)}_{\mathbf{p} \to \mathcal{P}_{k,n}} \right)^2 + \sum_{m \neq n} z^2_m z_n \left( \mathbf{Attn}^{(t)}_{\mathbf{p} \to \mathcal{P}_{k,m}} \right)^2 \right) \right]$$

$$= \mathbb{E}\left[ \mathbf{1}\{k_X = k, \mathcal{E}_{k,n} \cap \mathbf{p} \in \mathcal{M}\} \mathbf{Attn}^{(t)}_{\mathbf{p} \to \mathcal{P}_{k,n}} \cdot \left( z^3_n \left( 1 - \mathbf{Attn}^{(t)}_{\mathbf{p} \to \mathcal{P}_{k,n}} \right)^2 + \sum_{m \neq n} z^2_m z_n \left( \mathbf{Attn}^{(t)}_{\mathbf{p} \to \mathcal{P}_{k,m}} \right)^2 \right) \right]$$

$$+ \mathbb{E}\left[ \mathbf{1}\{k_X = k, \mathcal{E}^c_{k,n} \cap \mathbf{p} \in \mathcal{M}\} \mathbf{Attn}^{(t)}_{\mathbf{p} \to \mathcal{P}_{k,n}} \cdot \left( z^3_n \left( 1 - \mathbf{Attn}^{(t)}_{\mathbf{p} \to \mathcal{P}_{k,n}} \right)^2 + \sum_{m \neq n} z^2_m z_n \left( \mathbf{Attn}^{(t)}_{\mathbf{p} \to \mathcal{P}_{k,m}} \right)^2 \right) \right]$$

$$\gtrsim \mathbb{P}(\mathsf{M} \in \mathcal{E}_{k,n}) \cdot$$

$$\mathbb{E}\left[ \mathbf{1}\{k_X = k, \mathbf{p} \in \mathcal{M}\} \mathbf{Attn}^{(t)}_{\mathbf{p} \to \mathcal{P}_{k,n}} \cdot \left( z^3_n \left( 1 - \mathbf{Attn}^{(t)}_{\mathbf{p} \to \mathcal{P}_{k,n}} \right)^2 + \sum_{m \neq n} z^2_m z_n \left( \mathbf{Attn}^{(t)}_{\mathbf{p} \to \mathcal{P}_{k,m}} \right)^2 \right) \Big| \mathcal{E}_{k,n} \right]$$

$$+ O(1) \cdot \mathbb{P}(\mathsf{M} \in \mathcal{E}^c_{k,n})$$

$$\gtrsim \Omega(\epsilon)$$

where the last inequality invokes Lemma F.24, Lemma D.6 and the fact that

$$\epsilon \geq \exp(- \operatorname{polylog}(K)) \gg \exp(-c_{n,1} C_n).$$

$\square$

**Lemma F.27.** *For* $n > 1$*, if Induction Hypothesis E.1 and Induction Hypothesis F.4 hold at iteration* $T_{n,3} < t \leq T^{\epsilon}_{n,4}$*, then* $\alpha^{(t)}_{\mathbf{p} \to v_{k,1}} < 0$ *and satisfies*

$$|\alpha^{(t)}_{\mathbf{p} \to v_{k,m}}| \leq \max\left\{ O\Big( \frac{\alpha^{(t)}_{\mathbf{p} \to v_{k,n}}}{P^{(1-\kappa_c) + \frac{L}{U}(\Delta/2 - 0.01)}} \Big), O\Big( \frac{\alpha^{(t)}_{\mathbf{p} \to v_{k,n}}}{P^{2(1-\kappa_c) + \frac{L}{U}(\Delta - 0.02) - (1 - \frac{c^*_1 L}{U})(1 - \kappa_s)}} \Big) \right\}$$

The proof follows the similar arguments Lemma F.20 by noticing that $\epsilon \gg \mathbb{P}(\mathsf{M} \in \mathcal{E}_{k,m}^c)$ for any $m \neq n$.

**Lemma F.28.** *For $n > 1$, if Induction Hypothesis E.1 and Induction Hypothesis F.4 hold at iteration $T_2 < t \leq T_2^\epsilon$, then for any $m > 1$ with $m \neq n$, the following holds*

$$-O\left(\frac{\alpha_{\mathbf{p} \to v_{k,n}}^{(t)}}{P^{1-\kappa_s}}\right) \leq \alpha_{\mathbf{p} \to v_{k,m}}^{(t)} \leq 0$$

*Proof.* We first note that

$$- z_1 z_n^2 \left(1 - \mathbf{Attn}_{\mathbf{p} \to \mathcal{P}_{k,n}}^{(t)}\right) \mathbf{Attn}_{\mathbf{p} \to \mathcal{P}_{k,n}}^{(t)} - z_m^3 \left(1 - \mathbf{Attn}_{\mathbf{p} \to \mathcal{P}_{k,m}}^{(t)}\right) \mathbf{Attn}_{\mathbf{p} \to \mathcal{P}_{k,1}}^{(t)} + \sum_{a \neq 1,n} z_a^2 z_m \left(\mathbf{Attn}_{\mathbf{p} \to \mathcal{P}_{k,a}}^{(t)}\right)^2$$

$$\leq z_m \left(\max_{a \neq m,n} z_a^2 \mathbf{Attn}_{\mathbf{p} \to \mathcal{P}_{k,a}}^{(t)} - z_n^2 \mathbf{Attn}_{\mathbf{p} \to \mathcal{P}_{k,n}}^{(t)} - z_m^2 \mathbf{Attn}_{\mathbf{p} \to \mathcal{P}_{k,m}}^{(t)}\right) \left(1 - \mathbf{Attn}_{\mathbf{p} \to \mathcal{P}_{k,n}}^{(t)} - \mathbf{Attn}_{\mathbf{p} \to \mathcal{P}_{k,m}}^{(t)}\right)$$

$$\lesssim -\Omega(1 - \mathbf{Attn}_{\mathbf{p} \to \mathcal{P}_{k,n}}^{(t)})$$

since when $\mathsf{M} \in \mathcal{E}_{k,n}$, we have $\mathbf{Attn}_{\mathbf{p} \to \mathcal{P}_{k,n}}^{(t)} = \Omega(1) \gg \mathbf{Attn}_{\mathbf{p} \to \mathcal{P}_{k,a}}^{(t)}$. Thus, we have

$$0 \geq \alpha_{\mathbf{p} \to v_{k,m}}^{(t)} \gtrsim -\mathbb{E}\left[\mathbf{1}\{k_X = k, \mathcal{E}_{k,n} \cap \mathbf{p} \in \mathcal{M}\} \mathbf{Attn}_{\mathbf{p} \to \mathcal{P}_{k,m}}^{(t)} \cdot \Omega(1 - \mathbf{Attn}_{\mathbf{p} \to \mathcal{P}_{k,n}}^{(t)})\right]$$

$$\geq -O\left(\frac{\alpha_{\mathbf{p} \to v_{k,n}}^{(t)}}{P^{1-\kappa_s}}\right).$$

$\square$

### F.4.3 BOUNDING THE GRADIENT UPDATES OF POSITIONAL CORRELATIONS

We then summarize the properties for gradient updates of positional correlations, which utilizes the identical calculations as in Section F.1.3.

**Lemma F.29.** *For $n > 1$, if Induction Hypothesis E.1 and Induction Hypothesis F.4 hold at iteration $T_2 + 1 \leq t \leq T_2^\epsilon$, then*

  a. *if $a_{k,\mathbf{q}} = n$ and $\mathbf{q} \neq \mathbf{p}$, $\beta_{k,\mathbf{p} \to \mathbf{q}}^{(t)} \geq 0$; $\beta_{k,\mathbf{p} \to \mathbf{q}}^{(t)} = \Theta\left(\frac{\alpha_{\mathbf{p} \to v_{k,n}}^{(t)}}{C_n}\right)$ and $|\beta_n^{(t)}| = O\left(\frac{\alpha_{\mathbf{p} \to v_{k,n}}^{(t)} - \alpha_{\mathbf{p} \to v_{k,1}}^{(t)}}{P}\right)$.*

  b. *if $a_{k,\mathbf{q}} = 1$, $|\beta_{k,\mathbf{p} \to \mathbf{q}}^{(t)}| = O\left(\frac{\alpha_{\mathbf{p} \to v_{k,n}}^{(t)} - \alpha_{\mathbf{p} \to v_{k,1}}^{(t)}}{P}\right) + O\left(\frac{|\alpha_{\mathbf{p} \to v_{k,1}}^{(t)}|}{C_1}\right)$.*

  c. *if $a_{k,\mathbf{q}} = m$ and $m \neq 1, n$, $|\beta_{k,\mathbf{p} \to \mathbf{q}}^{(t)}| = O\left(\frac{\alpha_{\mathbf{p} \to v_{k,n}}^{(t)} - \alpha_{\mathbf{p} \to v_{k,1}}^{(t)}}{P}\right)$.*

### F.4.4 END OF PHASE II, STAGE 2

**Lemma F.30.** *For $n > 1$, and $0 < \epsilon < 1$, suppose $\mathrm{polylog}(P) \gg \log(\frac{1}{\epsilon})$. Then Induction Hypothesis F.4 holds for all $T_2 < t \leq T_2^\epsilon = T_2 + O\left(\frac{\log(P\epsilon^{-1})}{\eta\epsilon}\right)$, and at iteration $t = T_2^\epsilon + 1$, we have*

  1. $\widetilde{\mathcal{L}}_{k,\mathbf{p}}(Q^{T_2^\epsilon+1}) < \frac{\epsilon}{2K}$;

  2. *If $\mathsf{M} \in \mathcal{E}_{k,n}$, we have $(1 - \mathbf{Attn}_{\mathbf{p} \to \mathcal{P}_{k,n}}^{(T_2^\epsilon+1)})^2 \leq O(\epsilon)$.*

*Proof.* The existence of $T_{2,k}^\epsilon = T_{2,k} + O(\frac{\log(P\epsilon^{-1})}{\eta\epsilon})$ directly follows from Lemma F.26. We further derive

$$\widetilde{\mathcal{L}}_{k,\mathbf{p}}(Q^{T_2^\epsilon+1}) =$$

$$\frac{1}{2}\mathbb{E}\left[\mathbf{1}\{k_X = k, \mathbf{p} \in \mathcal{M} \cap \mathsf{M} \in \mathcal{E}_{k,n}\}\left(z_n^2\left(1 - \mathbf{Attn}_{\mathbf{p}\to\mathcal{P}_{k,n}}\right)^2 + \sum_{m\neq n} z_m^2\left(\mathbf{Attn}_{n,m}\right)^2\right)\right]$$

$$\leq \frac{1}{2K} \cdot \gamma \cdot U^2 \cdot (1 + o(1)) \cdot O(\epsilon)$$

$$\leq \frac{\epsilon}{2K}$$

where the first inequality is due to direct calculations by the definition of $T_2^\epsilon$, and the second inequality can be obtained by setting $c_{n,2}$ in equation F.7 sufficiently large. □

## G  ANALYSIS FOR LOCAL AREAS WITH NEGATIVE INFORMATION GAP

In this section, we focus on a specific patch $\mathbf{p} \in \mathcal{P}$ with the $k$-th cluster for $k \in [K]$, and present the analysis for the case that $X_{\mathbf{p}}$ is located in the local area for the $k$-th cluster, i.e. $a_{k,\mathbf{p}} > 1$. Throughout this section, we denote $a_{k,\mathbf{p}} = n$ for simplicity. When $\Delta \leq -\Omega(1)$, we can show that the gap of attention correlation changing rate for the positive case does not exist anymore, and conversely $\alpha_{\mathbf{p}\to v_{k,n}}^{(t)} \gg \alpha_{\mathbf{p}\to v_{k,1}}^{(t)}$ from the beginning. We can reuse most of the gradient calculations in the previous section and only sketch them in this section.

**Stage 1:**  we define stage 1 as all iterations $0 \leq t \leq T_{\text{neg},1}$, where

$$T_{\text{neg},1} \triangleq \max\left\{t : \Phi_{\mathbf{p}\to v_{k,n}}^{(t)} \leq \frac{(1-\kappa_s)}{L}\log(P)\right\}.$$

We state the following induction hypothesis, which will hold throughout this stage:

**Induction Hypothesis G.1.**  For each $0 \leq t \leq T_{\text{neg},1}$, $\mathbf{q} \in \mathcal{P} \setminus \{\mathbf{p}\}$, the following holds:

a.  $\Phi_{\mathbf{p}\to v_{k,n}}^{(t)}$ is monotonically increasing, and $\Phi_{\mathbf{p}\to v_{k,n}}^{(t)} \in \left[0, \frac{(1-\kappa_s)}{L}\log(P)\right]$;

b.  $\Phi_{\mathbf{p}\to v_{k,1}}^{(t)}$ is monotonically decreasing and $\Phi_{\mathbf{p}\to v_{k,1}}^{(t)} \in \left[-O\left(\frac{\Phi_{\mathbf{p}\to v_{k,n}}^{(t)}}{P^{-\Delta}}\right), 0\right]$;

c.  $|\Phi_{\mathbf{p}\to v_{k,m}}^{(t)}| = O\left(\frac{\Phi_{\mathbf{p}\to v_{k,n}}^{(t)} - \Phi_{\mathbf{p}\to v_{k,1}}^{(t)}}{P^{1-\kappa_s}}\right)$ for $m \neq 1, n$;

d.  $\Upsilon_{k,\mathbf{p}\to\mathbf{q}}^{(t)} = O\left(\frac{\Phi_{\mathbf{p}\to v_{k,n}}^{(t)}}{C_n}\right)$ for $a_{k,\mathbf{q}} = n$, $|\Upsilon_{k,\mathbf{p}\to\mathbf{p}}^{(t)}| = O\left(\frac{\Phi_{\mathbf{p}\to v_{k,n}}^{(t)} - \Phi_{\mathbf{p}\to v_{k,1}}^{(t)}}{P}\right)$;

e.  $|\Upsilon_{k,\mathbf{p}\to\mathbf{q}}^{(t)}| = O\left(\frac{|\Phi_{\mathbf{p}\to v_{k,1}}^{(t)}|}{C_1}\right) + O\left(\frac{\Phi_{\mathbf{p}\to v_{k,n}}^{(t)} - \Phi_{\mathbf{p}\to v_{k,1}}^{(t)}}{P}\right)$ for $a_{k,\mathbf{q}} = 1$;

f.  $|\Upsilon_{k,\mathbf{p}\to\mathbf{q}}^{(t)}| = O\left(\frac{\Phi_{\mathbf{p}\to v_{k,n}}^{(t)} - \Phi_{\mathbf{p}\to v_{k,1}}^{(t)}}{P}\right)$ for $a_{k,\mathbf{q}} \neq 1, n$.

Through similar calculations for phase II, stage 1 in Appendix F.3, we obtain the following lemmas to control the gradient updates for attention correlations.

**Lemma G.1.**  *If Induction Hypothesis E.2 and Induction Hypothesis G.1 hold for $0 \leq t \leq T_{\text{neg},1}$, then we have*

$$\alpha_{\mathbf{p}\to v_{k,n}}^{(t)} \geq \min\left\{\Omega\left(\frac{1}{P^{(1-\kappa_s)}}\right), \Omega\left(\frac{1}{P^{2(\frac{U}{L}-1)(1-\kappa_s)}}\right)\right\}, \tag{G.1a}$$

$$0 \geq \alpha_{\mathbf{p}\to v_{k,1}}^{(t)} \geq -O\left(\frac{\alpha_{\mathbf{p}\to v_{k,n}}^{(t)}}{P^{-\Delta}}\right), \tag{G.1b}$$

$$|\alpha_{\mathbf{p}\to v_{k,m}}^{(t)}| \leq O\left(\frac{\alpha_{\mathbf{p}\to v_{k,n}}^{(t)} - \alpha_{\mathbf{p}\to v_{k,1}}^{(t)}}{P^{1-\kappa_s}}\right) \text{ for all } m \neq n, 1 \tag{G.1c}$$

$$\beta_{k,\mathbf{p}\to\mathbf{q}}^{(t)} = \Theta\Big(\frac{\alpha_{\mathbf{p}\to v_{k,n}}^{(t)}}{C_n}\Big) \text{ for } a_{k,\mathbf{q}} = n, \mathbf{q} \neq \mathbf{p} \tag{G.1d}$$

$$|\beta_{k,\mathbf{p}\to\mathbf{q}}^{(t)}| = O\Big(\frac{\alpha_{\mathbf{p}\to v_{k,n}}^{(t)}}{P}\Big) + O\Big(\frac{|\alpha_{\mathbf{p}\to v_{k,1}}^{(t)}|}{C_1}\Big) \text{ for } a_{k,\mathbf{q}} = 1, \tag{G.1e}$$

$$|\beta_{k,\mathbf{p}\to\mathbf{p}}^{(t)}|, |\beta_{k,\mathbf{p}\to\mathbf{q}}^{(t)}| = O\Big(\frac{\alpha_{\mathbf{p}\to v_{k,n}}^{(t)} - \alpha_{\mathbf{p}\to v_{k,1}}^{(t)}}{P}\Big) \quad \text{for all } a_{k,\mathbf{p}} \neq n, 1. \tag{G.1f}$$

Here $\Delta < 0$ implies $|\alpha_{\mathbf{p}\to v_{k,1}}^{(t)}| \ll \alpha_{\mathbf{p}\to v_{k,n}}^{(t)}$. Induction Hypothesis G.1 can be directly proved by Lemma G.1 and we have

$$T_{\text{neg},1} = O\Big(\frac{P^{\max\{1, 2(\frac{U}{L}-1)\}\cdot(1-\kappa_s)}\log(P)}{\eta}\Big). \tag{G.2}$$

**Stage 2:** Given any $0 < \epsilon < 1$, define

$$T_{\text{neg},1}^{\epsilon} \triangleq \max\left\{ t > T_1 : \Phi_{\mathbf{p}\to v_{k,n}}^{(t)} \leq \log\left( c_6\left( \Big(\frac{3}{\epsilon}\Big)^{\frac{1}{2}} - 1\right) P^{1-\kappa_s}\right)\right\}. \tag{G.3}$$

where $c_6$ is some largely enough constant. We then state the following induction hypotheses, which will hold throughout this stage:

**Induction Hypothesis G.2.** For $n > 1$, suppose $\text{polylog}(P) \gg \log(\frac{1}{\epsilon})$, for $\mathbf{q} \in \mathcal{P} \setminus \{\mathbf{p}\}$, and each $T_{\text{neg},1} < t \leq T_{\text{neg},1}^{\epsilon}$, the following holds:

a. $\Phi_{\mathbf{p}\to v_{k,n}}^{(t)}$ is monotonically increasing, and $\Phi_{\mathbf{p}\to v_{k,n}}^{(t)} \in \Big[\frac{(1-\kappa_s)}{L}\log(P), O(\log(P/\epsilon))\Big]$;

b. $\Phi_{\mathbf{p}\to v_{k,1}}^{(t)}$ is monotonically decreasing and $\Phi_{\mathbf{p}\to v_{k,1}}^{(t)} \in \Big[-O\Big(\frac{\Phi_{\mathbf{p}\to v_{k,n}}^{(t)}}{P^{-\Delta}}\Big), 0\Big]$;

c. $|\Phi_{\mathbf{p}\to v_{k,m}}^{(t)}| = O\Big(\frac{\Phi_{\mathbf{p}\to v_{k,n}}^{(t)} - \Phi_{\mathbf{p}\to v_{k,1}}^{(t)}}{P^{1-\kappa_s}}\Big)$ for $m \neq 1, n$;

d. $\Upsilon_{k,\mathbf{p}\to\mathbf{q}}^{(t)} = O\Big(\frac{\Phi_{\mathbf{p}\to v_{k,n}}^{(t)}}{C_n}\Big)$ for $a_{k,\mathbf{q}} = n$, $|\Upsilon_{k,\mathbf{p}\to\mathbf{p}}^{(t)}| = O\Big(\frac{\Phi_{\mathbf{p}\to v_{k,n}}^{(t)} - \Phi_{\mathbf{p}\to v_{k,1}}^{(t)}}{P}\Big)$;

e. $|\Upsilon_{k,\mathbf{p}\to\mathbf{q}}^{(t)}| = O\Big(\frac{|\Phi_{\mathbf{p}\to v_{k,1}}^{(t)}|}{C_1}\Big) + O\Big(\frac{\Phi_{\mathbf{p}\to v_{k,n}}^{(t)} - \Phi_{\mathbf{p}\to v_{k,1}}^{(t)}}{P}\Big)$ for $a_{k,\mathbf{q}} = 1$;

f. $|\Upsilon_{k,\mathbf{p}\to\mathbf{q}}^{(t)}| = O\Big(\frac{\Phi_{\mathbf{p}\to v_{k,n}}^{(t)} - \Phi_{\mathbf{p}\to v_{k,1}}^{(t)}}{P}\Big)$ for $a_{k,\mathbf{q}} \neq 1, n$.

**Lemma G.2.** *If Induction Hypothesis E.2 and Induction Hypothesis G.2 hold for* $T_{\text{neg},1} < t \leq T_{\text{neg},1}^{\epsilon}$, *then we have*

$$\alpha_{\mathbf{p}\to v_{k,n}}^{(t)} \geq \Omega(\epsilon), \tag{G.4a}$$

$$0 \geq \alpha_{\mathbf{p}\to v_{k,1}}^{(t)} \geq -O\Big(\frac{\alpha_{\mathbf{p}\to v_{k,n}}^{(t)}}{P^{-\Delta}}\Big), \tag{G.4b}$$

$$|\alpha_{\mathbf{p}\to v_{k,m}}^{(t)}| \leq O\Big(\frac{\alpha_{\mathbf{p}\to v_{k,n}}^{(t)} - \alpha_{\mathbf{p}\to v_{k,1}}^{(t)}}{P^{1-\kappa_s}}\Big) \text{ for all } m \neq n, 1 \tag{G.4c}$$

$$\beta_{k,\mathbf{p}\to\mathbf{q}}^{(t)} = \Theta\Big(\frac{\alpha_{\mathbf{p}\to v_{k,n}}^{(t)}}{C_n}\Big) \text{ for } a_{k,\mathbf{q}} = n, \mathbf{q} \neq \mathbf{p} \tag{G.4d}$$

$$|\beta_{k,\mathbf{p}\to\mathbf{q}}^{(t)}| = O\Big(\frac{\alpha_{\mathbf{p}\to v_{k,n}}^{(t)}}{P}\Big) + O\Big(\frac{|\alpha_{\mathbf{p}\to v_{k,1}}^{(t)}|}{C_1}\Big) \text{ for } a_{k,\mathbf{q}} = 1, \tag{G.4e}$$

$$|\beta_{k,\mathbf{p}\to\mathbf{p}}^{(t)}|, |\beta_{k,\mathbf{p}\to\mathbf{q}}^{(t)}| = O\Big(\frac{\alpha_{\mathbf{p}\to v_{k,n}}^{(t)} - \alpha_{\mathbf{p}\to v_{k,1}}^{(t)}}{P}\Big) \quad \text{for all } a_{k,\mathbf{p}} \neq n, 1. \tag{G.4f}$$

Induction Hypothesis G.2 can be directly proved by Lemma G.2. Furthermore, at the end of this stage, we will have:

**Lemma G.3.** *Suppose* $\mathrm{polylog}(P) \gg \log(\frac{1}{\epsilon})$, *then Induction Hypothesis G.2 holds for all* $T_{\mathrm{neg},1} < t \leq T_{\mathrm{neg},1}^{\epsilon} = T_{\mathrm{neg},1} + O\left(\frac{\log(P\epsilon^{-1})}{\eta\epsilon}\right)$, *and at iteration* $t = T_{\mathrm{neg},1}^{\epsilon} + 1$, *we have*

    *1.* $\widetilde{\mathcal{L}}_{k,\mathbf{p}}(Q^{T_{\mathrm{neg},1}^{\epsilon}+1}) < \frac{\epsilon}{2K}$;

    *2. If* $\mathsf{M} \in \mathcal{E}_{k,n}$, *we have* $\left(1 - \mathbf{Attn}_{\mathbf{p}\to\mathcal{P}_{k,n}}^{(T_{\mathrm{neg},1}^{\epsilon}+1)}\right)^2 \leq O(\epsilon).$

## H    ANALYSIS FOR THE GLOBAL AREA

When $a_{\mathbf{p},k} = 1$, i.e. the patch lies in the global area, the analysis is much simpler and does not depend on the value of $\Delta$. We can reuse most of the gradient calculations in Appendix F and only sketch them in this section.

For $X_{\mathbf{p}}$ in the global region $\mathcal{P}_{k,1}$, since the overall attention $\mathbf{Attn}_{\mathbf{p}\to\mathcal{P}_{k,1}}^{(0)}$ to the target feature already reaches $\Omega\left(\frac{C_1}{P}\right) = \Omega\left(\frac{1}{P^{1-\kappa_c}}\right)$ due to the large number of unmasked patches featuring $v_{k,1}$ when $\mathsf{M} \in \mathcal{E}_{k,1}$, which is significantly larger than $\mathbf{Attn}_{\mathbf{p}\to\mathcal{P}_{k,m}}^{(0)} = \Theta\left(\frac{1}{P^{1-\kappa_s}}\right)$ for all other $m > 1$. This results in large $\alpha_{\mathbf{p}\to v_{k,1}}^{(t)}$ initially, and thus the training directly enters phase II.

**Stage 1:**    we define stage 1 as all iterations $0 \leq t \leq T_{c,1}$, where

$$T_{c,1} \triangleq \max\left\{t : \Phi_{\mathbf{p}\to v_{k,1}}^{(t)} \leq \frac{(1-\kappa_c)}{L}\log(P)\right\}.$$

We state the following induction hypotheses, which will hold throughout this stage:

**Induction Hypothesis H.1.** *For each* $0 \leq t \leq T_{c,1}$, $\mathbf{q} \in \mathcal{P} \setminus \{\mathbf{p}\}$, *the following holds:*

    a. $\Phi_{\mathbf{p}\to v_{k,1}}^{(t)}$ *is monotonically increasing, and* $\Phi_{\mathbf{p}\to v_{k,1}}^{(t)} \in \left[0, \frac{(1-\kappa_c)}{L}\log(P)\right]$;

    b. $\Phi_{\mathbf{p}\to v_{k,m}}$ *is monotonically decreasing for* $m > 1$ *and* $\Phi_{\mathbf{p}\to v_{k,m}} \in \left[-O\left(\frac{\log(P)}{N}\right), 0\right]$;

    c. $\Upsilon_{k,\mathbf{p}\to\mathbf{q}}^{(t)} = O\left(\frac{\Phi_{\mathbf{p}\to v_{k,1}}^{(t)}}{C_1}\right)$ *for* $a_{k,\mathbf{q}} = 1$, $|\Upsilon_{k,\mathbf{p}\to\mathbf{p}}^{(t)}| = O\left(\frac{\Phi_{\mathbf{p}\to v_{k,1}}^{(t)}}{P}\right)$;

    d. $|\Upsilon_{k,\mathbf{p}\to\mathbf{q}}^{(t)}| = O\left(\frac{\Phi_{\mathbf{p}\to v_{k,1}}^{(t)}}{P}\right)$ *for* $a_{k,\mathbf{q}} \neq 1$.

Through similar calculations for phase II, stage 1 in Appendix F.3, we obtain the following lemmas to control the gradient updates for attention correlations.

**Lemma H.1.** *If Induction Hypothesis E.1 (or Induction Hypothesis E.2) and Induction Hypothesis H.1 hold for* $0 \leq t \leq T_{c,1}$, *then we have*

$$\alpha_{\mathbf{p}\to v_{k,1}}^{(t)} \geq \min\left\{\Omega\left(\frac{1}{P^{(1-\kappa_c)}}\right), \Omega\left(\frac{1}{P^{2(\frac{U}{L}-1)(1-\kappa_c)}}\right)\right\}, \tag{H.1a}$$

$$|\alpha_{\mathbf{p}\to v_{k,m}}^{(t)}| \leq O\left(\frac{\alpha_{\mathbf{p}\to v_{k,1}}^{(t)}}{P^{1-\kappa_s}}\right) \quad \text{for all } m \neq 1, \tag{H.1b}$$

$$\beta_{k,\mathbf{p}\to\mathbf{q}}^{(t)} = \Theta\left(\frac{\alpha_{\mathbf{p}\to v_{k,1}}^{(t)}}{C_1}\right), \text{ for } a_{k,\mathbf{q}} = 1, \mathbf{q} \neq \mathbf{p}, \tag{H.1c}$$

$$|\beta_{k,\mathbf{p}\to\mathbf{p}}^{(t)}|, |\beta_{k,\mathbf{p}\to\mathbf{q}}^{(t)}| = O\left(\frac{\alpha_{\mathbf{p}\to v_{k,1}}^{(t)}}{P}\right) \quad \text{for all } a_{k,\mathbf{q}} > 1. \tag{H.1d}$$

Induction Hypothesis H.1 can be directly proved by Lemma H.1 and we have

$$T_{c,1} = O\left(\frac{P^{\max\{1, 2(\frac{U}{L}-1)\}\cdot(1-\kappa_c)}\log(P)}{\eta}\right). \tag{H.2}$$

**Stage 2:**   Given any $0 < \epsilon < 1$, define

$$T_{c,1}^{\epsilon} \triangleq \max\left\{t > T_{c,1} : \Phi_{\mathbf{p}\to v_{k,1}}^{(t)} \leq \log\left(c_7\left(\left(\frac{3}{\epsilon}\right)^{\frac{1}{2}} - 1\right)P^{1-\kappa_c}\right)\right\}. \tag{H.3}$$

where $c_7$ is some largely enough constant. We then state the following induction hypotheses, which will hold throughout this stage:

**Induction Hypothesis H.2.** For $n > 1$, suppose $\mathrm{polylog}(P) \gg \log(\frac{1}{\epsilon})$, $\mathbf{q} \in \mathcal{P} \setminus \{\mathbf{p}\}$, for each $T_{c,1} + 1 \leq t \leq T_{c,1}^{\epsilon}$, the following holds:

a. $\Phi_{\mathbf{p}\to v_{k,1}}^{(t)}$ is monotonically increasing, and $\Phi_{\mathbf{p}\to v_{k,1}}^{(t)} \in \left[\frac{(1-\kappa_c)}{L}\log(P), O(\log(P/\epsilon))\right]$;

b. $\Phi_{\mathbf{p}\to v_{k,m}}$ is monotonically decreasing for $n > 1$ and $\Phi_{\mathbf{p}\to v_{k,m}} \in \left[-O\left(\frac{\log(P)}{N}\right), 0\right]$;

c. $\Upsilon_{k,\mathbf{p}\to\mathbf{q}}^{(t)} = O\left(\frac{\Phi_{\mathbf{p}\to v_{k,1}}^{(t)}}{C_1}\right)$ for $a_{k,\mathbf{q}} = 1$, $|\Upsilon_{k,\mathbf{p}\to\mathbf{p}}^{(t)}| = O\left(\frac{\Phi_{\mathbf{p}\to v_{k,1}}^{(t)}}{P}\right)$;

d. $|\Upsilon_{k,\mathbf{p}\to\mathbf{q}}^{(t)}| = O\left(\frac{\Phi_{\mathbf{p}\to v_{k,1}}^{(t)}}{P}\right)$ for $a_{k,\mathbf{q}} \neq 1$.

We also have the following lemmas to control the gradient updates for attention correlations.

**Lemma H.2.** *If Induction Hypothesis E.1 (or Induction Hypothesis E.2) and Induction Hypothesis H.1 hold for $T_{c,1} + 1 \leq t \leq T_{c,1}^{\epsilon}$, then we have*

$$\alpha_{\mathbf{p}\to v_{k,1}}^{(t)} \geq \Omega(\epsilon), |\alpha_{\mathbf{p}\to v_{k,m}}^{(t)}| \leq O\left(\frac{\alpha_{\mathbf{p}\to v_{k,1}}^{(t)}}{P^{1-\kappa_s}}\right) \quad \text{for all } m \neq 1 \tag{H.4a}$$

$$\beta_{k,\mathbf{p}\to\mathbf{q}}^{(t)} = \Theta\left(\frac{\alpha_{\mathbf{p}\to v_{k,1}}^{(t)}}{C_1}\right), \text{ for } a_{k,\mathbf{q}} = 1, \mathbf{q} \neq \mathbf{p} \tag{H.4b}$$

$$|\beta_{k,\mathbf{p}\to\mathbf{p}}^{(t)}|, |\beta_{k,\mathbf{p}\to\mathbf{q}}^{(t)}| = O\left(\frac{\alpha_{\mathbf{p}\to v_{k,1}}^{(t)}}{P}\right) \quad \text{for all } a_{k,\mathbf{q}} > 1. \tag{H.4c}$$

Induction Hypothesis H.2 can be directly proved by Lemma H.2. Furthermore, at the end of this stage, we will have:

**Lemma H.3.** *Suppose $\mathrm{polylog}(P) \gg \log(\frac{1}{\epsilon})$, then Induction Hypothesis H.2 holds for all $T_{c,1} < t \leq T_{c,1}^{\epsilon} = T_{c,1} + O\left(\frac{\log(P\epsilon^{-1})}{\eta\epsilon}\right)$, and at iteration $t = T_{c,1}^{\epsilon} + 1$, we have*

1. $\widetilde{\mathcal{L}}_{k,\mathbf{p}}(Q^{T_{c,1}^{\epsilon}+1}) < \frac{\epsilon}{2K}$;

2. *If $\mathsf{M} \in \mathcal{E}_{k,1}$, we have $\left(1 - \mathbf{Attn}_{\mathbf{p}\to\mathcal{P}_{k,1}}^{(T_{c,1}^{\epsilon}+1)}\right)^2 \leq O(\epsilon)$.*

# I   PROOF OF MAIN THEOREMS FOR MAE

## I.1   PROOF OF INDUCTION HYPOTHESES

We are now ready to show Induction Hypothesis E.1 (resp. Induction Hypothesis E.2) holds through the learning process.

**Theorem I.1** (Positive Information Gap). *For sufficiently large $P > 0$, $\eta \ll \log(P)$, $\Omega(1) \leq \Delta < 1$, Induction Hypothesis E.1 holds for all iterations $t = 0, 1, \cdots, T = O\left(\frac{e^{\mathrm{polylog}(P)}}{\eta}\right)$.*

**Theorem I.2** (Negative Information Gap). *For sufficiently large $P > 0$, $\eta \ll \log(P)$, $-0.5 < \Delta \leq -\Omega(1)$, Induction Hypothesis E.2 holds for all iterations $t = 0, 1, \cdots, T = O\left(\frac{e^{\mathrm{polylog}(P)}}{\eta}\right)$.*

**Proof of Theorem I.1.** It is easy to verify Induction Hypothesis E.1 holds at iteration $t = 0$ due to the initialization $Q^{(0)} = \mathbf{0}_{d \times d}$. At iteration $t > 0$:

- Induction Hypothesis E.1a. can be proven by Induction Hypothesis F.1-F.4 a and Induction Hypothesis H.1-H.2 a, combining with the fact that $\log(1/\epsilon) \ll \mathrm{polylog}(P)$.

- Induction Hypothesis E.1b. can be obtained by invoking Induction Hypothesis F.1-F.4 b.

- Induction Hypothesis E.1c. can be obtained by invoking Induction Hypothesis F.1-F.4 c and Induction Hypothesis H.1-H.2 b.

- To prove Induction Hypothesis E.1d., for $\mathbf{q} \neq \mathbf{p}$, $\Upsilon_{\mathbf{p} \to \mathbf{q}}^{(t)} = \sum_{k=1}^{K} \Upsilon_{k, \mathbf{p} \to \mathbf{q}}^{(t)}$. By item d-f in Induction Hypothesis F.1-F.4 and item c-d in Induction Hypothesis H.1-H.2, we can conclude that no matter the relative areas $\mathbf{q}$ and $\mathbf{p}$ belong to for a specific cluster, for all $k \in [K]$, throughout the entire learning process, the following upper bound always holds:

$$\Upsilon_{k, \mathbf{p} \to \mathbf{q}}^{(t)} \leq \max_{t \in [T]}(|\Phi_{\mathbf{p} \to v_{k,n}}^{(t)}| + |\Phi_{\mathbf{p} \to v_{k,1}}^{(t)}|) \max\left\{O\left(\frac{1}{C_1}\right), O\left(\frac{1}{C_n}\right), O\left(\frac{1}{P}\right)\right\} \leq \widetilde{O}\left(\frac{1}{C_n}\right).$$

  Moreover, since $K = O(\mathrm{polylog}(P))$, we then have $\Upsilon_{\mathbf{p} \to \mathbf{q}}^{(t)} = \widetilde{O}(\frac{1}{C_n})$, which completes the proof.

- The proof for Induction Hypothesis E.1d. is similar as before, by noticing that $\Upsilon_{k, \mathbf{p} \to \mathbf{p}}^{(t)} = \widetilde{O}(\frac{1}{P})$ for each $k \in [K]$, which is due to Induction Hypothesis F.1-F.4 d and Induction Hypothesis H.1-H.2 c.

The proof of Theorem I.2 mirrors that of Theorem I.1, with the only difference being the substitution of relevant sections with Induction Hypothesis E.2. For the sake of brevity, this part of the proof is not reiterated here.

## I.2   PROOF OF THEOREM 4.1 AND THEOREM 4.2 WITH POSITIVE INFORMATION GAP

**Theorem I.3.** *Suppose $\Omega(1) \leq \Delta \leq 1$. For any $0 < \epsilon < 1$, suppose $\mathrm{polylog}(P) \gg \log(\frac{1}{\epsilon})$. We apply GD to train the loss function given in equation 2.8 with $\eta \ll \mathrm{poly}(P)$. Then for each $\mathbf{p} \in \mathcal{P}$, we have*

1. *The loss converges: after $T^{\star} = O\left(\frac{\log(P)P^{\max\{2(\frac{U}{L}-1),1\}(1-\kappa_s)}}{\eta} + \frac{\log(P\epsilon^{-1})}{\eta\epsilon}\right)$ iterations, $\mathcal{L}_{\mathbf{p}}(Q^{(T^{\star})}) - \mathcal{L}_{\mathbf{p}}^{*} \leq \epsilon$, where $\mathcal{L}_{\mathbf{p}}^{\star}$ is the global minimum of patch-level construction loss in equation 4.2.*

2. *Attention score concentrates: given cluster $k \in [K]$, if $X_{\mathbf{p}}$ is masked, then the one-layer transformer nearly "pays all attention" to all unmasked patches in the same area $\mathcal{P}_{k,a_{k,\mathbf{p}}}$, i.e., $\left(1 - \mathbf{Attn}_{\mathbf{p} \to \mathcal{P}_{k,a_{k,\mathbf{p}}}}^{(T^{\star})}\right)^2 \leq O(\epsilon)$.*

3. **Local** *area learning feature attention correlation through* **two-phase**: *given $k \in [K]$, if $a_{k,\mathbf{p}} > 1$, then we have*

   (a) *$\Phi_{\mathbf{p} \to v_{k,1}}^{(t)}$ first quickly decrease with all other $\Phi_{\mathbf{p} \to v_{k,m}}^{(t)}$, $\Upsilon_{\mathbf{p} \to \mathbf{q}}^{(t)}$ not changing much;*

   (b) *after some point, the increase of $\Phi_{\mathbf{p} \to v_{k,a_{k,\mathbf{p}}}}^{(t)}$ takes dominance. Such $\Phi_{\mathbf{p} \to v_{k,a_{k,\mathbf{p}}}}^{(t)}$ will keep growing until convergence with all other feature and positional attention correlations nearly unchanged.*

4. **Global** *area learning feature attention correlation through* **one-phase**: *given $k \in [K]$, if $a_{k,\mathbf{p}} = 1$, throughout the training, the increase of $\Phi_{\mathbf{p} \to v_{k,1}}^{(t)}$ dominates, whereas all $A_{1,m}^{(t)}$ with $m \neq 1$ and position attention correlations remain close to 0.*

*Proof.* The first statement is obtained by letting $T^{\star} = \max\{T_2^{\epsilon}, T_{c,1}^{\epsilon}\} + 1$ in Lemma F.30 and Lemma H.3, combining wth Lemma D.9 and Lemma D.10, which lead to

$$\mathcal{L}_{\mathbf{p}}(Q^{(T^{\star})}) - \mathcal{L}_{\mathbf{p}}^{*} \leq \mathcal{L}_{\mathbf{p}}(Q^{(T^{\star})}) - \mathcal{L}_{\mathbf{p}}^{\mathrm{low}}$$

$$\leq \widetilde{\mathcal{L}}_{\mathbf{p}}(Q^{T^\star}) + O\Big( \exp\Big( -\big(c_3 P^{\kappa_c} + \mathbb{1}\big\{1 \notin \cup_{k\in[K]}\{a_{k,\mathbf{p}}\}\big\}c_4 P^{\kappa_s}\big)\Big)\Big)$$

$$\leq K \cdot \frac{\epsilon}{2K} + O\Big( \exp\Big( -\big(c_3 P^{\kappa_c} + \mathbb{1}\big\{1 \notin \cup_{k\in[K]}\{a_{k,\mathbf{p}}\}\big\}c_4 P^{\kappa_s}\big)\Big)\Big)$$

$$< \epsilon.$$

The second statement follows from Lemma F.30 and Lemma H.3. The third and fourth statements directly follow from the learning process described in Appendix F and Appendix H when Induction Hypothesis E.1 holds. □

### I.3 PROOF OF THEOREM 4.1 AND THEOREM 4.2 WITH NEGATIVE INFORMATION GAP

**Theorem I.4.** *Suppose* $-0.5 \leq \Delta \leq \Omega(1)$. *For any* $0 < \epsilon < 1$, *suppose* $\mathrm{polylog}(P) \gg \log(\frac{1}{\epsilon})$. *We apply GD to train the loss function given in equation 2.8 with* $\eta \ll \mathrm{poly}(P)$. *Then for each* $\mathbf{p} \in \mathcal{P}$, *we have*

1. *The loss converges: after* $T^\star = O\Big( \frac{\log(P)P^{\max\{2(\frac{U}{L}-1),1\}(1-\kappa_s)}}{\eta} + \frac{\log(P\epsilon^{-1})}{\eta\epsilon} \Big)$ *iterations,* $\mathcal{L}_{\mathbf{p}}(Q^{(T^\star)}) - \mathcal{L}_{\mathbf{p}}^* \leq \epsilon$, *where* $\mathcal{L}_{\mathbf{p}}^\star$ *is the global minimum of patch-level construction loss in equation 4.2.*

2. *Attention score concentrates: given cluster* $k \in [K]$, *if* $X_{\mathbf{p}}$ *is masked, then the one-layer transformer nearly "pays all attention" to all unmasked patches in the same area* $\mathcal{P}_{k,a_{k,\mathbf{p}}}$, *i.e.,* $\Big(1 - \mathbf{Attn}_{\mathbf{p}\to\mathcal{P}_{k,a_{k,\mathbf{p}}}}^{(T^\star)}\Big)^2 \leq O(\epsilon)$.

3. **All** *areas learning feature attention correlation through* **one-phase**: *given* $k \in [K]$, *throughout the training, the increase of* $\Phi_{\mathbf{p}\to v_{k,a_{k,\mathbf{p}}}}^{(t)}$ *dominates, whereas all* $\Phi_{\mathbf{p}\to v_{k,m}}^{(t)}$ *with* $m \neq 1$ *and position attention correlations* $\Upsilon_{\mathbf{p}\to\mathbf{q}}^{(t)}$ *remain close to* 0.

*Proof.* The first statement is obtained by letting $T^\star = \max\{T_{\mathrm{neg},1}^\epsilon, T_{c,1}^\epsilon\} + 1$ in Lemma G.3 and Lemma H.3, combining wth Lemma D.9 and Lemma D.10, which lead to

$$\mathcal{L}_{\mathbf{p}}(Q^{(T^\star)}) - \mathcal{L}_{\mathbf{p}}^* \leq \mathcal{L}_{\mathbf{p}}(Q^{(T^\star)}) - \mathcal{L}_{\mathbf{p}}^{\mathrm{low}}$$

$$\leq \widetilde{\mathcal{L}}_{\mathbf{p}}(Q^{T^\star}) + O\Big( \exp\Big( -\big(c_3 P^{\kappa_c} + \mathbb{1}\big\{1 \notin \cup_{k\in[K]}\{a_{k,\mathbf{p}}\}\big\}c_4 P^{\kappa_s}\big)\Big)\Big)$$

$$\leq K \cdot \frac{\epsilon}{2K} + O\Big( \exp\Big( -\big(c_3 P^{\kappa_c} + \mathbb{1}\big\{1 \notin \cup_{k\in[K]}\{a_{k,\mathbf{p}}\}\big\}c_4 P^{\kappa_s}\big)\Big)\Big)$$

$$< \epsilon.$$

The second statement follows from Lemma G.3 and Lemma H.3. The third and fourth statements directly follow from the learning process described in Appendix G and Appendix H when Induction Hypothesis E.2 holds. □

## J   PROOF OF MAIN THEOREMS IN CONTRASTIVE LEARNING

**Notations.** Throughout this section, we abbreviate $\mathbf{attn}_{\mathbf{p}\to\mathbf{q}}(X; Q^{(t)})$ as $\mathbf{attn}_{\mathbf{p}\to\mathbf{q}}^{(t)}(X)$. We also write $F^{\mathrm{cl}}$ as $F$ and $\mathcal{L}_{\mathrm{cl}}$ as $\mathcal{L}$ for simplicity. We abbreviate $\mathbf{Attn}_{\mathbf{p}\to\mathcal{P}_{k,m}}^{\mathrm{c}}(X; Q^{(t)})$ ($\mathbf{attn}_{\mathbf{p}\to\mathbf{q}}^{\mathrm{c}}(X; Q^{(t)})$) as $\mathbf{Attn}_{\mathbf{p}\to\mathcal{P}_{k,m}}^{(t)}$ ($\mathbf{attn}_{\mathbf{p}\to\mathbf{q}}^{(t)}$), when the context makes it clear. Furthermore, we denote

$$\ell_p(X, \mathfrak{B}) := \frac{e^{\mathsf{Sim}_F(X^+, X^{++})/\tau}}{\sum_{X\in\mathfrak{B}} e^{\mathsf{Sim}_F(X^+, X)/\tau}}, \quad \ell_s(X, \mathfrak{B}) := \frac{e^{\mathsf{Sim}_F(X^+, X^{-,s})/\tau}}{\sum_{X\in\mathfrak{B}} e^{\mathsf{Sim}_F(X^+, X)/\tau}}.$$

**Theorem J.1** (Learning with contrastive objective). *Suppose the information gap* $\Delta \in [-0.5, -\Omega(1)] \cup [\Omega(1), 1]$. *We train the ViTs in Def. 2.6 by GD to minimize (2.11) with* $\eta \ll \mathrm{poly}(P)$, $\sigma_0^2 = \frac{1}{d}$, $\tau = O(\frac{1}{\log d})$, *after* $T^\star = O(\frac{\mathrm{poly}(P)\log P}{\eta})$ *iterations, we have*

1. *Objective converges:* $\mathcal{L}_{cl}(Q^{(T^\star)}) \le \mathcal{L}^\star_{cl} + \frac{1}{\text{poly}(P)}$, *where* $\mathcal{L}^\star_{cl}$ *is the global minimum of the contrastive objective in* (2.11).

2. *Attention concentration on* **global** *area : given* $X \in \mathcal{D}^{cl}_k$ *with* $k \in [K]$, *for any* $\mathbf{p} \in \mathcal{P}$, *with high probability, we have* $1 - \mathbf{Attn}_{\mathbf{p} \to \mathcal{P}_{k,1}}(X'; Q^{(T^\star)}) = o(1)$ *for* $X' \in \{X^+, X^{++}\}$.

3. *All patches learn global FP correlation: given* $k \in [K]$, *for any* $\mathbf{p} \in \mathcal{P}$, $t \in [0, T^\star]$, $\Phi^{(t)}_{\mathbf{p} \to v_{k,1}} \gg \Phi^{(t)}_{\mathbf{p} \to v_{k,m}}$ *with* $m > 1$, *and at the convergence,* $\Phi^{(T^\star)}_{\mathbf{p} \to v_{k,1}} = \Theta(\log P)$, $\Phi^{(T^\star)}_{\mathbf{p} \to v_{k,m}} = o(1)$.

In the following, we will sketch the proof of the above theorem. Indeed, the roadmap of the analysis is similar to the masked reconstruction loss by using the induction argument, where the key difference is the properties for the gradient of the contrastive objective.

## J.1 PRELIMINARIES

In the following, we denote the contrastive loss without regularization as

$$\overline{\mathcal{L}}(Q) \triangleq \mathbb{E}_{X^+, X^{++}, \mathfrak{N}} \left[ -\tau \log \left( \frac{e^{\mathsf{Sim}_{F^{cl}}(X^+, X^{++})/\tau}}{\sum_{X' \in \mathfrak{B}} e^{\mathsf{Sim}\, F^{cl}(X^+, X')/\tau}} \right) \right].$$

**Lemma J.2** (feature gradient of contrastive loss)**.** *Given* $k \in [K]$, *for* $\mathbf{p} \in \mathcal{P}$, *let* $\widetilde{\alpha}^{(t)}_{\mathbf{p} \to v_{k,m}} := \frac{1}{\eta}\left(\Phi^{(t+1)}_{\mathbf{p} \to v_{k,m}} - \Phi^{(t)}_{\mathbf{p} \to v_{k,m}}\right)$ *for* $m \in [N_k]$, *then*

$$\widetilde{\alpha}^{(t)}_{\mathbf{p} \to v_{k,m}} = e_{\mathbf{p}}^\top \left( - \frac{\partial \mathcal{L}}{\partial Q}(Q^{(t)})\right)v_{k,m} = \alpha^{(t)}_{\mathbf{p} \to v_{k,m}} - \lambda \Phi^{(t)}_{\mathbf{p} \to v_{k,m}},$$

*where*

$$\alpha^{(t)}_{\mathbf{p} \to v_{k,m}} = e_{\mathbf{p}}^\top \left( - \frac{\partial \overline{\mathcal{L}}}{\partial Q}(Q^{(t)})\right)v_{k,m}$$

$$= \frac{1}{P}\mathbb{E}\Bigg[ \sum_{\mathbf{q} \in \mathcal{P}} \mathbf{attn}^{(t)}_{\mathbf{p} \to \mathbf{q}}(X^+)X_{\mathbf{q}}^{+\top} \Big( F(X^{++}; Q^{(t)}) - \sum_{X' \in \mathfrak{B}} \frac{e^{\mathsf{Sim}_F(X^+, X')/\tau}}{\sum_{X' \in \mathfrak{B}} e^{\mathsf{Sim}_F(X^+, X')/\tau}} F(X'; Q^{(t)})\Big)$$

$$\cdot \Big[X_{\mathbf{q}}^+ - \sum_{\mathbf{r}} \mathbf{attn}^{(t)}_{\mathbf{p} \to \mathbf{r}}(X^+)X_{\mathbf{r}}^+\Big]^\top v_{k,m}\Bigg].$$

*Proof.* Notice that

$$-\frac{\partial \mathcal{L}}{\partial Q} = -\frac{\partial \overline{\mathcal{L}}}{\partial Q} + \lambda Q.$$

Then for $-\frac{\partial \overline{\mathcal{L}}}{\partial Q}$, we begin with the chain rule and obtain

$$-\frac{\partial \overline{\mathcal{L}}}{\partial Q}$$

$$= \mathbb{E}\left[\frac{\partial}{\partial Q}\Big( \mathsf{Sim}_F(X^+, X^{++}) - \tau \log \Big( \sum_{X' \in \mathfrak{B}} e^{\mathsf{Sim}_F(X^+, X')/\tau}\Big)\Big)\right]$$

$$= \mathbb{E}\left[\frac{\partial F(X^+; Q)}{\partial Q}\Big( F(X^{++}; Q) - \sum_{X' \in \mathfrak{B}} \frac{e^{\mathsf{Sim}_F(X^+, X')/\tau}}{\sum_{X' \in \mathfrak{B}} e^{\mathsf{Sim}_F(X^+, X')/\tau}} F(X'; Q)\Big)\right]$$

$$= \frac{1}{P}\mathbb{E}\left[\sum_{\mathbf{p}, \mathbf{q} \in \mathcal{P}} \frac{\partial \mathbf{attn}_{\mathbf{p} \to \mathbf{q}}(X^+)}{\partial Q}X_{\mathbf{q}}^{+\top}\Big( F(X^{++}; Q) - \sum_{X' \in \mathfrak{B}} \frac{e^{\mathsf{Sim}_F(X^+, X')/\tau}}{\sum_{X' \in \mathfrak{B}} e^{\mathsf{Sim}_F(X^+, X')/\tau}} F(X'; Q)\Big)\right].$$

$$\tag{J.1}$$

We focus on the gradient for each attention score:

$$\frac{\partial \mathbf{attn}_{\mathbf{p} \to \mathbf{q}}(X^+)}{\partial Q} = \sum_{\mathbf{r}} \frac{\exp\left(e_{\mathbf{p}}^\top Q(X_{\mathbf{b}}^+ + X_{\mathbf{q}}^+)\right)}{\left(\sum_{\mathbf{r}} \exp(e_{\mathbf{p}}^\top QX_{\mathbf{r}})\right)^2} e_{\mathbf{p}}(X_{\mathbf{q}}^+ - X_{\mathbf{r}}^+)^\top$$

$$= \mathbf{attn_{p \to q}} \sum_{\mathbf{r}} \mathbf{attn_{p \to r}} e_{\mathbf{p}} (X_{\mathbf{q}}^+ - X_{\mathbf{r}}^+)^\top$$

$$= \mathbf{attn_{p \to q}}(X^+) e_{\mathbf{p}} \cdot \left[ X_{\mathbf{q}}^+ - \sum_{\mathbf{r}} \mathbf{attn_{p \to r}}(X^+) X_{\mathbf{r}}^+ \right]^\top .$$

Substituting the above equation into equation J.1, we have

$$-\frac{\partial \overline{\mathcal{L}}}{\partial Q} = \frac{1}{P} \mathbb{E} \left[ \sum_{\mathbf{p,q} \in \mathcal{P}} \mathbf{attn_{p \to q}}(X^+) X_{\mathbf{q}}^{+\top} \left( F(X^{++}; Q) - \sum_{X' \in \mathfrak{B}} \frac{e^{\mathsf{Sim}_F(X^+, X')/\tau}}{\sum_{X' \in \mathfrak{B}} e^{\mathsf{Sim}_F(X^+, X')/\tau}} F(X'; Q) \right) \right.$$
$$\left. \cdot e_{\mathbf{p}} \left[ X_{\mathbf{q}}^+ - \sum_{\mathbf{r}} \mathbf{attn_{p \to r}}(X^+) X_{\mathbf{r}}^+ \right]^\top \right].$$

Therefore,

$$\alpha_{\mathbf{p} \to v_{k,m}}^{(t)} = e_{\mathbf{p}}^\top (-\frac{\partial \overline{\mathcal{L}}}{\partial Q}) v_{k,m}$$

$$= \frac{1}{P} \mathbb{E} \left[ \sum_{\mathbf{q} \in \mathcal{P}} \mathbf{attn_{p \to q}}(X^+) X_{\mathbf{q}}^{+\top} \left( F(X^{++}; Q) - \sum_{X' \in \mathfrak{B}} \frac{e^{\mathsf{Sim}_F(X^+, X')/\tau}}{\sum_{X' \in \mathfrak{B}} e^{\mathsf{Sim}_F(X^+, X')/\tau}} F(X'; Q) \right) \right.$$
$$\left. \cdot \left[ X_{\mathbf{q}}^+ - \sum_{\mathbf{r}} \mathbf{attn_{p \to r}}(X^+) X_{\mathbf{r}}^+ \right]^\top v_{k,m} \right].$$

$\square$

We then present a high-probability event ensuring that the number of common unmasked patches in each area between positive augmented data pairs is proportional to the total number of patches in that area.

**Lemma J.3** (masking overlap). *Given a sample $X \sim \mathcal{D}^{cl}$, with propbability $1 - e^{-\Theta(P^{\kappa_s})}$ over the randomness of masking augmentation to obtain $X^+, X^{++}$, supposing $X$ belongs to the $k$-th cluster, it holds that*

$$\sum_{\mathbf{p} \in \mathcal{P}_{k,m}} \mathbf{1} \left\{ X_{\mathbf{p}}^+ \neq \mathbf{0} \right\} \mathbf{1} \left\{ X_{\mathbf{p}}^{++} \neq \mathbf{0} \right\} = \Theta(C_{k,m}), \quad \forall m \in [N_k]$$

*We denote the event that the above inequalities hold as $\mathcal{A}_{1,com}$. Similarly, we have the following event for $X^+$ and $X^{++}$ hols with high probability:*

$$\mathcal{A}_{1,+} := \left\{ \sum_{\mathbf{p} \in \mathcal{P}_{k,m}} \mathbf{1} \left\{ X_{\mathbf{p}}^+ \neq \mathbf{0} \right\} = \Theta(C_{k,m}), \forall m \in [N_k] \right\}$$

$$\mathcal{A}_{1,++} := \left\{ \sum_{\mathbf{p} \in \mathcal{P}_{k,m}} \mathbf{1} \left\{ X_{\mathbf{p}}^{++} \neq \mathbf{0} \right\} = \Theta(C_{k,m}), \forall m \in [N_k] \right\}.$$

*Proof.* The proof is similar to the analysis of Lemma D.6 by using the concentration property of hypergeometric distribution. $\square$

## J.2 INITIAL STAGE: GLOBAL CORRELATIONS EMERGE

For the training process at the initial stage, we define the stage transition time $T_1$ to be the iteration when $\Phi_{\mathbf{p} \to v_{k,1}}^{(t)} \geq (1 - \kappa_c) \log(P)$ for all $\mathbf{p} \in \mathcal{P}$ and $k \in [K]$.

We state the following induction hypothesis, which will hold throughout this stage:

**Induction Hypothesis J.1.** For each $0 \leq t \leq T_1 = O(\frac{\log(P) P^{3-2\kappa_c}}{\eta})$, $k \in [K]$, letting $\lambda = \frac{2}{P^{3-s\kappa_c} \log(P)}$ the following holds:

    a. $\Phi_{\mathbf{p} \to v_{k,1}}^{(t)}$ is monotonically increasing, and $\Phi_{\mathbf{p} \to v_{k,1}}^{(t)} \in \left[ 0, (1 - \kappa_c) \log(P) \right]$;

b. $\left|\Phi_{\mathbf{p}\to v_{k,m}}^{(t)}\right| \le O\big(\max\{P^{\kappa_s - 1}, P^{2(\kappa_s - \kappa_c)} \cdot \Phi_{\mathbf{p}\to v_{k,1}}^{(t)}\}\big)$ for $m > 1$.

**Lemma J.4** (bounding the noise correlation). *Let us define a noiseless version of the attention score and the network output as*

$$\widehat{\mathbf{attn}}_{\mathbf{p}\to\mathbf{q}}(X) := \frac{e^{e_{\mathbf{p}}^{\top} Q(X_{\mathbf{q}} - \xi_{\mathbf{q}})}}{\sum_{\mathbf{r}\in\mathcal{P}} e^{e_{\mathbf{p}}^{\top} Q(X_{\mathbf{r}} - \xi_{\mathbf{r}})}}, \quad for \ \mathbf{p},\mathbf{q}\in\mathcal{P}. \tag{J.2}$$

$$\widehat{F}(X; Q) := \frac{1}{P} \sum_{\mathbf{p},\mathbf{q}\in\mathcal{P}} \widehat{\mathbf{attn}}_{\mathbf{p}\to\mathbf{q}}(X) \cdot X_{\mathbf{q}} \quad \in \mathbb{R}^d. \tag{J.3}$$

$$\widehat{\ell}_p(X, \mathfrak{B}) := \frac{e^{\mathsf{Sim}_{\widehat{F}}\left(X^+, X^{++}\right)/\tau}}{\sum_{X\in\mathfrak{B}} e^{\mathsf{Sim}_{\widehat{F}}(X^+, X)/\tau}}, \quad \widehat{\ell}_s(X, \mathfrak{B}) := \frac{e^{\mathsf{Sim}_{\widehat{F}}\left(X^+, X^{-,s}\right)/\tau}}{\sum_{X\in\mathfrak{B}} e^{\mathsf{Sim}_{\widehat{F}}(X^+, X)/\tau}}.$$

*Then supposing Induction Hypothesis J.1 holds for $t \le T_1$, with high probability over the randomness of $X^+, X^{++}, \mathfrak{N}$, then for $X \in \mathfrak{B}$, any $\mathbf{p}, \mathbf{q} \in \mathcal{P}$, $s \in [N_c]$, it holds that*

$$\left|\widehat{\mathbf{attn}}_{\mathbf{p}\to\mathbf{q}}^{(t)}(X) - \mathbf{attn}_{\mathbf{p}\to\mathbf{q}}^{(t)}(X)\right| \le \frac{1}{\mathrm{poly}(d)};$$

$$\left\|\widehat{F}^{(t)}(X; Q) - F^{(t)}(X; Q)\right\| \le \frac{1}{\mathrm{poly}(d)};$$

$$\left|\ell_p^{(t)}(X, \mathfrak{B}) - \widehat{\ell}_p^{(t)}(X, \mathfrak{B})\right|, \left|\ell_s^{(t)}(X, \mathfrak{B}) - \widehat{\ell}_s^{(t)}(X, \mathfrak{B})\right| \le \frac{1}{\mathrm{poly}(d)}.$$

*We denote the event that the above inequalities hold as $\mathcal{A}_2$.*

*Proof.* The result follows directly from the concentration of Gaussian random variables, the boundedness of the feature vectors and the boundedness of $\|e_{\mathbf{p}}Q\|_2 \le \Phi_{k\to v_{k,m}}$ due to the Induction Hypothesis J.1 .

$\square$

**Lemma J.5** (attention score). *Suppose the Induction Hypothesis J.1 holds for $t \le T_1$, given $\{X^+, X^{++}, \mathfrak{N}\}$, assuming $X \in \mathcal{D}_k^{c1}$ with $k \in [K]$, then for $m \in [N_k]$, $\mathbf{p} \in \mathcal{P}$, we have*

1. *for $a \in \{+, ++\}$, if $X^a \in \mathcal{A}_{2,a}$, then*

   (a) $1 - \mathbf{Attn}_{\mathbf{p}\to\mathcal{P}_{k,1}}^{(t)}(X^a) \ge \Omega(1)$ *and* $\mathbf{Attn}_{\mathbf{p}\to\mathcal{P}_{k,1}}^{(t)}(X^a) \ge \Omega(\frac{1}{P^{1-\kappa_c}})$ ;

   (b) *for $m > 1$, $\mathbf{Attn}_{\mathbf{p}\to\mathcal{P}_{k,m}}^{(t)}(X^a) = \Theta\big(\frac{1 - \mathbf{Attn}_{\mathbf{p}\to\mathcal{P}_{k,1}}^{(t)}(X^a)}{P^{1-\kappa_s}}\big)$;*

2. *for $X' \in \mathfrak{N}$, we have*

   (a) $1 - \widetilde{\mathbf{Attn}}_{\mathbf{p}\to\mathcal{P}_{k,1}}^{(t)}(X') \ge \Omega(1)$ *and* $\widetilde{\mathbf{Attn}}_{\mathbf{p}\to\mathcal{P}_{k,1}}^{(t)}(X') \ge \Omega(\frac{1}{P^{1-\kappa_c}})$;

   (b) *for $m > 1$, $\widetilde{\mathbf{Attn}}_{\mathbf{p}\to\mathcal{P}_{k,m}}^{(t)}(X') = \Theta\big(\frac{1 - \widetilde{\mathbf{Attn}}_{\mathbf{p}\to\mathcal{P}_{k,1}}^{(t)}(X')}{P^{1-\kappa_s}}\big)$.*

The intuition behind this lemma is that, due to the zero initialization of $Q$, the attention scores are nearly uniform. As a result, the area attention score $\mathbf{Attn}_{\mathbf{p}\to\mathcal{P}_{k,m}}(X+)$ is proportional to the number of unmasked patches in this area. If Induction Hypothesis J.1 holds, we can easily conclude that only the area attention score for the global area will increase, while the relative relationships among the local area attention scores will be preserved.

**Lemma J.6** (logit score). *Suppose the Induction Hypothesis J.1 holds for $t \le T_1$, given $\{X^+, X^{++}, \mathfrak{N}\}$, suppose $X \in \mathcal{D}_k^{c1}$, we have*

$$1 - \ell_q^{(t)}(X, \mathfrak{B}) \ge \Omega(1), \quad \ell_q^{(t)}(X, \mathfrak{B}) \ge \Omega(\frac{1}{N_s}), \quad q \in \mathfrak{B} \cap \mathcal{D}_k^{cl}$$

$$\ell_q^{(t)}(X, \mathfrak{B}) \le O(\frac{1}{N_s}), \quad else.$$

**Lemma J.7** (feature gradient near initialization). *Suppose the Induction Hypothesis J.1 holds for $t \le T_0$, then for $t \le T_1$, given $k \in [K]$, $m \in [N_k]$, for $\mathbf{p} \in \mathcal{P}$,*

- *For the global feature $m = 1$,*

$$\alpha^{(t)}_{\mathbf{p} \to v_{k,1}} = \Theta\left(\frac{1}{P}\mathbb{E}\left[z_1(1 - \ell_p)\mathbf{Attn}_{\mathbf{p} \to \mathcal{P}_{k,1}}(X^+)\mathbf{Attn}_{\mathbf{p} \to \mathcal{P}_{k,1}}(X^{++})\right]\right)$$

- *For the local feature $m > 1$*

$$\alpha^{(t)}_{\mathbf{p} \to v_{k,m}} = \Theta\left(\frac{1}{P}\mathbb{E}\left[z_m(1 - \ell_p)\mathbf{Attn}_{\mathbf{p} \to \mathcal{P}_{k,m}}(X^+)\mathbf{Attn}_{\mathbf{p} \to \mathcal{P}_{k,m}}(X^{++})\right]\right)$$
$$+ O\left(\frac{1}{P}\mathbb{E}\left[z_m(1 - \ell_p)\mathbf{Attn}_{\mathbf{p} \to \mathcal{P}_{k,m}}(X^+)\mathbf{Attn}_{\mathbf{p} \to \mathcal{P}_{k,1}}(X^+)\mathbf{Attn}_{\mathbf{p} \to \mathcal{P}_{k,1}}(X^{++})\right]\right)$$

*Proof.*

$$\alpha^{(t)}_{\mathbf{p} \to v_{k,m}} = \frac{1}{P}\mathbb{E}\left[\sum_{\mathbf{q} \in \mathcal{P}} \mathbf{attn}_{\mathbf{p} \to \mathbf{q}}(X^+)X_{\mathbf{q}}^{+\top}\left((1 - \ell_p)F(X^{++};Q) - \sum_{s=1}^{N_c}\ell_s F(X^{-,s};Q)\right)\right.$$
$$\left. \cdot \left[X_{\mathbf{q}}^+ - \sum_{\mathbf{r}} \mathbf{attn}_{\mathbf{p} \to \mathbf{r}}(X^+)X_{\mathbf{r}}^+\right]^\top v_{k,m}(\mathbf{1}_{\mathcal{A}_1} + \mathbf{1}_{\mathcal{A}_1^c})\right]$$

$$\overset{(a)}{=} \frac{1}{P}\mathbb{E}\left[\sum_{\mathbf{q} \in \mathcal{P}} \widehat{\mathbf{attn}}_{\mathbf{p} \to \mathbf{q}}(X^+)X_{\mathbf{q}}^{+\top}\left((1 - \widehat{\ell}_p)\widehat{F}(X^{++};Q) - \sum_{s=1}^{N_c}\widehat{\ell}_s \widehat{F}(X^{-,s};Q)\right)\right.$$
$$\left. \cdot \left[X_{\mathbf{q}}^+ - \sum_{\mathbf{r}} \widehat{\mathbf{attn}}_{\mathbf{p} \to \mathbf{r}}(X^+)X_{\mathbf{r}}^+\right]^\top v_{k,m}(\mathbf{1}_{\mathcal{A}_1} + \mathbf{1}_{\mathcal{A}_1^c})\right] + \Xi_{\mathbf{p},k,m,1}$$

$$= \frac{1}{P}\mathbb{E}\left[\sum_{i=1}^{N_k}\sum_{\mathbf{q} \in \mathcal{P}_{k,i} \cap \mathcal{U}^+} \widehat{\mathbf{attn}}_{\mathbf{p} \to \mathbf{q}}(X^+)(z_i v_{k,i})^\top\left((1 - \widehat{\ell}_p)\widehat{F}(X^{++};Q) - \sum_{s=1}^{N_c}\widehat{\ell}_s \widehat{F}(X^{-,s};Q)\right)\right.$$
$$\left. \cdot \left[z_i v_{k,i} - \sum_{j=1}^{N_k}\sum_{\mathbf{r} \in \mathcal{P}_{k,j} \cap \mathcal{U}^+} \widehat{\mathbf{attn}}_{\mathbf{p} \to \mathbf{r}}(X^+)z_j v_{k,j}\right]^\top v_{k,m}\right] \qquad (J_1)$$

$$+ \frac{1}{P}\mathbb{E}\left[\sum_{i=1}^{N_k}\sum_{\mathbf{q} \in \mathcal{P}_{k,i} \cap \mathcal{U}^+} \widehat{\mathbf{attn}}_{\mathbf{p} \to \mathbf{q}}(X^+)(z_i v_{k,i})^\top\left((1 - \widehat{\ell}_p)\widehat{F}(X^{++};Q) - \sum_{s=1}^{N_c}\widehat{\ell}_s \widehat{F}(X^{-,s};Q)\right)\right.$$
$$\left. \cdot \left[\xi_{\mathbf{q}} - \sum_{\mathbf{r} \in \mathcal{P} \cap \mathcal{U}^+} \widehat{\mathbf{attn}}_{\mathbf{p} \to \mathbf{r}}(X^+)\xi_{\mathbf{r}}\right]^\top v_{k,m}\right] \qquad (J_2)$$

$$+ \frac{1}{P}\mathbb{E}\left[\sum_{\mathbf{q} \in \mathcal{P} \cap \mathcal{U}^+} \widehat{\mathbf{attn}}_{\mathbf{p} \to \mathbf{q}}(X^+)\xi_{\mathbf{q}}^\top\left((1 - \widehat{\ell}_p)\widehat{F}(X^{++};Q) - \sum_{s=1}^{N_c}\widehat{\ell}_s \widehat{F}(X^{-,s};Q)\right)\right.$$
$$\left. \cdot \left[z_i v_{k,i} - \sum_{j=1}^{N_k}\sum_{\mathbf{r} \in \mathcal{P}_{k,j} \cap \mathcal{U}^+} \widehat{\mathbf{attn}}_{\mathbf{p} \to \mathbf{r}}(X^+)z_j v_{k,j}\right]^\top v_{k,m}\right] \qquad (J_3)$$

$$+ \frac{1}{P}\mathbb{E}\left[\sum_{\mathbf{q} \in \mathcal{P} \cap \mathcal{U}^+} \widehat{\mathbf{attn}}_{\mathbf{p} \to \mathbf{q}}(X^+)\xi_{\mathbf{q}}^\top\left((1 - \widehat{\ell}_p)\widehat{F}(X^{++};Q) - \sum_{s=1}^{N_c}\widehat{\ell}_s \widehat{F}(X^{-,s};Q)\right)\right.$$
$$\left. \cdot \left[\xi_{\mathbf{q}} - \sum_{\mathbf{r} \in \mathcal{P} \cap \mathcal{U}^+} \widehat{\mathbf{attn}}_{\mathbf{p} \to \mathbf{r}}(X^+)\xi_{\mathbf{r}}\right]^\top v_{k,m}\right] \qquad (J_4)$$

$$+ \Xi_{\mathbf{p},k,m,1}$$

where $(a)$ is bounded by Lemma J.4 with error up to $\Xi_{\mathbf{p},k,m,1} \le \frac{1}{\text{poly}(d)}$, $\mathcal{U}^+$ is the set of masked patches for $X^+$. We first look at the term $J_1$, notice that $\xi_{\mathbf{q}}$ is the random Gaussian noise with zero mean, and is independent of $\widehat{\mathbf{attn}}$ and $\widehat{\ell}$, we then have

$$
\begin{aligned}
J_4 &= \frac{1}{P^2}\mathbb{E}\Bigg[ \sum_{\mathbf{q}\in\mathcal{P}_{k,m}\cap\mathcal{U}^+} \widehat{\mathbf{attn}}_{\mathbf{p}\to\mathbf{q}}(X^+)\xi_{\mathbf{q}}^\top\Big((1-\widehat{\ell}_p)\sum_{\mathbf{p}'\in\mathcal{P}}\sum_{\mathbf{r}\in\mathcal{P}_{k,m}\cap\mathcal{U}^{++}}\widehat{\mathbf{attn}}_{\mathbf{p}'\to\mathbf{r}}(X^{++})z_m v_{k,m}\Big) \\
&\qquad\qquad \cdot\Big[\xi_{\mathbf{q}} - \sum_{\mathbf{r}\in\mathcal{P}\cap\mathcal{U}^+}\widehat{\mathbf{attn}}_{\mathbf{p}\to\mathbf{r}}(X^+)\xi_{\mathbf{r}}\Big]^\top v_{k,m}\Bigg] \\
&\quad + \frac{1}{P^2}\mathbb{E}\Bigg[\sum_{\mathbf{q}\in\mathcal{P}_{k,m}\cap\mathcal{U}^+}\widehat{\mathbf{attn}}_{\mathbf{p}\to\mathbf{q}}(X^+)\xi_{\mathbf{q}}^\top\Big((1-\widehat{\ell}_p)\sum_{\mathbf{p}'\in\mathcal{P}}\sum_{\mathbf{r}\in\mathcal{U}^{++}}\widehat{\mathbf{attn}}_{\mathbf{p}'\to\mathbf{r}}(X^{++})\xi_{\mathbf{r}}\Big) \\
&\qquad\qquad \cdot\Big[\xi_{\mathbf{q}} - \sum_{\mathbf{r}\in\mathcal{P}\cap\mathcal{U}^+}\widehat{\mathbf{attn}}_{\mathbf{p}\to\mathbf{r}}(X^+)\xi_{\mathbf{r}}\Big]^\top v_{k,m}\Bigg] \\
&= \frac{1}{P^2}\mathbb{E}\Bigg[z_m\sum_{\mathbf{q}\in\mathcal{P}_{k,m}\cap\mathcal{U}^+}\widehat{\mathbf{attn}}_{\mathbf{p}\to\mathbf{q}}(X^+)\Big((1-\widehat{\ell}_p)\sum_{\mathbf{p}'\in\mathcal{P}}\sum_{\mathbf{r}\in\mathcal{P}_{k,m}\cap\mathcal{U}^{++}}\widehat{\mathbf{attn}}_{\mathbf{p}'\to\mathbf{r}}(X^{++})\Big)\Big[1-\widehat{\mathbf{attn}}_{\mathbf{p}\to\mathbf{q}}(X^+)\Big]\Bigg] \\
&= \frac{1}{P^2}\mathbb{E}\Bigg[z_m\Big[\mathbf{Attn}_{\mathbf{p}\to\mathcal{P}_{k,m}}(X^+) - \sum_{\mathbf{q}\in\mathcal{P}_{k,m}\cap\mathcal{U}^+}\mathbf{attn}^2_{\mathbf{p}\to\mathbf{q}}(X^+)\Big]\Big((1-\ell_p)\sum_{\mathbf{p}'\in\mathcal{P}}\mathbf{Attn}_{\mathbf{p}'\to\mathcal{P}_{k,m}}(X^{++})\Big)\Bigg] + \Xi_{\mathbf{p},k,m,2} \\
&= \Theta\Bigg(\frac{1}{P}\mathbb{E}\Big[z_m(1-\ell_p)\mathbf{Attn}_{\mathbf{p}\to\mathcal{P}_{k,m}}(X^+)\mathbf{Attn}_{\mathbf{p}\to\mathcal{P}_{k,m}}(X^{++})\Big]\Bigg)
\end{aligned}
$$

where $(a)$ is bounded by invoking Lemma J.4 with error up to $\Xi_{\mathbf{p},k,m,2} \le \frac{1}{\text{poly}(d)}$, and the last equality is due to Lemma J.5.

$$
\begin{aligned}
J_2 &= \frac{1}{P}\mathbb{E}\Bigg[\mathbb{E}\Big[\sum_{i=1}^{N_k}\sum_{\mathbf{q}\in\mathcal{P}_{k,i}\cap\mathcal{U}^+}\widehat{\mathbf{attn}}_{\mathbf{p}\to\mathbf{q}}(X^+)(z_i v_{k,i})^\top(1-\widehat{\ell}_p)\widehat{F}(X^{++};Q) \\
&\qquad\qquad \cdot\Big[\xi_{\mathbf{q}} - \sum_{\mathbf{r}\in\mathcal{P}\cap\mathcal{U}^+}\widehat{\mathbf{attn}}_{\mathbf{p}\to\mathbf{r}}(X^+)\xi_{\mathbf{r}}\Big]^\top v_{k,m}\Big|\xi\Big]\Bigg] \\
&= \frac{1}{P^2}\mathbb{E}\Bigg[\mathbb{E}\Big[\sum_{i=1}^{N_k}\sum_{\mathbf{q}\in\mathcal{P}_{k,i}\cap\mathcal{U}^+}\widehat{\mathbf{attn}}_{\mathbf{p}\to\mathbf{q}}(X^+)(z_i v_{k,i})^\top(1-\widehat{\ell}_p)\sum_{\mathbf{p}'\in\mathcal{P},\mathbf{r}\in\mathcal{P}\cap\mathcal{U}^{++}}\widehat{\mathbf{attn}}_{\mathbf{p}'\to\mathbf{r}}(X^{++})\xi_{\mathbf{r}} \\
&\qquad\qquad \cdot\Big[\xi_{\mathbf{q}} - \sum_{\mathbf{r}\in\mathcal{P}\cap\mathcal{U}^+}\widehat{\mathbf{attn}}_{\mathbf{p}\to\mathbf{r}}(X^+)\xi_{\mathbf{r}}\Big]^\top v_{k,m}\Big|\xi\Big]\Bigg] \\
&= \frac{1}{P^2}\mathbb{E}\Bigg[\mathbb{E}\Big[\sum_{\mathbf{q}\in\mathcal{P}_{k,m}\cap\mathcal{U}^+}\widehat{\mathbf{attn}}_{\mathbf{p}\to\mathbf{q}}(X^+)(z_m v_{k,m})^\top(1-\widehat{\ell}_p)\sum_{\mathbf{p}'\in\mathcal{P},\mathbf{r}\in\mathcal{P}\cap\mathcal{U}^{++}}\widehat{\mathbf{attn}}_{\mathbf{p}'\to\mathbf{r}}(X^{++})\xi_{\mathbf{r}} \\
&\qquad\qquad \cdot\Big[\xi_{\mathbf{q}} - \sum_{\mathbf{r}\in\mathcal{P}\cap\mathcal{U}^+}\widehat{\mathbf{attn}}_{\mathbf{p}\to\mathbf{r}}(X^+)\xi_{\mathbf{r}}\Big]^\top v_{k,m}\Big|\xi\Big]\Bigg] \\
&= \frac{1}{P^2}\mathbb{E}\Bigg[z_m\sum_{\mathbf{q}\in\mathcal{P}_{k,m}\cap\mathcal{U}^+}\widehat{\mathbf{attn}}_{\mathbf{p}\to\mathbf{q}}(X^+)(1-\widehat{\ell}_p) \\
&\qquad \cdot\Big(\widehat{\mathbf{attn}}_{\mathbf{p}\to\mathbf{q}}(X^+)\mathbf{1}_{\mathbf{q}\in\mathcal{U}^{++}} - \sum_{\mathbf{p}'\in\mathcal{P},\mathbf{r}\in\mathcal{P}\cap\mathcal{U}^{++}\cap\mathcal{U}^+}\widehat{\mathbf{attn}}_{\mathbf{p}'\to\mathbf{r}}(X^{++})\widehat{\mathbf{attn}}_{\mathbf{p}\to\mathbf{r}}(X^+)\Big)\Bigg] \\
&= \frac{1}{P^2}\mathbb{E}\Bigg[z_m\sum_{\mathbf{q}\in\mathcal{P}_{k,m}\cap\mathcal{U}^+}\mathbf{attn}_{\mathbf{p}\to\mathbf{q}}(X^+)(1-\ell_p) \\
&\qquad \cdot\Big(\mathbf{attn}_{\mathbf{p}\to\mathbf{q}}(X^+)\mathbf{1}_{\mathbf{q}\in\mathcal{U}^{++}} - \sum_{\mathbf{p}'\in\mathcal{P},\mathbf{r}\in\mathcal{P}\cap\mathcal{U}^{++}\cap\mathcal{U}^+}\mathbf{attn}_{\mathbf{p}'\to\mathbf{r}}(X^{++})\mathbf{attn}_{\mathbf{p}\to\mathbf{r}}(X^+)\Big)\Bigg] + \Xi_{\mathbf{p},k,m,3}
\end{aligned}
$$

Thus, by invoking Lemma J.5, we have

$$|J_2| \le O\Big( \frac{1}{P} \mathbb{E}\Big[ z_m \sum_{\mathbf{q} \in \mathcal{P}_{k,m} \cap \mathcal{U}^+} \mathbf{attn}_{\mathbf{p} \to \mathbf{q}}(X^+)(1 - \ell_p) \cdot \Big( \max_{\mathbf{r} \in \mathcal{P} \cap \mathcal{U}^{++} \cap \mathcal{U}^+} \mathbf{attn}_{\mathbf{p} \to \mathbf{r}}(X^+)) \Big] \Big) + \Xi_{\mathbf{p},k,m,3}$$

$$\le O\Big( \frac{1}{P \cdot C_{k,1}} \mathbb{E}\Big[ z_m(1 - \ell_p) \mathbf{Attn}_{\mathbf{p} \to \mathcal{P}_{k,m}}(X^+) \cdot \mathbf{Attn}_{\mathbf{p} \to \mathcal{P}_{k,1}}(X^{++}) \Big] \Big)$$

$$J_3 = \frac{1}{P} \mathbb{E}\Big[ \sum_{\mathbf{q} \in \mathcal{P} \cap \mathcal{U}^+} \widehat{\mathbf{attn}}_{\mathbf{p} \to \mathbf{q}}(X^+) \xi_{\mathbf{q}}^\top \Big( (1 - \widehat{\ell}_p) \widehat{F}(X^{++}; Q) - \sum_{s=1}^n \widehat{\ell}_s \widehat{F}(X^{-,s}; Q) \Big)$$

$$\cdot \Big[ z_i v_{k,i} - \sum_{j=1}^{N_k} \sum_{\mathbf{r} \in \mathcal{P}_{k,j} \cap \mathcal{U}^+} \widehat{\mathbf{attn}}_{\mathbf{p} \to \mathbf{r}}(X^+) z_j v_{k,j} \Big]^\top v_{k,m} \Big]$$

$$= \frac{1}{P} \mathbb{E}\Big[ \mathbb{E}\Big[ \sum_{\mathbf{q} \in \mathcal{P}_{k,m} \cap \mathcal{U}^+} \widehat{\mathbf{attn}}_{\mathbf{p} \to \mathbf{q}}(X^+) \xi_{\mathbf{q}}^\top \Big( (1 - \widehat{\ell}_p) \widehat{F}(X^{++}; Q) \Big)$$

$$\cdot z_m \Big[ 1 - \sum_{\mathbf{r} \in \mathcal{P}_{k,m} \cap \mathcal{U}^+} \widehat{\mathbf{attn}}_{\mathbf{p} \to \mathbf{r}}(X^+) \Big] \Big| \xi \Big] \Big]$$

$$= \frac{1}{P^2} \mathbb{E}\Big[ z_m \sum_{\mathbf{q} \in \mathcal{P}_{k,m} \cap \mathcal{U}^+ \cap \mathcal{U}^{++}} \widehat{\mathbf{attn}}_{\mathbf{p} \to \mathbf{q}}(X^+)(1 - \widehat{\ell}_p)$$

$$\cdot \sum_{\mathbf{p}' \in \mathcal{P}} \widehat{\mathbf{attn}}_{\mathbf{p}' \to \mathbf{q}}(X^{++}) \Big[ 1 - \sum_{\mathbf{r} \in \mathcal{P}_{k,m} \cap \mathcal{U}^+} \widehat{\mathbf{attn}}_{\mathbf{p} \to \mathbf{r}}(X^+) \Big] \Big]$$

$$= \frac{1}{P^2} \mathbb{E}\Big[ z_m \sum_{\mathbf{q} \in \mathcal{P}_{k,m} \cap \mathcal{U}^+ \cap \mathcal{U}^{++}} \mathbf{attn}_{\mathbf{p} \to \mathbf{q}}(X^+)(1 - \ell_p)$$

$$\cdot \sum_{\mathbf{p}' \in \mathcal{P}} \mathbf{attn}_{\mathbf{p}' \to \mathbf{q}}(X^{++}) \Big[ 1 - \mathbf{Attn}_{\mathbf{p} \to \mathcal{P}_{k,m}}(X^+) \Big] \Big] + \Xi_{\mathbf{p},k,m,4}$$

$$\le O\Big( \frac{1}{P^2} \mathbb{E}\Big[ z_m \mathbf{Attn}_{\mathbf{p} \to \mathcal{P}_{k,m}}(X^+) \Big( 1 - \mathbf{Attn}_{\mathbf{p} \to \mathcal{P}_{k,m}}(X^+) \Big)(1 - \ell_p) \cdot \sum_{\mathbf{p}' \in \mathcal{P}} O(\frac{1}{C_{k,m}}) \cdot \mathbf{Attn}_{\mathbf{p}' \to \mathcal{P}_{k,m}}(X^{++}) \Big] \Big)$$

$$\le O\Big( \frac{J_4}{C_{k,m}} \Big).$$

where the last inequality is due to Lemma J.5.

$$J_1 = \frac{1}{P^2} \mathbb{E}\Big[ \sum_{i=1}^{N_k} \sum_{\mathbf{q} \in \mathcal{P}_{k,i} \cap \mathcal{U}^+} \widehat{\mathbf{attn}}_{\mathbf{p} \to \mathbf{q}}(X^+)(z_i v_{k,i})^\top$$

$$\cdot \Big( (1 - \widehat{\ell}_p) \sum_{p' \in \mathcal{P}} \sum_{j=1}^{N_k} \sum_{\mathbf{q}' \in \mathcal{P}_{k,j} \cap \mathcal{U}^{++}} \widehat{\mathbf{attn}}_{\mathbf{p}' \to \mathbf{q}'}(X^{+,+}) z_j v_{k,j}$$

$$- \sum_{X^{-,s} \in \mathfrak{N} \cap \mathcal{D}_k^{cl}} \widehat{\ell}_s \sum_{p' \in \mathcal{P}} \sum_{j=1}^{N_k} \sum_{\mathbf{q}' \in \mathcal{P}_{k,j}} \widehat{\mathbf{attn}}_{\mathbf{p}' \to \mathbf{q}'}(X^{-,s}) z_{s,j} v_{k,j} \Big)$$

$$\cdot \Big[ z_i v_{k,i} - \sum_{j=1}^{N_k} \sum_{\mathbf{r} \in \mathcal{P}_{k,j} \cap \mathcal{U}^+} \widehat{\mathbf{attn}}_{\mathbf{p} \to \mathbf{r}} z_j v_{k,j} \Big]^\top v_{k,m} \Big]$$

$$= \frac{1}{P^2} \mathbb{E}\Big[ \sum_{i=1}^{N_k} \sum_{\mathbf{q} \in \mathcal{P}_{k,i} \cap \mathcal{U}^+} \mathbf{attn}_{\mathbf{p} \to \mathbf{q}}(X^+)(z_i v_{k,i})^\top$$

$$\cdot \left( (1-\ell_p) \sum_{p' \in \mathcal{P}} \sum_{j=1}^{N_k} \sum_{\mathbf{q}' \in \mathcal{P}_{k,j} \cap \mathcal{U}^{++}} \mathbf{attn}_{\mathbf{p}' \to \mathbf{q}'}(X^{+,+}) z_j v_{k,j} \right.$$

$$\left. - \sum_{X^{-,s} \in \mathfrak{N} \cap \mathcal{D}_k^{cl}} \ell_s \sum_{p' \in \mathcal{P}} \sum_{j=1}^{N_k} \sum_{\mathbf{q}' \in \mathcal{P}_{k,j}} \mathbf{attn}_{\mathbf{p}' \to \mathbf{q}'}(X^{-,s}) z_{s,j} v_{k,j} \right)$$

$$\cdot \left[ z_i v_{k,i} - \sum_{j=1}^{N_k} \sum_{\mathbf{r} \in \mathcal{P}_{k,j} \cap \mathcal{U}^+} \mathbf{attn}_{\mathbf{p} \to \mathbf{r}} z_j v_{k,j} \right]^\top v_{k,m} \right] + \Xi_{\mathbf{p},k,m,5}$$

$$= \frac{1}{P^2} \mathbb{E}\left[ \mathbf{Attn}_{\mathbf{p} \to \mathcal{P}_{k,m}}(X^+) \left( z_m^2 \left( 1 - \mathbf{Attn}_{\mathbf{p} \to \mathcal{P}_{k,m}}(X^+) \right) v_{k,m} - \sum_{i \neq m} z_m z_i \mathbf{Attn}_{\mathbf{p} \to \mathcal{P}_{k,i}}(X^+) v_{k,i} \right)^\top \right.$$

$$\cdot \left( (1-\ell_p) \sum_{\mathbf{p}' \in \mathcal{P}} \sum_{j=1}^{N_k} \sum_{\mathbf{q}' \in \mathcal{P}_{k,j} \cap \mathcal{U}^{++}} \mathbf{attn}_{\mathbf{p}' \to \mathbf{q}'}(X^{+,+}) z_j v_{k,j} \right.$$

$$\left. \left. - \sum_{X^{-,s} \in \mathfrak{N} \cap \mathcal{D}_k^{cl}} \ell_s \sum_{p' \in \mathcal{P}} \sum_{j=1}^{N_k} \sum_{\mathbf{q}' \in \mathcal{P}_{k,j}} \mathbf{attn}_{\mathbf{p}' \to \mathbf{q}'}(X^{-,s}) z_{s,j} v_{k,j} \right) \right] + \Xi_{\mathbf{p},k,m,5}$$

$$= \frac{1}{P^2} \mathbb{E}\left[ \mathbf{Attn}_{\mathbf{p} \to \mathcal{P}_{k,m}}(X^+) \left( z_m^2 \left( 1 - \mathbf{Attn}_{\mathbf{p} \to \mathcal{P}_{k,m}}(X^+) \right) v_{k,m} - \sum_{i \neq m} z_m z_i \mathbf{Attn}_{\mathbf{p} \to \mathcal{P}_{k,i}}(X^+) v_{k,i} \right)^\top \right.$$

$$\left. \cdot \left( (1-\ell_p) \sum_{\mathbf{p}' \in \mathcal{P}} \sum_{j=1}^{N_k} \mathbf{Attn}_{\mathbf{p}' \to \mathcal{P}_{k,j}}(X^{++}) z_j v_{k,j} - \sum_{X^{-,s} \in \mathfrak{N} \cap \mathcal{D}_k^{cl}} \ell_s \sum_{\mathbf{p}' \in \mathcal{P}} \sum_{j=1}^{N_k} \widetilde{\mathbf{Attn}}_{\mathbf{p}' \to \mathcal{P}_{k,j}}(X^{-,s}) z_{s,j} v_{k,j} \right) \right]$$

$$+ \Xi_{\mathbf{p},k,m,5}$$

$$= \frac{1}{P^2} \mathbb{E}\left[ \mathbf{Attn}_{\mathbf{p} \to \mathcal{P}_{k,m}}(X^+) \left( z_m^2 \left( 1 - \mathbf{Attn}_{\mathbf{p} \to \mathcal{P}_{k,m}}(X^+) \right) \right. \right.$$

$$\left. \left. \cdot \left( \sum_{\mathbf{p}' \in \mathcal{P}} \left( (1-\ell_p) z_m \mathbf{Attn}_{\mathbf{p}' \to \mathcal{P}_{k,m}}(X^{++}) - \sum_{X^{-,s} \in \mathfrak{N} \cap \mathcal{D}_k^{cl}} z_{s,m} \ell_s \widetilde{\mathbf{Attn}}_{\mathbf{p}' \to \mathcal{P}_{k,m}}(X^{-,s}) \right) \right) \right) \right]$$

$$(J_{1,1})$$

$$- \frac{1}{P^2} \mathbb{E}\left[ \mathbf{Attn}_{\mathbf{p} \to \mathcal{P}_{k,m}}(X^+) \left( \sum_{i \neq m} z_m z_i \mathbf{Attn}_{\mathbf{p} \to \mathcal{P}_{k,i}}(X^+) \right. \right.$$

$$\left. \left. \cdot \left( \sum_{\mathbf{p}' \in \mathcal{P}} \left( (1-\ell_p) z_i \mathbf{Attn}_{\mathbf{p}' \to \mathcal{P}_{k,i}}(X^{++}) - \sum_{X^{-,s} \in \mathfrak{N} \cap \mathcal{D}_k^{cl}} z_{s,i} \ell_s \widetilde{\mathbf{Attn}}_{\mathbf{p}' \to \mathcal{P}_{k,i}}(X^{-,s}) \right) \right) \right) \right]$$

$$(J_{1,2})$$

$$+ \Xi_{\mathbf{p},k,m,5}$$

Notice that $J_{1,1} = \Theta(J_4)$. Furthermore, when $m = 1$, $J_{1,2}$ is negligible compared to $J_1$, else

$$|J_{1,2}| \leq O\left( \frac{1}{P^2} \mathbb{E}\left[ \mathbf{Attn}_{\mathbf{p} \to \mathcal{P}_{k,m}}(X^+) \left( z_m z_1 \mathbf{Attn}_{\mathbf{p} \to \mathcal{P}_{k,1}}(X^+) \right. \right. \right.$$

$$\left. \left. \left. \cdot \left( \sum_{\mathbf{p}' \in \mathcal{P}} \left( (1-\ell_p) z_1 \mathbf{Attn}_{\mathbf{p}' \to \mathcal{P}_{k,1}}(X^{++}) - \sum_{X^{-,s} \in \mathfrak{N} \cap \mathcal{D}_k^{cl}} z_{s,1} \ell_s \widetilde{\mathbf{Attn}}_{\mathbf{p}' \to \mathcal{P}_{k,1}}(X^{-,s}) \right) \right) \right) \right] \right)$$

$$\leq O\left( \frac{1}{P} \mathbb{E}\left[ z_m (1-\ell_p) \mathbf{Attn}_{\mathbf{p} \to \mathcal{P}_{k,m}}(X^+) \mathbf{Attn}_{\mathbf{p} \to \mathcal{P}_{k,1}}(X^+) \mathbf{Attn}_{\mathbf{p} \to \mathcal{P}_{k,1}}(X^{++}) \right] \right)$$

Putting all the terms together, and noticed that

$$O\left( \frac{1}{P} \mathbb{E}\left[ z_m (1-\ell_p) \mathbf{Attn}_{\mathbf{p} \to \mathcal{P}_{k,m}}(X^+) \mathbf{Attn}_{\mathbf{p} \to \mathcal{P}_{k,1}}(X^+) \mathbf{Attn}_{\mathbf{p} \to \mathcal{P}_{k,1}}(X^{++}) \right] \right)$$

$$\geq O\Big(\frac{1}{P \cdot C_{k,1}}\mathbb{E}\Big[z_m(1-\ell_p)\mathbf{Attn_{p\to\mathcal{P}_{k,m}}}(X^+) \cdot \mathbf{Attn_{p\to\mathcal{P}_{k,1}}}(X^{++})\Big]\Big),$$

then we complete the proof. □

*Proof of Induction Hypothesis J.1.* By Lemma J.7 and Lemma J.6, at the initial stage of the learning process, we have

$$\alpha^{(0)}_{\mathbf{p}\to v_{k,1}} \propto \frac{1}{P}\mathbb{E}\Big[\mathbf{Attn_{p\to\mathcal{P}_{k,1}}}(X^+)\mathbf{Attn_{p\to\mathcal{P}_{k,1}}}(X^{++})\Big]$$

$$\alpha^{(0)}_{\mathbf{p}\to v_{k,1}} \lesssim \frac{1}{P}\max\Big\{\mathbb{E}\Big[\mathbf{Attn_{p\to\mathcal{P}_{k,m}}}(X^+)\mathbf{Attn_{p\to\mathcal{P}_{k,m}}}(X^{++})\Big],$$

$$\mathbb{E}\Big[\mathbf{Attn_{p\to\mathcal{P}_{k,m}}}(X^+)\mathbf{Attn_{p\to\mathcal{P}_{k,1}}}(X^+)\mathbf{Attn_{p\to\mathcal{P}_{k,1}}}(X^{++})\Big]\Big\}$$

Then by the relations of attention score in Lemma J.5, focusing on the high-propbability event $A_{1,+}$ and $A_{1,++}$ we have

$$|\alpha^{(t)}_{\mathbf{p}\to v_{k,m}}| \leq O\big(\max\{P^{\kappa_s-1}, P^{2(\kappa_s-\kappa_c)} \cdot \Phi^{(t)}_{\mathbf{p}\to v_{k,1}}\}\big)|\alpha^{(t)}_{\mathbf{p}\to v_{k,1}}| \quad \text{for } m > 1 \tag{J.4}$$

By Lemma J.7 and Lemma J.5, we have $\alpha^{(t)}_{\mathbf{p}\to\mathcal{P}_{k,1}} \geq \Omega(\frac{1}{P^{3-2\kappa_c}}) \geq \lambda\Phi^{(t)}_{\mathbf{p}\to\mathcal{P}_{k,1}}$, which implies the regularization in this stage is not violated for the dominated FP correlation $\Phi^{(t)}_{\mathbf{p}\to\mathcal{P}_{k,1}}$. Hence, we could focus on the relation between $\alpha^{(t)}_{\mathbf{p}\to\mathcal{P}_{k,m}}$ and $\alpha^{(t)}_{\mathbf{p}\to\mathcal{P}_{k,1}}$ for $m > 1$.

Therefore, the existence of $T_1$ can be directly obtained by the gradient estimation in Lemma J.7 and the lower bound for the area attention of the global area in Lemma J.5. The induction argument follows directly from J.4.

□

The key takeaway from the first stage is that the growth of feature-position attention correlation for the global area is dominant, specifically, $\alpha^{(t)}_{\mathbf{p}\to v_{k,1}} \gg |\alpha^{(t)}_{\mathbf{p}\to v_{k,m}}|$. After this initial stage, $\Phi_{\mathbf{p}\to v_{k,1}}$ reaches $\Omega(\log(P))$, $\mathbf{Attn_{p\to\mathcal{P}_{k,1}}}$ has reached $\Omega(1)$ and $1-\ell_p$ still keeps at a constant level. The dominance of global FP correlation will be preserved in the following and the learning process will enter the convergence stage.

## J.3 CONVERGENCE

At this stage, we are going to prove that as long as the ViTs have already learned the global FP correlations, they will indeed converge to these global solutions, which leads to the collapsed global representation. We present the statement of our convergence theorem below.

**Theorem J.8** (Convergence guarantees). *Letting* $T_2 = \Omega(\frac{P^4 \log P}{\eta})$, *for any* $T \in [T_2, O((\frac{\text{poly}(P) \log P}{\eta}))]$, *letting* $\lambda = \Theta(\frac{1}{P\log P})$ *we have*

$$\frac{1}{T}\sum_{t=T_2}^{T}\mathcal{L}(Q^{(t)}) \leq \mathcal{L}^{\star}_{cl} + \frac{1}{\text{poly}\,P}.$$

*where $\mathcal{L}^{\star}_{cl}$ is the global minimum of the regularized contrastive objective.*

We have the following hypothesis for the end of the learning process.

**Induction Hypothesis J.2.** For $t \in [\Omega(\frac{P^4 \log P}{\eta}), O((\frac{\text{poly}(P) \log P}{\eta}))]$, we have the following resutls:

- For any $k \in [K]$, $\mathbf{p} \in \mathcal{P}$, and $m \in [N_k]$

$$\Phi^{(t)}_{\mathbf{p}\to v_{k,1}} \in [C_1^*, C_2^*]\log P \quad |\Phi^{(t)}_{\mathbf{p}\to v_{k,m}}| \leq \widetilde{O}(\frac{1}{P^{\delta_*}}).$$

where $C_1^*, C_2^* > 0$ are some constants and $\delta_* \in (0, 1)$ is some small constant.

- Attention score from the global area: given $X \in \mathcal{D}_k$, $1 - \mathbf{Attn}^{(t)}_{\mathbf{p} \to \mathcal{P}_{k,1}}(X^a) \leq \frac{1}{\mathrm{poly}(P)}$ for $a \in \{+, ++\}$ and $1 - \widetilde{\mathbf{Attn}}^{(t)}_{\mathbf{p} \to \mathcal{P}_{k,1}}(X^{n,s}) \leq \frac{1}{\mathrm{poly}(P)}$ for $s \in [N_c]$ with high probability.

- Bounded gradient for the loss:

$$\|\nabla_Q \mathcal{L}(Q^{(t)})\|_F^2 \leq \widetilde{O}(\frac{1}{\mathrm{poly}\, P}).$$

We can reuse most of the calculations in the proof of Induction Hypothesis J.1 to prove the hypothesis and here we only discuss how to bound the gradient of the objective. If the regularization is not violated, i.e., $\alpha^{(t)}_{\mathbf{p} \to v_{k,m}} \geq \frac{\Phi^{(t)}_{\mathbf{p} \to v_{k,m}}}{\lambda}$, we have $\Phi^{(t)}_{\mathbf{p} \to v_{k,m}} \leq O(\log(P))$. For $t \geq T_1$, denote the first time when $\alpha^{(t)}_{\mathbf{p} \to v_{k,m}} - \frac{\Phi^{(t)}_{\mathbf{p} \to v_{k,m}}}{\lambda} \leq O(\frac{1}{P^4})$ as $\widetilde{T}_1$, by Lemma J.7, we have $\widetilde{\alpha}^{(t)}_{\mathbf{p} \to v_{k,m}} \geq \Omega(\frac{1}{P^4})$ for $t \in [T_1, \widetilde{T}_1]$, and $\Phi^{(\widetilde{T}_1)}_{\mathbf{p} \to v_{k,m}} = \widetilde{C} \log P$ for some constant $\widetilde{C} > 0$. Then we have $\widetilde{T}_1 \leq O(\frac{P^4 \log P}{\eta})$. Thus, for $t \geq \widetilde{T}_1$,

$$\|\nabla_Q \mathcal{L}(Q^{(t)})\|_F^2 \leq O\left( \sum_{k=1}^{K} \sum_{\mathbf{p} \in \mathcal{P}} (\alpha^{(t)}_{\mathbf{p} \to v_{k,1}} - \frac{\Phi^{(t)}_{\mathbf{p} \to v_{k,m}}}{\lambda})^2 \right) \leq O(\frac{1}{\mathrm{poly}(P)}).$$

*Proof of convergence.* We first define a learning network that we deem as the "optimal" network with the global feature-position attention pattern. Specifically, we define $Q^\star$ as a matrix satisfied $e_{\mathbf{p}}^\top Q^\star v_{k,1} = \sigma_\star$ with $\sigma_\star^2 = \frac{\|\bar{Q}\|_F}{P(\sum_{k=1}^{K} N_k)}$ and $e_{\mathbf{p}}^\top Q^\star v_{k,m} = 0$ for $\mathbf{p} \in \mathcal{P}$ and $k \in [K]$, $m \in [N_k]$. Furthermore, $w_1^\top Q^\star w_2 = 0$, where $w_1, w_2 \in \mathrm{Span}\left( \{e_{\mathbf{p}}\}_{\mathbf{p} \in \mathcal{P}} \cap \{v_{k,m}\}_{k \in [K, m \in [N_k]]} \right)^\perp$. Here we suppose $\mathcal{L}^\star_{\mathrm{c}1}$ is achieved at the matrix $Q = \bar{Q}$.

Moreover, We consider the following **pseudo** losses and objective: define the linearized learner $\widetilde{F}^{(t)}(Q, X) = F(Q^{(t)}, X) + \nabla_Q F(Q^{(t)}, X)(Q - Q^{(t)})$,

$$\widetilde{\mathcal{L}}_t(Q) := \mathbb{E}\left[ -\tau \log \left( \frac{e^{\langle \widetilde{F}^{(t)}(Q, X^+), F(Q^{(t)}; X^{++}) \rangle / \tau}}{\sum_{X' \in \mathfrak{B}} e^{\langle \widetilde{F}^{(t)}(Q, X^+), F(Q^{(t)}; X') \rangle / \tau}} \right) \right],$$

$$\widetilde{\mathbf{Obj}}_t(Q) := \widetilde{\mathcal{L}}_t(Q) + \frac{\lambda}{2}\|Q\|_2^2,$$

and

$$\widehat{\mathcal{L}}_t(Q) := \mathbb{E}\left[ -\tau \log \left( \frac{e^{\langle F(Q, X^+), F(Q^{(t)}; X^{++}) \rangle / \tau}}{\sum_{X' \in \mathfrak{B}} e^{\langle F(Q, X^+), F(Q^{(t)}; X') \rangle / \tau}} \right) \right].$$

Then we discuss the values of different losses at $Q = Q^\star$. We have the following properties:

$$\mathcal{L}(Q^\star) \leq \mathcal{L}^\star_{\mathrm{c}1} + O(\frac{1}{\mathrm{poly}(d)}), \tag{J.5}$$

$$|\widehat{\mathcal{L}}(Q^\star) - \overline{\mathcal{L}}_t(Q^\star)| \leq O(\frac{1}{\mathrm{poly}(d)}), \tag{J.6}$$

$$|\widetilde{\mathcal{L}}_t(Q^\star) - \widehat{\mathcal{L}}_t(Q^\star)| \leq \frac{1}{\mathrm{poly}\, d}. \tag{J.7}$$

For the first property, we only need to consider the contrastive loss at the global minimum. Notice that for our data distribution, the global minimum of the contrastive loss is achieved when the network can perfectly distinguish the samples from different clusters. Thus, we have $\mathcal{L}^\star_{\mathrm{c}1} = \Theta(\log \frac{N_c}{K})$. Notice that on the event $\mathcal{A}_{1,com}$, supposing $X \in \mathcal{D}_k^{cl}$, which happens with prob $\geq 1 - e^{-P^{\kappa_s}}$ we have

$$\langle F(Q^\star, X^+), F(Q^\star; X^{++}) \rangle = \langle v_{k,1}, v_{k,1} \rangle + \frac{1}{|\Theta(C_{k,1})|^2} \sum_{\mathbf{p} \in \mathcal{P}_{k,1} \cap \mathcal{U}^+ \cap \mathcal{U}^{++}} \|\xi_{\mathbf{p}}\|_2^2 \pm o(1)$$

$$\langle F(Q^\star, X^+), F(Q^\star; X') \rangle = \langle v_{k,1}, v_{k,1} \rangle \pm o(1) \text{ for } X' \in \mathfrak{N} \cap \mathcal{D}_k^{cl}$$

Furthermore, by Bernstein's inequality, we have with probability $\geq 1 - \frac{1}{\text{poly}(d)}$, we have $\|\xi_\mathbf{P}\|_2^2 = \sigma_0^2 d \pm \widetilde{O}(\frac{1}{\text{poly}(d)}) = 1 \pm \widetilde{O}(\frac{1}{\text{poly}(d)})$, we denote such an event as $\mathcal{A}_3$. Suppose we consider the temperature $\tau = O(\frac{1}{\log d})$, then conditioned on $\mathcal{A}_{1,com} \cap \mathcal{A}_3$, we have $\langle F(Q^\star, X^+), F(Q^{(t)}; X') \rangle = \omega(\log d) \pm o(1)$ for $X' \in \mathfrak{B} \cap \mathcal{D}_k^{cl}$, which could minimize the loss to the level of $\Theta(\log \frac{N_c}{K})$ up to the error of $O(\frac{1}{\text{poly}(d)})$. Then we have

$$\mathcal{L}(Q^\star) \leq (1 - \frac{1}{\text{poly}(d)}) \Theta(\log \frac{N_c}{K}) + \widetilde{O}(\frac{1}{\text{poly}(d)}) \leq \mathcal{L}(\overline{Q}) + O(\frac{1}{\text{poly}(d)}).$$

The second property follows from the observation that

$$|\widehat{\mathcal{L}}_t(Q^\star) - \overline{\mathcal{L}}(Q^\star)| \leq O(\|\nabla_Q \overline{\mathcal{L}}(Q^\star)\|_2) \|F(Q^\star, X^{++}) - F(Q^{(t)}, X^{++})\|_2$$
$$\leq O\Big(\|\nabla_Q \overline{\mathcal{L}}(Q^\star)\|_2 \cdot \Big(1 - \mathbf{Attn}_{\mathbf{p} \to \mathcal{P}_{k,1}}^{(t)}(X^{++})\Big)\Big) \leq \widetilde{O}(\frac{1}{\text{poly}(P)}).$$

Similarly, the third property follows from the fact that

$$|\widetilde{\mathcal{L}}_t(Q^\star) - \widehat{\mathcal{L}}_t^{cl}(Q^\star)| \leq O(\|\nabla_Q \widetilde{\mathcal{L}}_t(Q^\star)\|_2) \|\widetilde{F}^{(t)}(Q^\star, X^+) - F(Q^\star, X^+)\|_2 \leq \widetilde{O}(\frac{1}{\text{poly}(P)}).$$

Now we will use the tools from online learning to obtain a loss guarantee:

$$\eta \langle \nabla_Q \mathcal{L}(Q^{(t)}), Q^{(t)} - Q^\star \rangle$$
$$= \frac{1}{2}\eta^2 \|\nabla_Q \mathcal{L}(Q^{(t)})\|_F^2 - \frac{1}{2}\|Q^{(t)} - Q^\star\|_F^2 + \frac{1}{2}\|Q^{(t+1)} - Q^\star\|_F^2$$
$$= \frac{\eta^2}{2} \cdot \frac{1}{\text{poly}(P)} - \frac{1}{2}\|Q^{(t)} - Q^\star\|_F^2 + \frac{1}{2}\|Q^{(t+1)} - Q^\star\|_F^2.$$

Notice that $\widetilde{\mathbf{Obj}}_t(Q)$ is a convex function over $Q$ and $\widetilde{\mathbf{Obj}}_t(Q^{(t)}) = \mathcal{L}(Q^{(t)})$, thus

$$\langle \nabla_Q \mathcal{L}(Q^{(t)}), Q^{(t)} - Q^\star \rangle = \langle \nabla_Q \widetilde{\mathbf{Obj}}_t(Q^{(t)}), Q^{(t)} - Q^\star \rangle$$
$$\geq \widetilde{\mathbf{Obj}}_t(Q^{(t)}) - \widetilde{\mathbf{Obj}}_t(Q^\star) \qquad \text{(by convexity)}$$
$$\geq \widetilde{\mathbf{Obj}}_t(Q^{(t)}) - \mathcal{L}(Q^\star) - \widetilde{O}(\frac{1}{\text{poly}(P)}) \qquad \text{(by J.6 and J.7)}$$
$$\geq \widetilde{\mathbf{Obj}}_t(Q^{(t)}) - \mathcal{L}_{cl}^\star - \widetilde{O}(\frac{1}{\text{poly}(P)}) \qquad \text{(by J.5)}$$
$$= \mathcal{L}(Q^{(t)}) - \mathcal{L}_{cl}^\star - \widetilde{O}(\frac{1}{\text{poly}(P)}) \qquad \text{(by definition of } \widetilde{\mathbf{Obj}})$$

Thus by a telescoping summation, we have

$$\frac{1}{T - T_2} \sum_{t=T_2}^{T} \mathcal{L}(Q^{(t)}) - \mathcal{L}_{cl}^\star$$
$$\leq \sum_{t=T_2}^{T} \langle \nabla_Q \mathcal{L}(Q^{(t)}), Q^{(t)} - Q^\star \rangle + O(\frac{1}{\text{poly}(P)})$$
$$\leq O(\frac{\|Q^{(T)} - Q^{(\star)}\|_2^2}{T\eta}) \leq O(\frac{1}{\text{poly}(P)})$$

which completes the proof. $\qquad \square$

