# OpenReview forum: "A Theoretical Analysis of Self-Supervised Learning for Vision Transformers"
_ICLR.cc/2025/Conference — ICLR 2025 Poster_

### Official Review · Reviewer_ChhH · 2024-10-23

**Soundness:** 4
**Presentation:** 3
**Contribution:** 4
**Rating:** 8
**Confidence:** 4

**Summary:**

This paper discusses the reasons for the different attention mechanisms between discriminative (contrastive-learning) and generative (MAE) approaches for self-supervised learning with solid proof. This mainly contributes to the community of how these two learning strategies can be chosen and also further pushes the detailed understanding of the mechanism behind these two methods.

**Strengths:**

1. The quality of the paper is great; details for the problem setup, statement, and proof are given in the easy-to-understand setup.
2. The theoretical analysis provides a solid step to understanding the detailed differences between the mechanisms and can help future researchers develop more approaches based on this analysis.

**Weaknesses:**

1. The finding within the Sec. B should appear in the main paper instead of the Appendix. Some of the sections, such as Sec. 2.3 and 2.4 can be shortened since there are some overlapped concepts. Afterward, the Fig. 4 and the explanation can be moved up to provide more insight into local and global differences.

**Questions:**

1. In the first paragraph of Sec. 2.1 about **Masked reconstruction-based learning**, should it be $\mathcal{M}_i \subset \mathcal{P}$ instead of $\mathcal{M}_i \subset [P]$?
2. Does the same cluster exist in different $\mathcal{D}_k$ defined in Sec. 2.2 and depicted in Fig.2? If this is true, should this assumption be a little bit overwhelming? For CL, only entities obtained from the same images are considered positive. As for MAE, only local features are important. I am sure changing this assumption won't affect the proof, but I want to clarify this.
3. From the proof, can we determine the required ratio between positive and negative samples to ensure the global convergence properties? Since approaches such as MoCo require more negative samples to prevent collapsing.

---

> ### Author Response · Authors · 2024-11-19
> **Response to Reviewer ChhH**
>
> We thank the reviewer for providing the valuable feedback! In our revised version, we have highlighted all changes using red-colored text for clarity.  Below we address your concerns.
>
> > **Q1.** The finding within the Sec. B should appear in the main paper instead of the Appendix.
>
> Thank you for your helpful suggestions! In the revision, we have included our empirical findings (see Figure 3) and discussed insights for local and global differences (see line 237-240)  in the main body of the paper as suggested by the reviewer.
>
> > **Q2.** Should it be $\mathcal{M}_i\in \mathcal{P}$ instead of $\mathcal{M}_i\in [P]$
>
> Thank you for pointing out this issue. We have updated this notation in the revision to maintain consistency and clarity throughout the paper.
>
> > **Q3.**  Does the same cluster exist in different $\mathcal{D}_k$ defined in Sec. 2.2 and depicted in Fig.2?
>
> We would like to clarify that the area partitions of different $\mathcal{D}_k$ need not be the same. The only assumption we impose on the area partitions is about the number of patches of the areas within one cluster, without specifying any fixed spatial arrangements across different clusters.
>
> > **Q4.** From the proof, can we determine the required ratio between positive and negative samples to ensure the global convergence properties?
>
> Thank you for your question. Our current analysis only estimates the number of negative samples needed to ensure certain concentration properties essential for our analysis. A more fine-grained characterization of the positive-to-negative sample ratio, particularly in relation to global convergence rates, would require a detailed examination of data properties and complex calculations of the dynamics. This is beyond our current goal but is an interesting question on its own. In fact, we would like to see some extensions or investigations into how transformers detect the most significant features, not just in the sense of number of patches, but maybe also how much magnitude of the activations they produce as compared to other features. We believe there are still many unrevealed mysteries on the inductive bias of vision transformers and self-supervised learning.
>
> ***
> We thank the reviewer again for your highly inspiring comments. We hope that our responses have resolved your concerns. In case you have further questions, please feel free to let us know, and we will be more than happy to answer them.

---

> > ### Comment · Reviewer_ChhH · 2024-11-21
> >
> > I appreciate the authors' response, which addresses most of my concerns.

---

### Official Review · Reviewer_WFmo · 2024-10-28

**Soundness:** 3
**Presentation:** 3
**Contribution:** 3
**Rating:** 6
**Confidence:** 2

**Summary:**

This paper provides a theoretical framework for understanding self-supervised learning (SSL) in vision transformers, comparing masked autoencoders (MAE) and contrastive learning (CL). It examines how these methods handle imbalances between global and local visual features, revealing that MAE learns both global and local features, resulting in diverse attention patterns, whereas CL tends to focus on global features, often collapsing into uniform attention patterns. This distinction supports empirical observations, showcasing MAE’s effectiveness in capturing spatially varied data, while CL is better suited for global pattern recognition.

**Strengths:**

1. This paper introduces a novel theoretical framework for analyzing self-supervised learning (SSL) in vision transformers, filling a gap in the field that has been primarily empirical.
2. The paper effectively illustrates the differences in feature capture between MAE and CL, showing how MAE learns both global and local features, while CL focuses more on global features.
3. Through a detailed gradient descent analysis of a single-layer transformer, the study offers concrete findings on how MAE and CL converge to distinct attention patterns, ensuring result reliability and reproducibility. The study focuses on a single-layer transformer model, which may overlook the dynamics of self-supervised learning in deeper or more complex networks.

**Weaknesses:**

The study focuses on a single-layer transformer model, which may overlook the dynamics of self-supervised learning in deeper or more complex networks.

**Questions:**

Given that this study primarily focuses on single-layer transformer models, how might the dynamics of self-supervised learning differ in deeper or more complex transformer architectures? Specifically, Could the authors discuss potential approaches or challenges in extending their analysis to multi-layer transformers? Are there specific aspects of the current proof techniques that may or may not generalize to deeper architectures?

---

> ### Author Response · Authors · 2024-11-19
> **Response to Reviewer WFmo**
>
> We thank the reviewer for the helpful comments! In our revised version, we have highlighted all changes using red-colored text for clarity. Below we address your concerns.
>
> > **Q1.** Regarding deeper or more complex transformer architectures?
>
> *a.* We want to point out that a rigorous analysis of the training dynamics of deep transformers remains an extremely challenging open problem, currently beyond known technical tools.  The difficulty can arise from different directions. The interactions between the weights and gradients in multiple layers are unclear and unlikely to be fully understood by the current tools. The architectural inductive bias may be more compounded with more layers involved, blurring the analysis of individual components. And as we known currently, there is hardly any general technical tools for non-convex optimizations, and the existing general tools rarely produces results that connects to the practice of deep learning. This is evidenced by the fact that most of the recent theoretical studies [1-7] of characterizing the training dynamics of transformers and ViTs focused on the single attention layer, which already required highly non-trivial techniques.
>
> *b.* A rigorous theoretical analysis for training dynamics of multi-layer transformers is only possible when stricter assumptions are added (for example, when assuming layer-wise training or using even more specialized training recipes), and is in general beyond what the current theoretical tools can handle.
>
> *c.* Lastly, we would like to emphasize that  the goal of this paper is to offer a theoretical explanation for an interesting behavior gap of ViTs pretrained with MAE and CL. Our theoretical findings show that even with one-layer ViTs, we can provably characterize a similar behavior separation which aligns well with the empirical observations of full ViTs models. This suggests that our theoretical insights are not limited to a single self-attention layer but may apply more broadly to full ViTs.
>
> ***
> We thank the reviewer again for your highly inspiring comments. We hope that our responses have resolved your concerns. In case you have further questions, please feel free to let us know, and we will be more than happy to answer them.
>
> [1] Jelassi et al. Vision transformers provably learn spatial structure
>
> [2] Tian et al. Scan and Snap: Understanding Training Dynamics and Token Composition in 1-layer Transformer
>
> [3] Zhang et al. Trained transformers learn linear models in-context
>
> [4] Huang et al. In-Context Convergence of Transformers
>
> [5] Tarzanagh et al. Max-Margin Token Selection in Attention Mechanism
>
> [6] Li et al., A Theoretical Understanding of Shallow Vision Transformers: Learning, Generalization, and Sample Complexity.
>
> [7] Li et al., How Do Nonlinear Transformers Learn and Generalize in In-Context Learning?

---

> > ### Author Response · Authors · 2024-11-22
> > **Follow Up Reminder**
> >
> > Dear Reviewer WFmo,
> >
> > We've taken your initial feedback into careful consideration in our response. Could you kindly confirm whether our responses have appropriately addressed your concerns?
> >
> > In case you have further questions, please feel free to let us know, and we will be more than happy to answer them.
> >
> > Thank you for your time and effort in reviewing our work!
> >
> > Many thanks, Authors

---

> > > ### Comment · Reviewer_WFmo · 2024-11-26
> > >
> > > I have read the author's response, and I maintain my rating and confidence.

---

### Official Review · Reviewer_3zvq · 2024-11-04

**Soundness:** 3
**Presentation:** 2
**Contribution:** 3
**Rating:** 5
**Confidence:** 3

**Summary:**

This paper claim to present a theoretical analysis of self-supervised learning for Vision Transformers (ViTs), focusing on the differences between contrastive learning (CL) and masked autoencoders (MAE). The study introduces a novel framework to model visual data distribution, considering both dominant global features and minuscule local features. By analyzing the training dynamics of one-layer softmax-based self-attention using gradient descent on spatial structured data, the authors demonstrate that MAE effectively learns both global and local features, while CL tends to focus predominantly on global features. The research provides a rigorous theoretical explanation for empirically observed behaviors of MAE and CL in ViTs. It highlights how the degree of feature imbalance affects the learning process, offering insights into why MAE can capture both global and subtle local information simultaneously, while CL tends to emphasize global patterns.

**Strengths:**

1. This paper proposes a comprehensive framework to theoretically analyse attention mechanism for CL and MIM. The framework simplify the analysis, avoiding disentangling patches and positional encoding.
2. This work attempt to answers the empirical observations for CL and MIM with ViTs: the MIM captures diverse local patterns, while the CL focuses on the dominant patterns.
3. Base on the proposed theory, this paper finds that there are two phases to learn the feature-position correlations when information gap is positive.

**Weaknesses:**

1. Vision transformers and transformers in general consists of two main blocks multi-head self-attention block as well as so called FFN consisting of two point-wise convolutions. The discussion is based on only one-layer of self-attention (with over simplification to only one weight matrix for self-attention) within vision transformer. Thus it cannot fully explain the complex behaviour of ViTs.
2. There are a lot of assumptions, such as the input data distribution and positional encoding. It is not clear the differences in real problems.
3. What is the insight from this work to develop new methods? How to prevent the attention collapse in CL?
4. The discussion is based on SGD. However, Adam is the popular optimizer for ViTs.
5. The original MIM (first MIM work with ViTs which outperformed supervised pretraining) work SiT [1] from which idea of MAE [4] is derived/copied from (only MIM part of SiT is taken in MAE) combines both MIM and CL, a strategy proven to be better and more principled time and again. Furthermore, state-of-the-art methods like iBoT [2] and its extension DINOv2 [3] also follow combined MIM and CL. It would be more fruitful to build theoretical framework for this combined MIM and CL setting (which seems to be the way forward in SSL) rather than building framework for MIM and CL separately.

[1] Atito etal., "SiT: Self-supervised vIsion Transformer", arxiv 04, 2021.
[2] Zhou etal., "iBOT: Image BERT Pre-Training with Online Tokenizer", arxiv 11, 2021.
[3] Oquab etal., "DINOv2: Learning Robust Visual Features without Supervision", arxiv 04, 2023.
[4] He etal., "Masked Autoencoders Are Scalable Vision Learners", arxiv 11, 2021.

**Questions:**

- The theory particularly for CL seems to be build on the notion of dominant concept which takes most of the area of the input. This assumption is not valid for almost all practical datasets for SSL. For instance, the average object to total image area ratio is 0.35 in ImageNet. Which means the so called background is almost always going to be dominant concept according this paper's definition. In that case the attention from any part of the image should focus on the background most of the time. If this is the case the kNN evaluation of DINO and MoCo should not be so high for imagenet and more importantly for other datasets.
- Is the proposed theory would still be valid for datasets where the global area is small?
- What will happen if the local areas are larger than the global area? This situation is going to be the dominant case in most practical datasets.
Minors:
- what is $k_s$? It seems the sum of weights for global area.
- What $\Delta$ means? Can you explain in details, such as when $\Delta=-1,0,1$?

---

> ### Author Response · Authors · 2024-11-19
> **Response to Reviewer 3zvq Part 1/3**
>
> We thank the reviewer for the time and thoughtful comments on our work. In our revised version, we have highlighted all changes using red-colored text for clarity. Below we address your concerns.
>
>
> > **Q1.** based on only one-layer of self-attention, cannot fully explain the complex behavior of ViTs
>
> *a.* Along the research line of characterizing the training dynamics of transformers and ViTs, even the theoretical analysis of a single attention layer with one weight matrix simplification requires highly non-trivial techniques, evidenced by the most recent such studies [1-5] on the training dynamics of one-layer transformers under various settings and data models.
>
> *b.* Even with the one-layer attention-only setting, our objective is highly non-convex, leading to complicated multi-phase decompositions in analyzing the training dynamics. A rigorous theoretical analysis for multi-layer ViTs is only possible when stricter assumptions are added (for example, when assuming layer-wise training or using even more specialized training recipes), and is in general beyond what the current theoretical tools can handle.
>
> *c.* The goal of this paper is to offer a theoretical explanation for an interesting behavior gap of ViTs pretrained with MAE and CL. Our theoretical findings show that even with one-layer ViTs, we can provably characterize a similar behavior separation which aligns well with the empirical observations of full ViTs models. This suggests that our theoretical insights are not limited to a single self-attention layer but may apply more broadly to full ViTs. We hope this work could inspire interests in the theory community to extend our results.
>
> > **Q2.** a lot of assumptions:
>
> *a.* We would like to emphasize two key points about these assumptions: i). Most assumptions, such as the input data distribution, are abstracted from real-world datasets that exhibit both global and local features, as evidenced by various empirical studies [6-8]; ii). Our assumptions align with those commonly made in existing theoretical work. For instance, positional encoding is a standard assumption in many studies, as referenced in [1,9].
>
> *b.* Our assumptions represent necessary compromises to make complex real-world phenomena amenable to mathematical analysis. For instance, precisely characterizing real data distributions [1,4] or optimizing multi-layer transformers [1-5] mathematically is challenging. Our focus is on distilling the core architecture—specifically the attention mechanism— to better understand and simulate these phenomena. Our approach simplifies the complex designs of self-supervised learning and ViTs to uncover fundamental dynamics using mathematical tools, serving as a prototype for practical observations.
>
> > **Q3.** insight from this work to develop new methods? How to prevent the attention collapse in CL?
>
> *a.* Our theoretical results implied that CL by nature tends to capture the most prominent features in an image, which are often global, while MIM can better capture minor, local features. This complementary capability motivates the potential advantage for hybrid objectives that merge the strengths of both approaches. By doing so, we can harness CL's efficiency at highlighting critical global information alongside MIM's precision in local feature detection.
>
> *b.* It is important to note that avoiding attention collapse in CL is not universally advantageous. There are scenarios where such collapse might be beneficial, particularly when the goal is to highlight dominant image features explicitly. However, for applications requiring a broader exploration of feature space, integrating the CL framework with objectives that prioritize local detail—similar to those used in MAE methods—can be beneficial. Such an approach will ensure a balanced attention mechanism that leverages the strengths of both global and local feature extraction.
>
>
> > **Q4.** The discussion is based on SGD. However, Adam is the popular optimizer for ViTs.
>
> Although Adam is commonly used for training ViTs, Theoretical works typically focus on from SGD/GD to provide a cleaner analysis and clearer presentations of the training behaviors. Adam, due to its momentum and normalization, results in significantly more complex training dynamics. To the best of our knowledge, nearly all of the existing theories for analyzing training dynamics for transformers and ViTs e.g., [1-5,9], including ours, focus on gradient descent. However, we believe our analysis techniques can be further extended to study Adam by incorporating the previous optimization tools for handling Adam.

---

> > ### Author Response · Authors · 2024-11-19
> > **Response to Reviewer 3zvq Part 2/3**
> >
> > > **Q5.** It would be more fruitful to build theoretical frameworks for this combined MIM and CL setting (which seems to be the way forward in SSL) rather than building frameworks for MIM and CL separately.
> >
> > *a.* We appreciate the reviewer’s suggestion and recognize the high value of it. However, a comprehensive analysis of the combination of MIM and CL is challenging without a solid foundational understanding of each approach separately. To our knowledge, our work is the first to provide theoretical characterizations and comparative analyses of both MIM and CL with ViTs architecture. We believe that the theoretical tools developed here are crucial building blocks towards analyzing combined methodologies in the future.
> >
> > *b.* It's important to note that theory often trails empirical work, which can explore numerous possibilities quickly. Theoretical approaches, in contrast, require rigorous and precise formulation. Our goal is to establish strong theoretical foundations that enhance our understanding and guide the development of more principled methods in the field.
> >
> > > **Q6.** The theory particularly for CL seems to be built on the notion of dominant concept which takes most of the area of the input. This assumption is not valid for almost all practical datasets for SSL.
> >
> > Thank you for your insightful question, which highlighted potential misunderstandings from our writing. We are glad to clarify some points here:
> >
> > *a.*  Firstly, in our model's global and local data setting, we assume that *the number of patches in the global feature area are greater than the number of patches in **each** local feature area*, which ruled out the existence of a larger local feature as suggested in the review. Referring to the example of ImageNet mentioned by the reviewer—where "the average object to total image area ratio is 0.35 in ImageNet"—it seems there might be a misunderstanding that our assumption implies the number of patches in the global feature area should be greater than the **total** number of patches across all local feature areas, which is not the case as we only require the global area to be larger than any other individual feature areas.
> >
> > *b.* Another note (possibly further clarifying the above point)  is that we do not consider the entire background as a single, unified feature. Instead, we define features based on groups of co-occurring patches that exhibit strong spatial correlations. For example, backgrounds typically comprise numerous smaller objects, such as trees, leaves, and buildings. These background elements do not usually have fixed spatial layout across the data distribution, and therefore do not exhibit strong spatial correlations. Our setting defines these background elements as individual local features, rather than a unified background feature. Thus, it is natural that **each** of these background local features tends to have fewer patches than the global feature.
> >
> >
> > > **Q7.**  What will happen if the local areas are larger than the global area? This situation is going to be the dominant case in most practical datasets.
> >
> > Thank you for your question. We hope that our response to Question 6 has also addressed this concern. Specifically, our assumption requires that the number of patches in the global area exceeds the number of patches in **any individual** local area, which is consistent with the characteristics of many practical datasets. Importantly, this assumption allows for scenarios where the **total number** of patches across all local areas is larger than the number in the global area.
> >
> > > **Q8.** Is the proposed theory would still be valid for datasets where the global area is small?
> >
> > Our theoretical analysis will continue to hold without relying on whether one area arises as a global area with a dominant size than other areas. The sizes of these areas determine only their relative convergence rate for MAE. Namely, a global area (if there is one) would have a faster convergence rate than other areas; otherwise, if all areas have relatively the same size, they all converge at a similar rate.

---

> ### Author Response · Authors · 2024-11-19
> **Response to Reviewer 3zvq Part 3/3**
>
> > **Q9.** what is  $\kappa_s$?  It seems the sum of weights for global area
>
> As outlined in Assumption 2.2,  $P^{\kappa_s}$ represents the number of patches in **each** local area. Therefore,  $\kappa_s$ specifically measures the number of patches in each local area. Instead, $P^{\kappa_c}$ refers to the number of patches in the global area.
>
> > **Q10.** what $\Delta$ means?
>
> *a.* As outlined in Assumption 2.2, $\kappa_c$ and $\kappa_s$ determine the number of patches belonging to the global ($ P^{\kappa_c} $) and each local ($ P^{\kappa_s} $) area respectively. Consequently, the difference $ \Delta = (1-\kappa_s) - 2(1-\kappa_c) \approx \kappa_c - \kappa_s $, as specified in Equation (4.1), quantifies the disparity in the number of patches between these areas. This metric allows us to gauge to what extent the global area contains more information than each local area, due to each patch carrying feature information specific to its designated area.
>
> *b.* Specifically, a larger $\Delta $, such as 1, indicates that the number of global features substantially exceeds the number of local features, highlighting a significant imbalance. Conversely, a smaller $\Delta$, such as smaller than 0, suggests a more balanced distribution between global and local features.
>
> ***
> We thank the reviewer again for your highly inspiring comments. We hope that our responses have resolved your concerns. If so, we wonder if the reviewer could kindly consider increasing your score. Certainly, we are more than happy to answer your further questions.
>
>
> [1] Jelassi et al. Vision transformers provably learn spatial structure
>
> [2] Tian et al. Scan and Snap: Understanding Training Dynamics and Token Composition in 1-layer Transformer
>
> [3] Zhang et al. Trained transformers learn linear models in-context
>
> [4] Huang et al. In-Context Convergence of Transformers
>
> [5] Tarzanagh et al. Max-Margin Token Selection in Attention Mechanism
>
> [6] Park et al., What Do Self-Supervised Vision Transformers Learn?
>
> [7] Wei et al., Contrastive learning rivals masked image modeling in fine-tuning via feature distillation
>
> [8] Xie et al., Revealing the Dark Secrets of Masked Image Modeling
>
> [9]Wang et al., Transformers Provably Learn Sparse Token Selection
> While Fully-Connected Nets Cannot

---

> > ### Author Response · Authors · 2024-11-22
> > **Follow Up Reminder**
> >
> > Dear Reviewer 3zvq,
> >
> > We've taken your initial feedback into careful consideration in our response. Could you kindly confirm whether our responses have appropriately addressed your concerns?
> >
> > If you find that we have properly addressed your concerns, could you please kindly consider increasing your initial score accordingly? Please let us know if you have further comments.
> >
> > Thank you for your time and effort in reviewing our work!
> >
> > Many thanks, Authors

---

> > > ### Comment · Reviewer_3zvq · 2024-12-01
> > >
> > > Overall, this paper has contributed to understanding the behaviours of CL and MIM, but the contributions are limited in the proposed theoretical framework, lacking a reality check and insights for algorithm design. The work is solid, but the paper should be improved from the following aspects:
> > >
> > > 1. This theoretical framework introduces several assumptions. It could be reasonable to use some of them. However, it is not clear for the aggregation of them. As a result, the reliability is a concern. Experiments induced from the proposed Theorems are crucial to examine the theoretical framework. For example, in line 514 : global FP correlations are learned first, and focusing on these global correlations is sufficient for the CL objective to converge. Could you prove this by experiments to validate the correctness?
> > >
> > > 2. This work is limited in theoretical analysis, lacking new messages for SSL design. How can the proposed theory guide the design of SSL algorithms? The "insights for more principled future methods" from the authors that the methods should be hybrid to benefit global as well as local information is already the standard practice in both NLP for 6 years (BERT is using Masked AutoEncoders (MAE) as well as global contrastive loss) and in vision for 3.5 years (SiT [1] the first work for  vision transformers based SSL which employed both MAE and CL. Subsequently, iBoT [2] followed this established practice from SiT to do feature level recovery for MAE along with global loss and many more (surprisingly both iBoT and SiT are not even cited in your paper!). The same hybrid principle is now used for large scale vision self-supervised pretraining and is a go to method, e.g., DINOv2 [3]. Besides, the fact that these references are not cited in the paper suggests gaps in the authors' literature review, raising concerns about the paper's awareness of key developments.
> > > An alternative perspective of looking at the authors insights that derived works like so called MAE [4] and SimMIM are not the way forward as it will not benefit from global characteristic and neither are SIMCLR, BYOL, BarlowTwin, DINO as it will only benefit from global characteristics. But that is already well known (was even well known in vision SSL community at the time of MAE and SimMIM publication on arxiv) as neither MAE nor SimMIM outperformed original works like iBoT and SiT. Moreover, now the vision SSL community is using SiT and iBoT based principles as default for pretraining, as in CMAE [5], DINOv2 [3] etc. In summary, my concern remains, and I believe any new insights and suggestions when combining CL or MIM method would have been useful.
> > >
> > > 3. The FFN/MLP layers of transformers are very important. They store core knowledge which a transformer is learning. There are several work showing that a large number of attention layers can be removed/pruned (approximated by linear operation) while only slightly compromising the performance at inference time.  Ignoring MLP layers for analysis is not useful. Particularly, the title "A theoretical Analysis of SSL for ViT" suggest that the work should considered more than just one attention layer with a single layer, i.e., combination of multiple transformer blocks (including both attention and FFN layer) is needed to justify the name and scope of the work.
> > >
> > > [1] Atito etal., "SiT: Self-supervised vIsion Transformer", arxiv 04, 2021.
> > > [2] Zhou etal., "iBOT: Image BERT Pre-Training with Online Tokenizer", arxiv 11, 2021.
> > > [3] Oquab etal., "DINOv2: Learning Robust Visual Features without Supervision", arxiv 04, 2023.
> > > [4] He etal., "Masked Autoencoders Are Scalable Vision Learners", arxiv 11, 2021.
> > > [5] Huang etal, "Contrastive Masked Autoencoders are Stronger Vision Learners", TPAMI 4, 2024

---

> ### Author Response · Authors · 2024-12-02
>
> We sincerely hope the reviewer could evaluate the contributions of the paper by positioning the paper in the state-of-the-art of the **theoretical study** of transformers.  Please note that the theoretical progress of transformers is still highly restricted by the existing mathematical tools, which can handle only limited architectures and under certain assumptions. Expecting the theoretical setting to fully capture the complicated experimental settings is somewhat unrealistic given the current techniques that have been developed. This said, this paper makes a significant step forward in developing rigorous techniques and frameworks for analyzing transformers in SSL. The technical problem that we have handled is already one of the most challenging mathematical problems compared to the existing theoretical studies on the training dynamics of transformers in other tasks.
>
> > **Q1. This theoretical framework introduces several assumptions.**
>
> We would like to highlight that our work aims to provide a **theoretical** framework explaining the behavior gap between two SSL approaches. The results serve as proof techniques to offer insights, not as definitive representations of practical phenomena. Expecting absolute scientific accuracy in such complex systems exceeds the scope of theoretical studies. Moreover:
>
> 1. **Generality vs Specificity**: Theoretical works that aim to provide very general theorems often fail to provide fine grained characterization of an example with special structures. We carefully chose our assumptions that align with standard theoretical studies of deep learning to enable analytical tractability. Simplified settings, such as ours, are widely used to provide foundational insights [1–9]. And even though they have added certain ad hoc assumptions to enable mathematical analysis, the shared mechanisms are still manifested in the proofs and experiments.  It is our contribution, not drawbacks, to find a set of assumptions to enable these analyses and make them simple, to reach sufficiently interesting results in the theoretical sense. We have explained the role of specific assumptions in the paper and our initial response. Finally, due to the approach we took, our theory is highly interpretable, and we are open to addressing further specific questions if needed.
>
> 2. **Comparison Study via Theory is difficult:** Creating a unified setting to compare both SSL methods is significantly more challenging than developing separate theoretical results for each with fewer assumptions. This effort necessitates compromises, which are reflected in our design.
>
> 3. **Empirical Validation**: The claim regarding “global FP correlations” has been supported by existing empirical explorations [11-13], including our own attention distance experiments. Unfortunately, we cannot offer new experiments given the short fuse, and we kindly remind the reviewer  that **per ICLR policy, requests for additional experiments at this stage are not permissible**.
>
> > **Q2. This work is limited in theoretical analysis, lacking new messages for SSL design**
>
> 1. As noted in our initial response, our work includes empirical insights that could inform SSL algorithm design, but its primary goal is to provide a theoretical explanation. Expecting every theoretical contribution to directly inspire state-of-the-art designs is beyond the foundational scope of theoretical research.
>
> 2. We respectfully disagree with the critique regarding the omission of the paper mentioned by the reviewer, as it does not reflect a lack of awareness of key developments. Our focus is on the theoretical comparison of the two methods, rather than on the empirical design of combining them. Therefore, while the referenced paper may be tangentially related, it is not core to our contribution and does not directly align with the objectives of our work.

---

> > ### Author Response · Authors · 2024-12-02
> >
> > > **Q3. The FFN/MLP layers of transformers are very important.**
> >
> > 1. **Focus on Attention and Exclusion of MLP Layers:**
> >    - Our study focuses on attention patterns, as they are central to the metrics motivated by prior empirical research [11–13]. The attention-only setting in our work successfully reproduces (to some extent) key observations from these studies, offering valuable insights into the dynamics of attention. Within this setting, there is no evidence to suggest that hidden mechanisms within MLP layers would significantly impact the attention patterns studied.
> >    - While MLP layers play an important role in most transformers, studies such as Physics of Language Models 3.3 [10] show that MLP layers are not indispensable. Attention-only models can store knowledge effectively and even outperform MLP layers on a per-parameter basis in scaling experiments.
> >    - Furthermore, most existing works on transformer training dynamics also focus on attention-only layers [3–8] or rely on extreme assumptions when incorporating MLPs [9]. This underscores that excluding MLP layers is both an accepted practice in the theoretical community to allow analytical tractability, as jointly considering MLPs currently exceeds the limits of theoretical tools.
> >
> > 2. **Challenges in Multi-Layer Analysis:** Rigorous analysis of multi-layer transformer dynamics remains an open problem, as current tools cannot handle the complexity of gradient errors accumulating across layers. This limitation is evident from most recent studies [3–9], which focus on single-layer models and already require advanced techniques. Addressing multi-layer transformers requires stricter assumptions or specialized training schemes, which are beyond the reach of current techniques.
> >
> > ---
> >
> > **References:**
> >
> > [1] Allen-Zhu et al. Towards Understanding Ensemble, Knowledge Distillation and Self-Distillation in Deep Learning
> >
> > [2] Wen et al. Toward understanding the feature learning process of self-supervised contrastive learning
> >
> > [3] Jelassi et al. Vision transformers provably learn spatial structure
> >
> > [4] Zhang et al. Trained transformers learn linear models in-context
> >
> > [5] Huang et al. In-Context Convergence of Transformers
> >
> > [6] Nichani et al.  How Transformers Learn Causal Structure with Gradient Descent
> >
> > [7] Wang et al. Transformers Provably Learn Sparse Token Selection
> > While Fully-Connected Nets Cannot
> >
> > [8] Huang et al. Non-asymptotic convergence of training transformers for next-token prediction
> >
> > [9] Li et al.  A Theoretical Understanding of shallow Vision Transformers: Learning, Generalization, and Sample Complexity
> >
> > [10] Allen-Zhu et al.   Physics of Language Models: Part 3.3, Knowledge Capacity Scaling Laws
> >
> > [11] Park et al. What Do Self-Supervised Vision Transformers Learn?
> >
> > [12] Wei et al.  Contrastive learning rivals masked image modeling in fine-tuning via feature distillation
> >
> > [13] Xie et al.  Revealing the Dark Secrets of Masked Image Modeling

---

### Official Review · Reviewer_i2aQ · 2024-11-04

**Soundness:** 2
**Presentation:** 1
**Contribution:** 3
**Rating:** 6
**Confidence:** 3

**Summary:**

This paper is about understanding and contrasting the behavior of two pretraining techniques used for ViT architectures: Masked Auto-encoding and Contrastive Learning. They aim at providing convergence guarantees for this behavior as well. The overall conclusion is that the MAE learning objective learn both global and local behaviors, while contrastive learning has a bias towards only learning global behaviors.

**Strengths:**

Extensive, well-formulated, and thorough proofs.

**Weaknesses:**

- Not sure where they conducted their experiments and where the results are. They alluded to some experiments here: “line 377: However, we observe that the selected area does not necessarily depend on the location of p”. Where are these experiments located?
- This work lacks serious experimental validation. They need some kind of information bottleneck need experiments which play with the information gap and the size of the ViT tokens/patches. How do you measure this imbalance between local and global features?
- The abstract mentions “analysis” and never mentions “convergence guarantees”. Initially, I thought this paper would have extensive experimentation documenting the global and local behavior of ViT pre-trained with different objectives. But the paper isn’t about that.

**Questions:**

- Information gap: What are the bounds/range on this? Can we see a visualization or even a histogram that shows how this value changes for different sizes of local image patches/tokens? Maybe apply it to different samples from various datasets and comment on the behaviour observed? The reason I am asking this is because a lot of theorems you mentioned make assumptions about the ranges of information gap and I think some approximations on the information gap’s behavior on some standard or simulated datasets might help.

- Orthonormality assumption: line 193. Why do you make that assumption? To maximize span?

---

> ### Author Response · Authors · 2024-11-19
> **Response to Reviewer i2aQ  Part 1/2**
>
> Thank you for your time in reviewing our paper. In our revised version, we have highlighted all changes using red-colored text for clarity. Below we address your concerns
>
>
> > **Q1**.  Not sure where they conducted their experiments and where the results are.
>
> *a*. First, we would like to clarify that the observation mentioned on line 377 is meant to explain the theoretical expressions given in lines 370-373 (lines 398-401 in the revision), suggesting that the area selected by the theoretically optimal solution may not depend on the location of $p$. We reworded the sentence to make it easier to understand from the context.
>
> *b*. As we mention in the main body of the paper, we have included the experimental results in Appendix Section B. Please note that the primary focus of our paper is on the theoretical analysis of phenomena, which has been well supported by the extensive experiments provided in [1, 2]. Our own experiments in Appendix B mainly serve to further validate our theoretical findings as complement to the existing experiments. Further, our experiments also validate our proposed new empirical metric – "attention diversity metric"-- designed to assess the attention patterns observed after MAE and CL pretraining.
>
> *c*. As also suggested by Reviewer ChhH, in the revision, we have added the experimental findings including Figure 3 and its explanations (line 237-240) in the main body of the paper.
>
> > **Q2**.  This work lacks serious experimental validation. They need some kind of information bottleneck need experiments which play with the information gap and the size of the ViT tokens/patches. How do you measure this imbalance between local and global features?
>
> *a.* As we mention in the main body of the paper, we have included the experimental results in Appendix Section B. Please note that the primary focus of our paper is on the theoretical analysis of the observed phenomenon, which is supported by the extensive experiments in prior works [1, 2]. Our own experiments in Appendix B mainly serve to further validate our theoretical findings as a complement to the existing experiments. Further, our experiments also validate our proposed new empirical metric – "attention diversity metric"-- designed to assess the attention patterns observed after MAE and CL pretraining.
>
> *b.* Regarding the notion of information gap, our definition here is not related to the notion of information bottleneck. It refers to the (theoretical) difference between the sizes of the global and local feature areas. If such a difference is above a certain threshold (i.e., the information gap is larger than a threshold), then there is an imbalance of local and global features. Our experiments have been conducted with respect to such a metric. Information bottleneck-type experiments do not fit the concept that we aim to demonstrate.
>
> > **Q3.** The abstract mentions “analysis” and never mentions “convergence guarantees”. Initially, I thought this paper would have extensive experimentation documenting the global and local behavior of ViT pre-trained with different objectives. But the paper isn’t about that.
>
> We respectfully disagree with the reviewer to count this point as a weakness of the paper. As our title suggests, we focus on **theoretical analysis** rather than empirical analysis. Our abstract also discusses our contribution in the context of “**theoretical understanding** remains limited”. Further, as the reviewer commented in the strength part”, the paper contains “Extensive, well-formulated, and thorough proofs.” Overall, it should be clear that the main contributions here should be on the theory side. Nevertheless, we added **theoretical analysis** in several places in our abstract to avoid future confusion.

---

> > ### Author Response · Authors · 2024-11-19
> > **Response to Reviewer i2aQ Part 2/2**
> >
> > > **Q4.** Information gap: What are the bounds/range on this? Can we see a visualization or even a histogram that shows how this value changes for different sizes of local image patches/tokens? Maybe apply it to different samples from various datasets and comment on the behaviour observed? The reason I am asking this is because a lot of theorems you mentioned make assumptions about the ranges of information gap and I think some approximations on the information gap’s behavior on some standard or simulated datasets might help.
> >
> > *a.* The notion of the information gap in our study is primarily theoretical, designed to introduce and model an imbalance between features, abstracting the real data imbalances observed in practice. It is intended to provide high-level implications for understanding feature interactions in ViTs rather than serving as an empirical statistic that can be directly measured from datasets.
> >
> > *b.* In practice, finding a convincing and interpretable classification of global and local features in general datasets is challenging.  Prior empirical works [1-3] that studied the difference between MAE and CL mainly relied on attention-based visualization or divergence statistics to characterize the relationships between different image patches in ViTs. While these methods provide valuable insights, their methodologies also demonstrate the difficulty of finding a clear, empirically verifiable separation between global and local features.
> >
> > *c.* Consistent with these empirical studies, our experiments in Section B also utilize a similar attention-based approach to examine the interactions among patches. Our work aligns with the established methodologies in the field and further explores the theoretical mechanism that these visual and empirical observations suggest, advancing our understanding of the distinct roles and interactions within ViTs.
> > > **Q5.** Orthonormality assumption
> >
> > *a.* The orthonormal assumption is made to simplify the analysis, so that our characterization of the convergence and the dynamics of the attention correlations won't be over-complicated by non-essential aspects, e.g., additional non-dominant terms that need to be bounded in gradient calculations. Note that for the same reason, the orthonormal and discrete set of features have been widely adopted in previous studies on deep learning theory, e.g. [4-7]. Our goal with this assumption is not to maximize the span, since linearly correlated features span the same space as orthonormal features.
> >
> > *b.*  We recognize that our analysis could be extended to non-orthogonal settings, such as feature vectors independently drawn from a multivariate Gaussian distribution. However, we refrain from doing so to maintain clarity and focus, as it will not change our main message, avoiding the additional complexity.
> >
> > ***
> > Finally, we thank the reviewer again for your time and efforts. If possible, we kindly ask the reviewer to re-evaluate our paper based on its theoretical contributions and mathematical proofs. If our response resolves your concerns, we wonder if you could kindly consider raising the rating of our work. We are also more than happy to answer your further questions.
> >
> > [1] Park et al., What Do Self-Supervised Vision Transformers Learn?
> >
> > [2] Wei et al., Contrastive learning rivals masked image modeling in fine-tuning via feature distillation
> >
> > [3] Xie et al., Revealing the Dark Secrets of Masked Image Modeling
> >
> > [4] Allen-Zhu &Li., Towards Understanding Ensemble, Knowledge Distillation and Self-Distillation in Deep Learning.
> >
> > [5] Jelassi et al., Vision Transformers provably learn spatial structure.
> >
> > [6] Li et al., A Theoretical Understanding of Shallow Vision Transformers: Learning, Generalization, and Sample Complexity.
> >
> > [7] Li et al., How Do Nonlinear Transformers Learn and Generalize in In-Context Learning?

---

> > > ### Author Response · Authors · 2024-11-22
> > > **Follow Up Reminder**
> > >
> > > Dear Reviewer i2aQ,
> > >
> > > We've taken your initial feedback into careful consideration in our response. Could you kindly confirm whether our responses have appropriately addressed your concerns?
> > >
> > > If you find that we have properly addressed your concerns, could you please kindly consider increasing your initial score accordingly? Please let us know if you have further comments.
> > >
> > > Thank you for your time and effort in reviewing our work!
> > >
> > > Many thanks, Authors

---

### Author Response · Authors · 2024-11-19
**Revision Uploaded**

We have uploaded a revised version of the paper, which incorporates the reviewers' suggestions. Below is a summary of the changes made:

1. The experimental findings, including Figure 3 and its explanations, have been moved to the main body of the paper.
2. Notations and wording have been revised in several places as per the reviewers' suggestions, with all changes highlighted in red-colored texts in the revision.

We sincerely thank all the reviewers for their valuable comments and suggestions. We are happy to address any further questions or concerns during the discussion period.

---

### Meta-Review · Area_Chair_RF1B · 2024-12-20

**Metareview:**

Summary:
This paper presents a theoretical analysis of self-supervised learning for Vision Transformers (ViTs), focusing on the differences between contrastive learning (CL) and masked autoencoders (MAE). By analyzing the training dynamics of one-layer softmax-based self-attention using gradient descent on spatially structured data, the authors demonstrate that MAE effectively learns both global and local features, while CL tends to focus predominantly on global features, offering insights into why MAE can capture both global and subtle local information simultaneously, while CL tends to emphasize global patterns.

The paper's strengths are providing a comprehensive theoretical explanation for a behavior gap of ViTs pretrained with MAE and CL.
The main weaknesses raised by the reviewers are 1) too many assumptions in the explanation (using a single-layer transformer model) and 2) lacking serious experimental validation.

All reviewers recognized the contribution of this paper, introducing a novel theoretical framework and providing a theoretical explanation for analyzing the behavior gap of ViTs pretrained with MAE and CL.

Although concerns were raised about the assumptions (specifically, the one-layer self-attention model) used in the theoretical explanation, as noted by Reviewer 3zvq and WFmo, this paper makes significant strides in uncovering insights into the behavior of the dominant architecture, vision transformers (ViTs).

Recognizing the importance of encouraging theoretical analysis of ViT models, the Area Chair recommends accepting this paper. Additionally, the authors are encouraged to incorporate the necessary clarifications and discussions into the final version.

**Additional Comments On Reviewer Discussion:**

Reviewer 3zvq and WFmo expressed concerns about the assumptions (one layer of self-attention) used in the theoretical explanation and the lack of a reality check and insights for algorithm design. In response, the authors provided additional experiments and clarifications to address these issues.

After discussion, most of the concerns were addressed, however, Reviewer 3zvq still expressed reservations about the assumptions used in the theoretical explanation..

The final scores are 8, 6, 6, and 5.

---

### Decision · Program_Chairs · 2025-01-22

Accept (Poster)